# Adaptive Methods through the Lens of SDEs: Theoretical Insights on the Role of Noise

**Enea Monzio Compagnoni**[1]**, Tianlin Liu**[1]**, Rustem Islamov**[1]**,**
**Frank Norbert Proske**[2]**, Antonio Orvieto**[3,4,5]**, Aurelien Lucchi**[1]
[1]University of Basel, Switzerland; [2]University of Oslo, Norway;
[3]Max Planck Institute for Intelligent Systems, Germany
[4]ELLIS Institute Tübingen, Germany
[5]Tübingen AI Center, Germany
{enea.monziocompagnoni, tianlin.liu, rustem.islamov}@unibas.ch,
 proske@math.uio.no, antonio@tue.ellis.eu, aurelien.lucchi@unibas.ch

## Abstract

Despite the vast empirical evidence supporting the efficacy of adaptive optimization methods in deep learning, their theoretical understanding is far from complete. This work introduces novel SDEs for commonly used adaptive optimizers: SignSGD, RMSprop(W), and Adam(W). These SDEs offer a quantitatively accurate description of these optimizers and help illuminate an intricate relationship between adaptivity, gradient noise, and curvature. Our novel analysis of SignSGD highlights a noteworthy and precise contrast to SGD in terms of convergence speed, stationary distribution, and robustness to heavy-tail noise. We extend this analysis to AdamW and RMSpropW, for which we observe that the role of noise is much more complex. Crucially, we support our theoretical analysis with experimental evidence by verifying our insights: this includes numerically integrating our SDEs using Euler-Maruyama discretization on various neural network architectures such as MLPs, CNNs, ResNets, and Transformers. Our SDEs accurately track the behavior of the respective optimizers, especially when compared to previous SDEs derived for Adam and RMSprop. We believe our approach can provide valuable insights into best training practices and novel scaling rules.

## 1 Introduction

Adaptive optimizers lay the foundation for effective training of modern deep learning models. These methods are typically employed to optimize an objective function expressed as a sum of losses across $N$ individual data points: $\min_{x \in \mathbb{R}^d}[f(x) := \frac{1}{N}\sum_{i=1}^{N} f_i(x)]$, where $f, f_i : \mathbb{R}^d \to \mathbb{R}, \ i = 1, \ldots, N$. Due to the practical difficulties of selecting the learning rate of stochastic gradient descent, adaptive methods have grown in popularity over the past decade. At a high level, these optimizers adjust the learning rate for each parameter based on the historical gradients. Popular optimizers that belong to this family are RMSprop (Tieleman and Hinton, 2012), Adam (Kingma and Ba, 2015), SignSGD (Bernstein et al., 2018), AdamW (Loshchilov and Hutter, 2019), and many other variants. SignSGD is often used for compressing gradients in distributed machine learning (Karimireddy et al., 2019a), but it also has gained popularity due to its connection to RMSprop and Adam (Balles and Hennig, 2018). The latter algorithms have emerged as the standard methods for training modern large language models, partly because of enhancements in signal propagation (Noci et al., 2022).

Although adaptive methods are widely favored in practice, their theoretical foundations remain enigmatic. Recent research has illuminated some of their advantages: Zhang et al. (2020b) demonstrated how gradient clipping addresses heavy-tailed gradient noise, Pan and Li (2022) related the success of Adam over SGD to sharpness, and Yang et al. (2024) showed that adaptive methods are more resilient to poor learning rate tuning than SGD. At the same time, many optimization studies focus on worst-case convergence rates: These rates (e.g., Défossez et al. (2022)) are valuable, yet they provide an incomplete depiction of algorithm behavior, showing no quantifiable advantage over standard SGD. One particular aspect still lacking clarity is the precise role of noise in the algorithm trajectory.

Our investigation aims to study how gradient noise influences the dynamics of adaptive optimizers and how it impacts their asymptotic behaviors in terms of expected loss and stationary distribution. In

particular, we want to understand which algorithms are more resilient to high (possibly heavy-tailed) gradient noise levels. To do this, we rely on stochastic differential equations (SDEs) which have become popular in the literature to study the behavior of optimization algorithms (Li et al., 2017; Jastrzebski et al., 2018). These continuous-time models unlock powerful tools from Itô calculus, enabling us to establish convergence bounds, determine stationary distributions, unveil implicit regularization, and elucidate the intricate interplay between landscape and noise. Notably, SDEs facilitate direct comparisons between optimizers by explicitly illustrating how each hyperparameter and certain landscape features influence their dynamics (Orvieto and Lucchi, 2019; Malladi et al., 2022; Compagnoni et al., 2023; 2024; 2025).

We begin by analyzing SignSGD, showing how the ratio between the gradient and the level of gradient noise affects its dynamics and elucidating the impact of noise at convergence. After examining the case where the gradient noise has an infinite variance, we extend our analysis to Adam and RMSprop with *decoupled* weight decay (Loshchilov and Hutter, 2019) – i.e. AdamW and RMSpropW: for both, we refine batch size scaling rules and compare the role of noise to SignSGD. Our analysis provides some theoretical grounding for the resilience of these adaptive methods to high noise levels. Importantly, we highlight that Adam and RMSprop are byproducts of our analysis and that our novel SDEs are derived under weaker assumptions than those in the literature (Zhou et al., 2020a; Malladi et al., 2022).

**Contributions** We identify our key contributions as follows:

1. We derive the first[1] SDE for SignSGD under very general assumptions: We show that SignSGD exhibits three different phases of the dynamics and characterize the loss behavior in these phases, including the stationary distribution and asymptotic loss value;

2. We prove that for SignSGD, noise inversely affects the convergence rate of both the loss and the iterates. Differently, it has a linear impact on the asymptotic expected loss and the asymptotic variance of the iterates. This is in contrast to SGD, where noise does not influence the convergence speed, but it has a quadratic effect on the loss and variance of the iterates. Finally, we show that, even if the noise has infinite variance, SignSGD is resilient: its performance is only marginally impacted. In the same conditions, SGD diverges;

3. We derive new, improved, SDEs for AdamW and RMSpropW and use them to (i) show a novel batch size scaling rule and (ii) inspect the stationary distribution and stationary loss value in convex quadratics. In particular, we dive into the properties of weight decay: while for vanilla Adam and RMSprop the effect of noise at convergence mimics SignSGD, something different happens in AdamW and RMSpropW — Due to an intricate interaction between noise, curvature, and regularization, *decoupled* weight decay plays a crucial stabilization role at high noise levels near the minimizer;

4. We empirically verify every theoretical insight we derive. Importantly, we integrate our SDEs with Euler-Maruyama to confirm that our SDEs faithfully track their respective optimizers. We do so on an MLP, a CNN, a ResNet, and a Transformer. For RMSprop and Adam, our SDEs exhibit superior modeling power than the SDEs already in the literature. We emphasize that while our results rely on certain regularity assumptions for loss functions and gradient noise, their applicability extends beyond these. For example, we validate our novel scaling rule for AdamW on a Pythia-like 160M LLM (Biderman et al., 2023) trained on $2.5B/10B$ tokens from the SlimPajama dataset (Soboleva et al., 2023).

## 2 RELATED WORK

**SDE approximations and applications.** (Li et al., 2017) introduced a formal theoretical framework aimed at deriving SDEs that effectively model the inherent stochastic nature of optimizers. Ever since, SDEs have found several applications in the field of machine learning, for instance in connection with *stochastic optimal control* to select the stepsize (Li et al., 2017; 2019) and batch size (Zhao et al., 2022), the derivation of *convergence bounds* and *stationary distributions* (Compagnoni et al., 2023; 2024), *implicit regularization* (Smith et al., 2021), and *scaling rules* (Jastrzebski et al., 2018). Previous work by Malladi et al. (2022) has already made strides in deriving SDE models for RMSprop and Adam, albeit under somewhat restrictive assumptions. They establish a scaling rule that they

---

[1]In a concurrent work, Xiao et al. (2024) derived an SDE for SignSGD in the high dimensional setting for a linear regression task: See Appendix F in Xiao et al. (2024) for a comparison with our SDE.

assert remains valid throughout the entirety of the dynamics. While their derivation builds on the approach of Jastrzebski et al. (2018), which may be problematic in the general setting (see Appendix E for further discussion), our analysis suggests that the SDEs proposed in Malladi et al. (2022) provide accurate approximations primarily near minima, implying that the corresponding scaling rules might not hold globally. (Zhou et al., 2020a) derived a Lévy SDE for Adam; however, the approximation relies on random coefficients, a technique that is theoretically sound only under very specific conditions (see Kohatsu-Higa et al. (1997); Bishop and Del Moral (2019)). Xie et al. (2022) modeled AdamW with an SDE to investigate the roles of learning rate adaptivity and momentum in saddle-point escaping and flat minima selection. However, since their derivation does not follow a formal framework, it currently lacks comprehensive approximation guarantees. Finally, Zhou et al. (2024) presented an informal SDE for the iterates of AdamW: Their derivation relies on several strong assumptions and approximations, which may benefit from further formal justification.

**Influence of noise on convergence.**    Several empirical papers demonstrate that adaptive algorithms adjust better to the noise during training. Specifically, (Zhang et al., 2020b) noticed a consistent gap in the performance of SGD and Adam on language models and connected that phenomenon with heavy-tailed noise distributions. (Pascanu et al., 2013) suggests using gradient clipping to deal with heavy tail noise, and consequently several follow-up works analyzed clipped SGD under heavy-tailed noise (Zhang et al., 2020a; Mai and Johansson, 2021; Puchkin et al., 2024). Kunstner et al. (2024) present thorough numerical experiments illustrating that a significant contributor to heavy-tailed noise during language model training is class imbalance, where certain words occur much more frequently than others. They demonstrate that adaptive optimization methods such as Adam and SignSGD can better adapt to such class imbalances. However, the theoretical understanding of the influence of noise in the context of adaptive algorithms is much more limited. The first convergence results on Adam and RMSprop were derived under bounded stochastic gradients assumption (De et al., 2018; Zaheer et al., 2018; Chen et al., 2019; Défossez et al., 2022). Later, this noise model was relaxed to weak growth condition (Zhang et al., 2022; Wang et al., 2022) and its coordinate-wise version (Hong and Lin, 2023; Wang et al., 2024) and sub-gaussian noise (Li et al., 2023a). SignSGD and its momentum version Signum were originally studied as a method for compressed communication (Bernstein et al., 2018) under bounded variance assumption, but with a requirement of large batches. Several works provided counterexamples where SignSGD fails to converge if stochastic and full gradients are not correlated enough (Karimireddy et al., 2019b; Safaryan and Richtarik, 2021). In the case of AdamW, (Zhou et al., 2022; 2024) provided convergence guarantees under restrictive assumptions such as bounded gradient and bounded noise. All aforementioned results only show that SignSGD, Adam, and RMSprop at least do not perform worse than vanilla SGD. None of them studied how noise affects the dynamics of the algorithm: In this work, we attempt to close this gap.

## 3 FORMAL STATEMENTS & INSIGHTS: THE SDES

This section provides the general formulations of the SDEs of SignSGD (Theorem 3.2) and AdamW (Theorem 3.12). Due to the technical nature of the analysis, we refer the reader to the appendix for the complete formal statements and proofs and only provide a sketch of the proof of key results.

**Assumptions and notation.**    In this section, we collect most of the notation and assumptions used in the paper. All our analysis take place on a filtered probability space $(\Omega, \mathcal{F}, \{\mathcal{F}_t\}_{t \geq 0}, \mathbb{P})$. The batches $\gamma$ are of size $B \geq 1$ and modeled as i.i.d. random variables uniformly distributed on $\{1, \ldots, N\}$. We assume that the stochastic gradient $\nabla f_\gamma(x) := \frac{1}{B} \sum_{i \in \gamma} \nabla(f_i(x))$ can be decomposed as $\nabla f(x) + Z(x)$, where $\nabla f(x)$ is the full gradient and $Z(x)$ is the batch noise. We assume that $\mathbb{E}[Z(x)] = 0$ and unless we study the cases where the gradient variance is unbounded, we write $Cov(Z(x)) = \Sigma(x)$ (we omit the size of the batch $\gamma$ unless relevant.) s.t. $\sqrt{\Sigma(x)}$ is bounded, Lipschitz, satisfies affine growth, and together with its derivatives, it grows at most polynomially fast (Definition 2.5 in Malladi et al. (2022)). Importantly, we assume that $Z(x)$ has a bounded and smooth probability density function whose derivatives are all integrable: A common assumption in the literature is for $Z(x)$ to be Gaussian[2] (Ahn et al., 2012; Chen et al., 2014; Mandt et al., 2016; Stephan et al., 2017; Zhu et al., 2019; Wu et al., 2020; Xie et al., 2021), while our assumption allows for heavy-tailed distributions such as the Student's t. Specifically, Li et al. (2017); Mertikopoulos and Staudigl (2018); Raginsky and Bouvrie (2012); Zhu et al. (2019); Mandt et al. (2016); Ahn et al. (2012); Jastrzebski et al. (2018) use a Gaussian noise with constant covariance matrix to model batch noise. To derive the stationary distribution around an optimum, we approximate the loss function

---

[2]See Jastrzebski et al. (2018) for the justification why this might be the case.

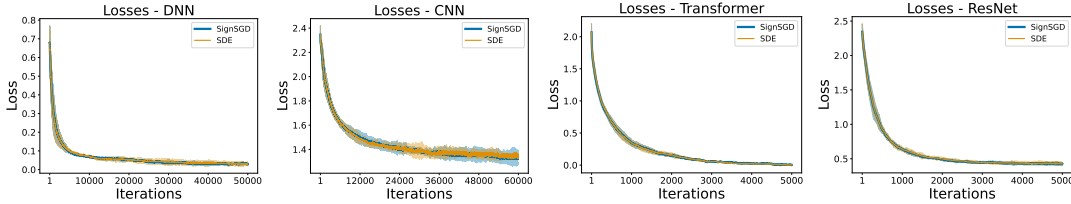

Figure 1: Comparison of SignSGD and its SDE in terms of $f(x)$: Our SDE successfully tracks the dynamics of SignSGD on several architectures, datasets, and hyperparameters: DNN on the Breast Cancer dataset (Left); CNN on MNIST (Center-Left); Transformer on MNIST (Center-Right); ResNet on CIFAR-10 (Right).

with a quadratic convex function $f(x) = \frac{1}{2}x^\top H x$ as commonly done in the literature (Ge et al., 2015; Levy, 2016; Jin et al., 2017; Poggio et al., 2017; Mandt et al., 2017; Compagnoni et al., 2023). Finally, $\eta > 0$ is the step size, the $\bar{\beta}$s refer to momentum parameters, $\theta > 0$ is the (decoupled) $L^2$-regularization parameter, and $\epsilon > 0$ is a scalar used for numerical stability. Finally, we use $W_t$ to indicate a Brownian motion.

The following definition formalizes the idea that an SDE can be a "good model" to describe an optimizer. It is drawn from the field of numerical analysis of SDEs (see Mil'shtein (1986)) and it quantifies the disparity between the discrete and the continuous processes.

**Definition 3.1** (Weak Approximation). *A continuous-time stochastic process $\{X_t\}_{t\in[0,T]}$ is an order $\alpha$ weak approximation (or $\alpha$-order SDE) of a discrete stochastic process $\{x_k\}_{k=0}^{\lfloor T/\eta \rfloor}$ if for every polynomial growth function $g$, there exists a positive constant $C$, independent of the stepsize $\eta$, such that $\max_{k=0,\ldots,\lfloor T/\eta \rfloor} |\mathbb{E}g(x_k) - \mathbb{E}g(X_{k\eta})| \leq C\eta^\alpha$.*

### 3.1 SIGNSGD SDE

In this section, we derive an SDE model for SignSGD, which we believe to be a novel addition to the existing literature. This derivation will reveal the unique manner in which noise influences the dynamics of SignSGD. First, we recall the update equation of SignSGD:

$$x_{k+1} = x_k - \eta \operatorname{sign}\left(\nabla f_{\gamma_k}(x_k)\right). \tag{1}$$

The following theorem derives a formal continuous-time model for SignSGD.

**Theorem 3.2** (Informal Statement of Theorem C.16). *Under sufficient regularity conditions, the solution of the following SDE is an order $1$ weak approximation of the discrete update of SignSGD:*

$$dX_t = -(1 - 2\mathbb{P}(\nabla f_\gamma(X_t) < 0))dt + \sqrt{\eta}\sqrt{\bar{\Sigma}(X_t)}dW_t, \tag{2}$$

*where $\bar{\Sigma}(x)$ is the noise covariance $\bar{\Sigma}(x) = \mathbb{E}[\xi_\gamma(x)\xi_\gamma(x)^\top]$, and $\xi_\gamma(x) := \operatorname{sign}(\nabla f_\gamma(x)) - 1 + 2\mathbb{P}(\nabla f_\gamma(x) < 0)$ is the noise of $\operatorname{sign}\left(\nabla f_\gamma(x)\right)$.*

*Proof idea.* One needs to prove that the first and second moments of the increments of the discretization of the SDE match those of SignSGD up to an error of order $\mathcal{O}(\eta)$ and $\mathcal{O}(\eta^2)$, respectively. □

For **didactic reasons**, we next present a corollary of Theorem 3.2 that provides a more interpretable SDE. To do so, we model the batch noise with a Gaussian distribution with constant covariance matrix,[3] which is a common approach in the literature (Li et al., 2017; Mertikopoulos and Staudigl, 2018; Raginsky and Bouvrie, 2012; Zhu et al., 2019; Mandt et al., 2016; Ahn et al., 2012; Jastrzebski et al., 2018). Figure 1 shows the empirical validation of this model for various neural network classes: All details are presented in Appendix F.

**Corollary 3.3** (Informal Statement of Corollary C.19). *Under the assumptions of Theorem 3.2, and that the stochastic gradient is $\nabla f_\gamma(x) = \nabla f(x) + Z$ such that $Z \sim \mathcal{N}(0, \Sigma)$, $\Sigma = \operatorname{diag}(\sigma_1^2, \cdots, \sigma_d^2)$, the following SDE provides a $1$ weak approximation of the discrete update of SignSGD*

$$dX_t = -Erf\left(\frac{\Sigma^{-\frac{1}{2}}\nabla f(X_t)}{\sqrt{2}}\right)dt + \sqrt{\eta}\sqrt{I_d - \operatorname{diag}\left(Erf\left(\frac{\Sigma^{-\frac{1}{2}}\nabla f(X_t)}{\sqrt{2}}\right)\right)^2}dW_t, \tag{3}$$

*where the error function $Erf(x) := \frac{2}{\sqrt{\pi}}\int_0^x e^{-t^2}dt$ and the square are applied component-wise.*

---

[3]See Section C.5 for more realistic noise structures.

While Eq. 3 may appear intricate at first glance, it becomes apparent upon closer inspection that the properties of the Erf($\cdot$) function enable a detailed exploration of the dynamics of SignSGD. In particular, we demonstrate that the dynamics of SignSGD can be categorized into three distinct phases. The left of Figure 2 empirically verifies this result on a convex quadratic function.

**Lemma 3.4.** *Under the assumptions of Coroll. 3.3, $Y_t := \frac{\Sigma^{-\frac{1}{2}}\nabla f(X_t)}{\sqrt{2}}$, $|\cdot|$ applied element-wise, and some constants[4] $m \in \mathbb{R}^+$, $\mathbf{q}^+ \in \mathbb{R}^d$, $\mathbf{q}^- \in \mathbb{R}^d$, the dynamics of SignSGD exhibits three **phases**:*

1. ***Phase 1:** If $|Y_t| > \frac{3}{2}$, the SDE coincides with the ODE of SignGD:*

$$dX_t = -\text{sign}(\nabla f(X_t))dt; \tag{4}$$

2. ***Phase 2:** If $1 < |Y_t| < \frac{3}{2}$:*

   (a) $-mY_t - \mathbf{q}^+ \le \frac{d\mathbb{E}[X_t]}{dt} \le -mY_t - \mathbf{q}^-$;
   
   (b) *For any $a > 0$, $\mathbb{P}\left[\|X_t - \mathbb{E}[X_t]\|_2^2 > a\right] \le \frac{\eta}{a}\left(d - \|mY_t + \mathbf{q}^-\|_2^2\right)$;*

3. ***Phase 3:** If $|Y_t| < 1$, the SDE is*

$$dX_t = -\sqrt{\frac{2}{\pi}}\Sigma^{-\frac{1}{2}}\nabla f(X_t)dt + \sqrt{\eta}\sqrt{I_d - \frac{2}{\pi}\text{diag}\left(\Sigma^{-\frac{1}{2}}\nabla f(X_t)\right)^2}dW_t. \tag{5}$$

**Remark:** *For ease of reading*, we will *informally* refer to the gradient $\nabla f(x)$ as the "signal" and to $\Sigma$ as the "noise". Then, Lemma 3.4 tells us that the behavior of SignSGD depends on the size of the "signal-to-noise" ratio. In particular, the SDE itself shows that in Phase 3, the inverse of the scale of the noise $\Sigma^{-\frac{1}{2}}$ premultiplies $\nabla f(x)$, affecting the rate of descent. This is not the case for SGD where $\Sigma$ only influences the diffusion term.[5] To better understand the role of the noise, we study how it affects the dynamics of the loss on strongly convex functions and compare it with SGD. The dynamics of $\mathbb{E}\left[\|\nabla f(X_t)\|_2^2\right]$ for general non-convex smooth functions is presented in Lemma C.24.

**Lemma 3.5.** *Let $f$ be $\mu$-strongly convex, $Tr(\nabla^2 f(x)) \le \mathcal{L}_\tau$, $\sigma_{max}^2$ be the maximum eigenvalue of $\Sigma$, and $S_t := f(X_t) - f(X_*)$. Then, during*

1. ***Phase 1**, $S_t \le \frac{1}{4}\left(\sqrt{\mu}t - 2\sqrt{S_0}\right)^2$, so SignSGD stays in this phase for at most $t_* = 2\sqrt{\frac{S_0}{\mu}}$;*

2. ***Phase 2** as $\Delta := \left(\frac{m}{\sqrt{2}\sigma_{max}} + \frac{\eta\mu m^2}{4\sigma_{max}^2}\right)$:*

$$\mathbb{E}[S_t] \le S_0 e^{-2\mu\Delta t} + \frac{\eta}{2}\frac{\left(\mathcal{L}_\tau - \mu d\hat{q}^2\right)}{2\mu\Delta}\left(1 - e^{-2\mu\Delta t}\right);$$

3. ***Phase 3** as $\Delta := \left(\sqrt{\frac{2}{\pi}}\frac{1}{\sigma_{max}} + \frac{\eta}{\pi}\frac{\mu}{\sigma_{max}^2}\right)$:*

$$\mathbb{E}[S_t] \le S_0 e^{-2\mu\Delta t} + \frac{\eta}{2}\frac{\mathcal{L}_\tau}{2\mu\Delta}\left(1 - e^{-2\mu\Delta t}\right).$$

*Proof idea.* For each **Phase**, we use the respective SDE of SignSGD from Lemma 3.4 to derive the SDE of $S_t$ via Itô's lemma. Then, we take its expectation to obtain the ODE of $\mathbb{E}[S_t]$ and leverage the assumptions to establish a bound. $\square$

As per Eq. 4, during Phase 1 SignSGD behaves like SignGD: Lemma 3.5 shows that, consistently with the analysis of SignGD in (Ma et al., 2022), such a strong decrease in the loss value explains the fast initial convergence of the optimizer as well as of RMSprop and Adam. In this phase, the loss undergoes a decrease which ensures the emergence of Phase 2 which in turn triggers that of Phase 3 which is characterized by an exponential decay to an asymptotic loss level: As a practical example, we verify the dynamics of the expected loss around a minimum in the center-left of Figure 2.

---

[4]See Lemma C.21 for their definitions.

[5]The SDE of SGD is $dX_t = -\nabla f(X_t)dt + \sqrt{\eta}\Sigma^{\frac{1}{2}}dW_t$.

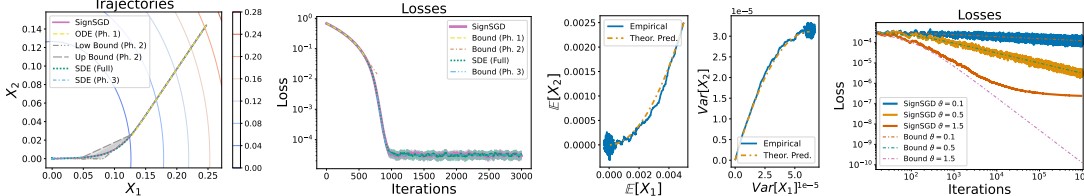

Figure 2: **Phases of SignSGD**: The ODE of Phase 1 and the SDE of Phase 3 overlap with the "Full" SDE as per **Lemma 3.4**. In Phase 2, the dynamics satisfies the prescribed bounds (Left); **Phases of the Loss**: The bounds derived in **Lemma 3.5** for the loss during the different phases correctly track the loss evolution (Center-Left); The **dynamics of the moments** of $X_t$ predicted in **Lemma 3.7** track the empirical ones (Center-Right); If the **schedulers** satisfy the condition in **Lemma 3.9**, the loss decays to 0 as prescribed. Otherwise, the loss does not converge to 0 (Right). For each figure, $f(x) = \frac{x^\top H x}{2}$ for $H = \mathrm{diag}(1, 2)$, $\eta = 0.001$, and $\Sigma = \sigma^2 I_2$ where $\sigma = 0.1$.

**Lemma 3.6.** *For SGD, the expected loss satisfies:* $\mathbb{E}[S_t] \leq S_0 e^{-2\mu t} + \frac{\eta}{2} \frac{\mathcal{L}_\tau \sigma_{max}^2}{2\mu} \left(1 - e^{-2\mu t}\right).$

**Remark:** The two key observations are that:

1. Both in Phase 2 and Phase 3, the noise level $\sigma_{max}$ inversely affects the exponential convergence speed, while this trend is not observed with SGD;

2. The asymptotic loss of SignSGD is (almost) linear in $\sigma_{max}$ while that of SGD is quadratic. Indeed, the asymptotic value of $\mathbb{E}[S_t]$ in Phase 3 scales with $\frac{1}{\Delta} = \frac{\pi \sigma_{max}}{\sqrt{2\pi} + \frac{\eta \mu}{\sigma_{max}}}$: when the noise $\sigma_{max}$ dominates the learning rate $\eta$ and/or the minimum eigenvalue $\mu$ of the Hessian, or in general when $\frac{\eta \mu}{\sigma_{max}} \sim 0$, we can conclude that the scaling is (almost) linear in $\sigma_{max}$.

Additionally, we characterize the stationary distribution of SignSGD around a minimum. To do this, we study the behavior of SignSGD on a quadratic loss function, which is a common approach in the literature (Ge et al., 2015; Levy, 2016; Jin et al., 2017; Poggio et al., 2017; Mandt et al., 2017; Compagnoni et al., 2023). Empirical validation is provided in the center-right of Figure 2.

**Lemma 3.7.** *Let* $H = \mathrm{diag}(\lambda_1, \ldots, \lambda_d)$ *and* $M_t := e^{-2\left(\sqrt{\frac{2}{\pi}} \Sigma^{-\frac{1}{2}} H + \frac{\eta}{\pi} \Sigma^{-1} H^2\right) t}$. *Then,*

*1.* $\mathbb{E}[X_t] = e^{-\sqrt{\frac{2}{\pi}} \Sigma^{-\frac{1}{2}} H t} X_0;$

*2.* $Cov[X_t] = \left(M_t - e^{-2\sqrt{\frac{2}{\pi}} \Sigma^{-\frac{1}{2}} H t}\right) X_0^2 + \frac{\eta}{2} \left(\sqrt{\frac{2}{\pi}} I_d + \frac{\eta}{\pi} H \Sigma^{-\frac{1}{2}}\right)^{-1} H^{-1} \Sigma^{\frac{1}{2}} (I_d - M_t).$

*Therefore, we have that the stationary distribution of SignSGD is:*

$$(\mathbb{E}[X_\infty], Cov[X_\infty]) = \left(0, \frac{\eta}{2} \left(\sqrt{\frac{2}{\pi}} I_d + \frac{\eta}{\pi} H \Sigma^{-\frac{1}{2}}\right)^{-1} H^{-1} \Sigma^{\frac{1}{2}}\right).$$

*Proof idea.* For the $\mathbb{E}[X_t]$, we take the expected value of the SDE of Phase 3 from Lemma 3.4 and integrate the resulting ODE. For $Cov[X_t]$, we derive the SDE of $X_t X_t^\top$ via Itô's lemma, take the expectation, and integrate the resulting ODE. Then, we subtract $\mathbb{E}[X_t]\mathbb{E}[X_t]^\top$. ☐

**Lemma 3.8.** *Under the same assumptions as Lemma 3.7, the stationary distribution for SGD is:*

$$\mathbb{E}[X_t] = e^{-Ht} X_0 \overset{t \to \infty}{\to} 0 \quad and \quad Cov[X_t] = \frac{\eta}{2} H^{-1} \Sigma \left(I_d - e^{-2Ht}\right) \overset{t \to \infty}{\to} \frac{\eta}{2} H^{-1} \Sigma.$$

As we observed above, the noise inversely affects the convergence rate of the iterates of SignSGD while it does not impact that of SGD. Additionally, while both covariance matrices essentially scale inversely to the Hessian, that of SignSGD scales with $\Sigma^{\frac{1}{2}}$ while that of SGD scales with $\Sigma$.

We conclude this section by presenting a condition on the step size scheduler that ensures the asymptotic convergence of the expected loss to 0 in Phase 3. For general schedulers, we characterize precisely the speed of convergence and the factors influencing it. Empirical validation is provided in the right of Figure 2 for a convex quadratic as we use $\eta_t^\vartheta = \frac{1}{(t+1)^\vartheta}$ for $\vartheta \in \left\{\frac{1}{10}, \frac{1}{2}, \frac{3}{2}\right\}$.

**Lemma 3.9.** *Under the assumptions of Lemma 3.5, any step size scheduler $\eta_t$ such that*

$$\int_0^\infty \eta_s ds = \infty \text{ and } \lim_{t\to\infty} \eta_t = 0 \implies \mathbb{E}[f(X_t) - f(X_*)] \overset{t\to\infty}{\lesssim} \frac{\mathcal{L}_\tau \sigma_{max}}{4\mu}\sqrt{\frac{\pi}{2}}\eta_t \overset{t\to\infty}{\to} 0. \quad (6)$$

**Remark:** Under the same conditions, SGD satisfies $\mathbb{E}[f(X_t) - f(X_*)] \overset{t\to\infty}{\lesssim} \frac{\mathcal{L}_\tau \sigma_{max}^2}{4\mu}\eta_t \overset{t\to\infty}{\to} 0.$

**Conclusion:** As noted in Bernstein et al. (2018), the "signal-to-noise" ratio is key in determining the dynamics of SignSGD. Our SDEs help clarify the mechanisms underlying the dynamics of SignSGD: we show that the effect of noise is radically different from SGD: 1) It affects the rate of convergence of the iterates, of the covariance of the iterates, and of the expected loss; 2) The asymptotic loss value and covariance of the iterates scale in $\Sigma^{\frac{1}{2}}$ while for SGD it does so in $\Sigma$. On the one hand, low levels of noise will ensure a faster and steadier loss decrease close to minima for SignSGD than for SGD. On the other, SGD will converge to much lower loss values. A symmetric argument holds for high levels of noise, which suggests that SignSGD is more resilient to high levels of noise.

### 3.1.1 HEAVY-TAILED NOISE

Interestingly, we can replicate the efforts above also in case the noise $Z(x)$ is heavy-tailed as it is distributed according to a Student's t distribution. Notably, we derive the SDE for the case where the noise has infinite variance and show how little marginal effect this has on the dynamics of SignSGD.

**Lemma 3.10.** *Under the assumptions of Corollary 3.3 where the noise on the gradients $Z \sim t_\nu(0, I_d)$ and $\nu \in \mathbb{Z}^+$, the following SDE is a 1 weak approximation of the discrete update of SignSGD*

$$dX_t = -2\Xi\left(\Sigma^{-\frac{1}{2}}\nabla f(X_t)\right) dt + \sqrt{\eta}\sqrt{I_d - 4\operatorname{diag}\left(\Xi\left(\Sigma^{-\frac{1}{2}}\nabla f(X_t)\right)\right)^2} dW_t, \quad (7)$$

*where $\Xi(x)$ is defined as $\Xi(x) := x\frac{\Gamma\left(\frac{\nu+1}{2}\right)}{\sqrt{\pi\nu}\Gamma\left(\frac{\nu}{2}\right)} {}_2F_1\left(\frac{1}{2}, \frac{\nu+1}{2}; \frac{3}{2}; -\frac{x^2}{\nu}\right)$, $\Gamma$ is the gamma function, and ${}_2F_1$ is the hypergeometric function. Above, the $\Xi(x)$ and the square are applied component-wise.*

We now characterize the dynamics of SignSGD when the noise on the gradient has infinite variance.

**Corollary 3.11.** *Under the assumptions of Lemma 3.10 and $\nu = 2$, the dynamics in Phase 3 is:*

$$dX_t = -\sqrt{\frac{1}{2}}\Sigma^{-\frac{1}{2}}\nabla f(X_t)dt + \sqrt{\eta}\sqrt{I_d - \frac{1}{2}\operatorname{diag}\left(\Sigma^{-\frac{1}{2}}\nabla f(X_t)\right)^2} dW_t. \quad (8)$$

**Conclusion:** We observe that the dynamics of SignSGD when the noise is Gaussian (Eq. 5) w.r.t. when it is heavy-tailed with unbounded variance (Eq. 8) are very similar: By comparing the constants ($\sqrt{1/2}$ and $\sqrt{2/\pi}$) in front of the drift terms $\Sigma^{-\frac{1}{2}}\nabla f(X_t)$, they are only $\sim 10\%$ apart, and the diffusion coefficients are comparable. Not only do we once more showcase the resilience of SignSGD to high levels of noise, but in alignment with (Zhang et al., 2020b), we provide theoretical support to the success of Adam in such a scenario where SGD would diverge.

All the results derived above can be extended to this heavy-tailed noise setting: See Compagnoni et al. (2025) for a detailed discussion in the distributed setting.

### 3.2 ADAMW SDE

In the last subsection, we showcased how SDEs can serve as powerful tools to understand the dynamics of the simplest among coordinate-wise adaptive methods: SignSGD. Here, we extend the discussion to Adam with *decoupled* weight decay, i.e. AdamW:

$$v_{k+1} = \beta_2 v_k + (1-\beta_2)\left(\nabla f_{\gamma_k}(x_k)\right)^2, \quad m_{k+1} = \beta_1 m_k + (1-\beta_1)\nabla f_{\gamma_k}(x_k),$$

$$x_{k+1} = x_k - \eta\frac{\hat{m}_{k+1}}{\sqrt{\hat{v}_{k+1}} + \epsilon} - \eta\theta x_k, \quad \hat{m}_k = \frac{m_k}{1-\beta_1^k}, \quad \hat{v}_k = \frac{v_k}{1-\beta_2^k}, \quad (9)$$

which, of course, covers Adam, RMSprop, and RMSpropW depending on the values of $\theta$ and $\beta_1$.

The following result proves the SDE of AdamW which we validate in Figure 3 for two simple landscapes and in Figure 4 for a Transformer and a ResNet.

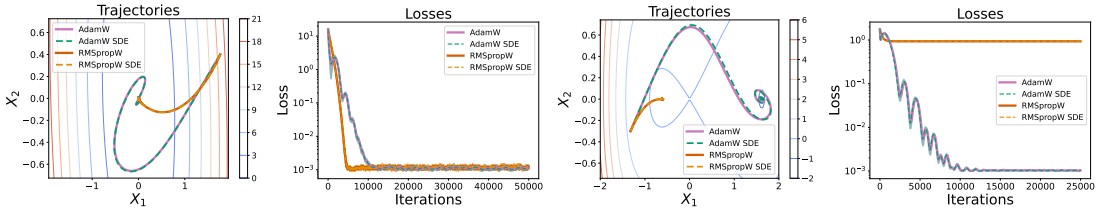

Figure 3: The two images on the left compare the SDEs of AdamW and RMSpropW with the respective optimizers in terms of trajectories and $f(x)$ for a convex quadratic function while the other two provide a comparison for an embedded saddle. In all cases, we observe good agreements.

**Theorem 3.12** (Informal Statement of Theorem C.53). *Under sufficient regularity conditions, $\rho_1 = \mathcal{O}(\eta^{-\varsigma})$ s.t. $\varsigma \in (0, 1)$, and $\rho_2 = \mathcal{O}(1)$, the order 1 weak approximation of AdamW is:*

$$dX_t = -\frac{\sqrt{\iota_2(t)}}{\iota_1(t)} P_t^{-1}(M_t + \eta\rho_1\left(\nabla f\left(X_t\right) - M_t\right))dt - \theta X_t dt \tag{10}$$

$$dM_t = \rho_1\left(\nabla f\left(X_t\right) - M_t\right)dt + \sqrt{\eta}\rho_1\sqrt{\Sigma\left(X_t\right)}dW_t \tag{11}$$

$$dV_t = \rho_2\left((\nabla f(X_t))^2 + \text{diag}\left(\Sigma\left(X_t\right)\right) - V_t\right)dt, \tag{12}$$

*where $\beta_i = 1 - \eta\rho_i \sim 1$, $\iota_i(t) = 1 - e^{-\rho_i t}$, $t > t_0$, and $P_t = \text{diag}\sqrt{V_t} + \epsilon\sqrt{\iota_2(t)}I_d$.*

$M_t$ and $V_t$ are the exponential moving averages of the gradient and the squared gradient, respectively. $P_t^{-1}$ acts as an adaptive preconditioner, scaling the parameter updates $X_t$ based on the accumulated squared gradients in $V_t$. While $M_t$ is the momentum term and captures the history of gradients to smooth out the updates, $\eta\rho_1\left(\nabla f\left(X_t\right) - M_t\right)$ adjusts $M_t$ towards the current gradient, ensuring responsiveness to recent changes. Finally, $-\theta X_t\, dt$ applies regularization by shrinking the parameters.

We highlight that in contrast to *Remark 4.3* of Malladi et al. (2022), which suggests that an SDE for Adam is only viable if $\sigma \gg \|\nabla f(x)\|$ and $\sigma \sim \frac{1}{\eta}$, our derivation that does not need these assumptions: See Remark C.46 for a deeper discussion, the implications, and the experimental comparisons.

The following result demonstrates how the asymptotic expected loss of AdamW scales with the noise level. Notably, it introduces the first scaling rule for AdamW, proposing an alternative to the one proposed for Adam in (Malladi et al., 2022) and extending it to include weight decay scaling. It is crucial to understand that, unlike the typical approach in the literature (see (Jastrzebski et al., 2018; Malladi et al., 2022)), our objective in deriving these rules is not to maintain the dynamics of the optimizers or the SDE unchanged. Instead, our goal is to offer a practical strategy for adjusting hyperparameters (e.g., from $\eta$ to $\tilde{\eta}$) to retain certain performance metrics or optimizer properties as the batch size increases (e.g., from $B$ to $\tilde{B}$). Therefore, in our upcoming analysis, we aim to derive scaling rules that *preserve* specific relevant aspects of the dynamics, such as the convergence bound on the loss or the speed. See Appendix E for a more detailed discussion motivating our approach.

**Lemma 3.13.** *If $f$ is $\mu$-strongly convex and $L$-smooth, $\text{Tr}(\nabla^2 f(x)) \leq \mathcal{L}_\tau$, $X_* = 0$, $\Sigma(x) = \sigma^2 I_d$, and $(\nabla f(x))^2 = \mathcal{O}(\eta)$, $\tilde{\eta} = \kappa\eta$, $\tilde{B} = B\delta$, and $\tilde{\rho}_i = \alpha_i\rho_i$, and $\tilde{\theta} = \xi\theta$, AdamW satisfies*

$$\mathbb{E}[f(X_t) - f(X_*)] \overset{t\to\infty}{\leq} \frac{\eta\mathcal{L}_\tau\sigma L}{2}\frac{\kappa}{2\mu\sqrt{B\delta}L + \sigma\xi\theta(L+\mu)}. \tag{13}$$

*We derive the novel scaling rule by 1) Preserving the upper bound, which requires that $\kappa = \sqrt{\delta}$ and $\xi = \kappa$; 2) Preserving the relative speed of $M_t$, $V_t$ and $X_t$, which requires that $\tilde{\beta}_i = 1 - \kappa(1 - \beta_i)$.*

The left of Figure 5 shows the empirical verification of the predicted loss value and scaling rule on a convex quadratic function: Consistently with Lemma 3.13, such a value is bounded w.r.t. $\sigma$, meaning that the loss of AdamW does not diverge to infinity even if there is an infinite level of gradient noise: See the right of Figure 6 for an experimental validation. Interestingly, the asymptotic loss value is not influenced by the choice of $\beta_i$: We argue that $\beta_i$ do not impact the asymptotic level of the loss, but rather drive the selection of the basin and speed at which AdamW converges to it — The center-right of Fig. 5 exemplifies this on a simple non-convex landscape. Finally, we observe that the scaling rule informally derived in Malladi et al. (2022) prescribes $\tilde{\beta}_i = 1 - \kappa^2(1 - \beta_i)$: This can be recovered by preserving other quantities. While there is no reason to prefer one over the other a priori, our analysis on LLMs in Appendix F.8 shows that our rescaling might be preferable in practice.

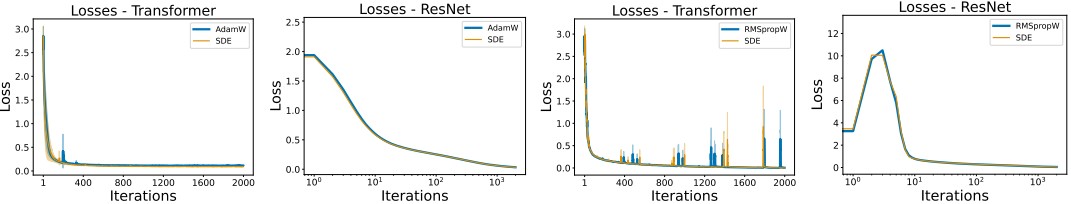

Figure 4: The two images on the left represent the comparison between AdamW and its SDE in terms of $f(x)$. The two on the right do the same for RMSpropW. In both cases, the first is a Transformer on MNIST and the second a ResNet on CIFAR-10: Our SDEs match the respective optimizers.

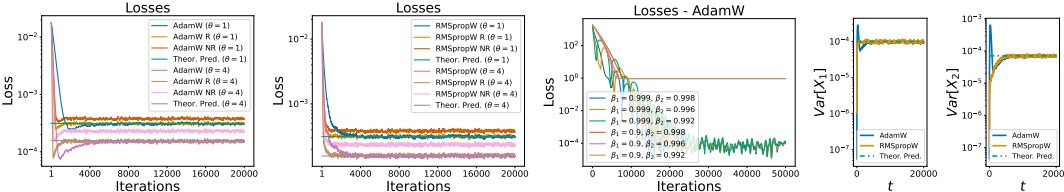

Figure 5: The loss predicted in Lemma 3.13 matches the experimental results on a convex quadratic function. *AdamW* is run with regularization parameter $\theta = 1$. *AdamW R* (AdamW Rescaled) is run as we apply the scaling rule with $\kappa = 2$. *AdamW NR* (AdamW **Not** Rescaled) is run as we apply the scaling rule with $\kappa = 2$ on all hyperparameters but $\theta$, which is left unchanged: Our scaling rule holds, and failing to rescale $\theta$ leads the optimizer not to preserve the asymptotic loss level. The same happens for $\theta = 4$ (Left); The same for RMSpropW (Center-Left); For AdamW, $\beta_1$ and $\beta_2$ influence which basin will attract the dynamics and how fast this will converge, but not the asymptotic loss level inside the basin (Center-Right). For both AdamW and RMSpropW, the variance at convergence predicted in Lemma 3.14 matches the experimental results (Right).

Interestingly, the fact that the weight decay is *decoupled* is key to determining the dependency of the asymptotic loss of AdamW w.r.t. the noise level $\sigma$. While the asymptotic loss of AdamW is upper-bounded in $\sigma$, the same does not hold if we use Adam on the $L^2$-regularized loss $f(x) + \frac{\theta\|x\|_2^2}{2}$. Under the same assumptions of Lemma 3.13, the dynamics of Adam on $f(x) + \frac{\theta\|x\|_2^2}{2}$ implies that

$$\mathbb{E}[f(X_t) - f(X_*)] \overset{t\to\infty}{\leq} \frac{\eta\mathcal{L}_\tau\sigma}{2}\frac{L}{2\mu L + \theta(L+\mu)}, \tag{14}$$

meaning that the asymptotic loss level grows linearly in $\sigma$: See Figure 14 for empirical validation.

We conclude this section with the stationary distribution of AdamW around a minimum which we empirically validate on the right of Figure 5.

**Lemma 3.14.** *If* $\Sigma(x) = \Sigma$, *the stationary distribution of AdamW is*

$$(\mathbb{E}[X_\infty], Cov[X_\infty]) = \left(0, \frac{\eta}{2}\left(I_d + \theta H^{-1}\Sigma^{\frac{1}{2}}\right)^{-1} H^{-1}\Sigma^{\frac{1}{2}}\right).$$

**RMSpropW** We derived the analogous results for RMSprop(W) and we reported them in Appendix C.7: importantly, we validate the SDE in Figure 3 for two simple landscapes and in Figure 4 for a Transformer and a ResNet. The results regarding the asymptotic loss level and stationary distributions are validated in the center-left and right of Figure 5 for a convex quadratic function.

**Conclusion:** While for both SignSGD and Adam the asymptotic loss value and the covariance of the iterates scale linearly with $\Sigma^{\frac{1}{2}}$, we observe for AdamW this is more intricate: The interaction between curvature, noise, and regularization implies that these two quantities are upper-bounded in $\Sigma^{\frac{1}{2}}$ and increasing $\Sigma$ to infinity does not lead to their explosion: *decoupled* weight decay plays a crucial stabilization role at high noise levels near the minimizer — See Figure 6 for a comparison across optimizers. Finally, we argue that $\beta_i$ play a key role in selecting the basin and the convergence speed to the asymptotic loss value rather than impacting the loss value itself.

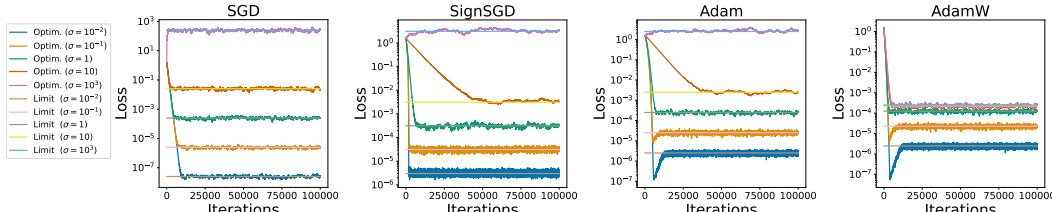

Figure 6: For SGD (Left), SignSGD (Center-Left), Adam (Center-Right), and AdamW: For each *optimizer*, we plot the loss value on a convex quadratic and compare its asymptotic value with the *limits* predicted by our theory. As we take $\Sigma = \sigma^2 I_d$, we confirm that the loss of SGD scales quadratically in $\sigma$ (Lemma 3.6), and linearly for SignSGD (Lemma 3.5) and Adam (Lemma 3.13 with $\theta = 0$). For AdamW, the maximum asymptotic loss value is bounded in $\sigma$ (Lemma 3.13 with $\theta > 0$). In accordance with the experiments, our theory predicts that adaptive methods are more resilient to noise.

## 4 EXPERIMENTS: SDE VALIDATION

The point of our experiments is to validate the theoretical results derived from the SDEs. Therefore, we first show that our SDEs faithfully represent the dynamics of their respective optimizers. To do so, we integrate the SDEs with Euler-Maruyama (Algorithm 1): This is particularly challenging and expensive as one needs to calculate the full gradients of the DNNs at each iteration.[6] Ours is the first set of validation experiments on various architectures and datasets: An MLP on the Breast Cancer dataset, a CNN and a Transformer on MNIST, and a ResNet on CIFAR-10. Details in Appendix F.

## 5 CONCLUSION

We derived the first formal SDE for SignSGD, enabling us to demonstrate its dynamics traversing three discernible phases. We characterize how the "signal-to-noise" ratio drives the dynamics of the loss in each of these phases, and we derive the asymptotic value of the loss function, as well as the stationary distribution. Regarding the role of noise, we draw a straightforward comparison with SGD. For SignSGD, the noise level $\sqrt{\Sigma}$ has an inverse linear effect on the convergence speed of the loss and the iterates. However, it linearly affects the asymptotic expected loss and the asymptotic variance of the iterates. In contrast, for SGD, noise does not influence the convergence speed but has a quadratic impact on the loss level and variance. We also examine the scenario where the noise has infinite variance and demonstrate the resilience of SignSGD, showing that its performance is only marginally affected. Finally, we generalize the analysis to include AdamW and RMSpropW. Specifically, we leverage our novel SDEs to derive the asymptotic value of the loss function, their stationary distribution on a convex quadratic, and a novel scaling rule. The key insight is that, similarly to SignSGD, the loss level and covariance matrix of the iterates of Adam and RMSprop scale linearly in the noise level $\Sigma^{\frac{1}{2}}$. For AdamW and RMSpropW, the complex interaction of noise, curvature, and regularization implies that these two quantities are bounded in terms of $\Sigma^{\frac{1}{2}}$, showing that *decoupled* weight decay plays a crucial stabilization role at high noise levels near the minimizer. Interestingly, the SDEs for Adam and RMSprop are a straightforward corollary of our general results and were derived under much less restrictive and more realistic assumptions than those in the literature. Finally, we thoroughly validate all our theoretical results: We compare the dynamics of the various optimizers with the respective SDEs and find good agreement on simple landscapes and DNNs. For Adam and RMSprop, our SDEs track them more faithfully than those derived in (Malladi et al., 2022).

**Future work** We believe our results can be extended to other optimizers commonly used in practice such as Signum, AdaGrad, AdaMax, and Nadam. Additionally, inspired by the insights from our SDE analysis, there is potential for designing new optimization algorithms that combine and preserve the strengths of existing methods while mitigating their weaknesses. For example, developing hybrid optimizers that adaptively switch between different strategies based on the training phase or current state of the optimization process could offer superior performance.

---

[6]Many papers derive SDEs to model optimizers, but most lack validation. Some use toy landscapes, while only Paquette et al. (2021); Compagnoni et al. (2023) validate on simple DNNs. See Appendix A for details.

## 6 ACKNOWLEDGMENTS

Enea Monzio Compagnoni, Rustem Islamov, and Aurelien Lucchi acknowledge the financial support of the Swiss National Foundation, SNF grant No 207392. Tianlin Liu acknowledges the financial support of the European Research Council Starting Grant 852821—SWING. Antonio Orvieto acknowledges the financial support of the Hector Foundation. Frank Norbert Proske acknowledges the financial support of the Norwegian Research Council (project No 274410) and MSCA4Ukraine (project No 101101923).

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

## A  ADDITIONAL RELATED WORKS

In this section, we list some papers that derived or used SDEs to model optimizers. In particular, we focus on the aspect of empirically verifying the validity of such SDEs in the sense that they indeed track the respective optimizers. We divide these into three categories: Those that did not carry out any type of validation, those that did it on simple landscapes (quadratic functions et similia), and those that did small experiments on neural networks.

While the following works contribute to the theoretical understanding, they do not provide direct experimental validation of their SDE approximations or the derived insights: (Liu et al., 2021; Hu et al., 2019; Bercher et al., 2020; Zhu and Ying, 2021; Cui et al., 2020; Maulén Soto, 2021; Wang and Wu, 2020; Lanconelli and Lauria, 2022; Ayadi and Turinici, 2021; Soto et al., 2022; Li and Wang, 2022; Wang and Mao, 2022; Bardi and Kouhkouh, 2022; Chen et al., 2022; Kunin et al., 2023; Zhang et al., 2023; Sun et al., 2023; Li et al., 2023b; Gess et al., 2024; Dambrine et al., 2024; Maulen-Soto et al., 2024).

The following ones carried out validation experiments on artificial landscapes, e.g. quadratic or quartic function, or easy regression tasks: (Li et al., 2017; 2019; Zhou et al., 2020b; An et al., 2020; Fontaine et al., 2021; Gu et al., 2021; Su and Lau, 2023; Ankirchner and Perko, 2024).

Some recent papers have performed experiments involving neural networks (see, e.g., (Paquette et al., 2021; Compagnoni et al., 2023)). In both works, the authors simulate the corresponding SDEs using numerical integrators and then compare the results with those of the associated optimizers. Specifically, (Paquette et al., 2021) validates the SDE on a shallow MLP, while (Compagnoni et al., 2023) performs the validation on both a shallow and a deep MLP.

In contrast, the studies by (Li et al., 2021; Malladi et al., 2022) do not empirically validate their derived SDEs through numerical integration. While their analysis of SVAG provides valuable insights, we believe there are conceptual aspects worth discussing:

1. After deriving an SDE for an optimizer which we refer to as "*Optimizer A*", one observes that simulating these SDEs can be computationally expensive;

2. To mitigate this computational challenge, (Li et al., 2021; Malladi et al., 2022) introduce a discrete-time algorithm called SVAG, which shares the same SDE as "*Optimizer A*". Importantly, SVAG does not perform a numerical integration of the original SDE, as it does not require access to either the drift or diffusion term;

3. (Li et al., 2021; Malladi et al., 2022) simulate SVAG and observe that it tracks "*Optimizer A*" closely, which they interpret as evidence that the SDE is a suitable approximation for "*Optimizer A*".

However, it is important to note that (Li et al., 2021; Malladi et al., 2022) do not directly validate the SDE by simulating it. To illustrate a potential concern with this approach, consider the following hypothetical scenario:

1. Derive an SDE for "*Optimizer A*";

2. Recognize that simulating the SDE is computationally costly;

3. Define another discrete-time algorithm called "*Optimizer B*", which coincides with "*Optimizer A*" and therefore, by definition, shares the same SDE;

4. Simulate "*Optimizer B*" and observe that it tracks "*Optimizer A*" perfectly, since they are identical by construction;

5. Conclude that the SDE is a good approximation for "*Optimizer A*".

While the reasoning in prior work (Li et al., 2021; Malladi et al., 2022) demonstrates that SVAG is a discrete-time optimizer that shares the same SDE as "*Optimizer A*", this alone does not establish that the SDE is the correct or most suitable model for "*Optimizer A*". Simply comparing two algorithms that share an SDE does not necessarily confirm the validity of the SDE itself. Otherwise, an optimizer compared with itself would trivially satisfy this criterion. A more direct validation of the SDE would require numerical integration using a method that explicitly incorporates the drift and diffusion terms (Higham, 2001; Milstein, 2013).

## B  STOCHASTIC CALCULUS

In this section, we summarize some important results in the analysis of Stochastic Differential Equations Mao (2007); Øksendal (1990). The notation and the results in this section will be used extensively in all proofs in this paper. We assume the reader to have some familiarity with Brownian motion and with the definition of stochastic integral (Ch. 1.4 and 1.5 in Mao (2007)).

### B.1  ITÔ'S LEMMA

We start with some notation: Let $(\Omega, \mathcal{F}, \{\mathcal{F}_t\}_{t\geq 0}, \mathbb{P})$ be a filtered probability space. We say that an event $E \in \mathcal{F}$ holds almost surely (a.s.) in this space if $\mathbb{P}(E) = 1$. We call $\mathcal{L}^p([a,b], \mathbb{R}^d)$, with $p > 0$, the family of $\mathbb{R}^d$-valued $\mathcal{F}_t$-adapted processes $\{f_t\}_{a\leq t\leq b}$ such that

$$\int_a^b \|f_t\|^p dt \leq \infty.$$

Moreover, we denote by $\mathcal{M}^p([a,b], \mathbb{R}^d)$, with $p > 0$, the family of $\mathbb{R}^d$-valued processes $\{f_t\}_{a\leq t\leq b}$ in $\mathcal{L}([a,b], \mathbb{R}^d)$ such that $\mathbb{E}\left[\int_a^b \|f_t\|^p dt\right] \leq \infty$. We will write $h \in \mathcal{L}^p(\mathbb{R}_+, \mathbb{R}^d)$, with $p > 0$, if $h \in \mathcal{L}^p([0,T], \mathbb{R}^d)$ for every $T > 0$. Similar definitions hold for matrix-valued functions using the Frobenius norm $\|A\| := \sqrt{\sum_{ij} |A_{ij}|^2}$.

Let $W = \{W_t\}_{t\geq 0}$ be a one-dimensional Brownian motion defined on our probability space and let $X = \{X_t\}_{t\geq 0}$ be an $\mathcal{F}_t$-adapted process taking values on $\mathbb{R}^d$.

**Definition B.1.** Let the *drift* be $b \in \mathcal{L}^1(\mathbb{R}_+, \mathbb{R}^d)$ and the diffusion term be $\sigma \in \mathcal{L}^2(\mathbb{R}_+, \mathbb{R}^{d\times m})$. $X_t$ is an Itô process if it takes the form

$$X_t = x_0 + \int_0^t b_s ds + \int_0^t \sigma_s dW_s.$$

We shall say that $X_t$ has the stochastic differential

$$dX_t = b_t dt + \sigma_t dW_t. \tag{15}$$

**Theorem B.2** (Itô's Lemma). *Let $X_t$ be an Itô process with stochastic differential $dX_t = b_t dt + \sigma_t dW_t$. Let $f(x,t)$ be twice continuously differentiable in $x$ and continuously differentiable in $t$, taking values in $\mathbb{R}$. Then $f(X_t, t)$ is again an Itô process with stochastic differential*

$$df(X_t, t) = \partial_t f(X_t, t)) dt + \langle \nabla f(X_t, t), b_t \rangle dt + \frac{1}{2} Tr\left(\sigma_t \sigma_t^\top \nabla^2 f(X_t, t)\right) dt + \langle \nabla f(X_t, t), \sigma_t \rangle dW_t. \tag{16}$$

### B.2  STOCHASTIC DIFFERENTIAL EQUATIONS

Stochastic Differential Equations (SDEs) are equations of the form

$$dX_t = b(X_t, t) dt + \sigma(X_t, t) dW_t.$$

First of all, we need to define what it means for a stochastic process $X = \{X_t\}_{t\geq 0}$ with values in $\mathbb{R}^d$ to solve an SDE.

**Definition B.3.** Let $X_t$ be as above with deterministic initial condition $X_0 = x_0$. Assume $b : \mathbb{R}^d \times [0, T] \to \mathbb{R}^d$ and $\sigma : \mathbb{R}^d \times [0, T] \to \mathbb{R}^{d \times m}$ are Borel measurable; $X_t$ is called a solution to the corresponding SDE if

1. $X_t$ is continuous and $\mathcal{F}_t$-adapted;

2. $b \in \mathcal{L}^1\left([0, T], \mathbb{R}^d\right)$;

3. $\sigma \in \mathcal{L}^2\left([0, T], \mathbb{R}^{d \times m}\right)$;

4. For every $t \in [0, T]$

$$X_t = x_0 + \int_0^t b(X_s, s)ds + \int_0^t \sigma(X_s, s)dW(s) \quad a.s.$$

Moreover, the solution $X_t$ is said to be unique if any other solution $X_t^\star$ is such that

$$\mathbb{P}\left\{X_t = X_t^\star, \text{ for all } 0 \le t \le T\right\} = 1.$$

Notice that since the solution to an SDE is an Itô process, we can use Itô's lemma. The following theorem gives a sufficient condition on $b$ and $\sigma$ for the existence of a solution to the corresponding SDE.

**Theorem B.4.** *Assume that there exist two positive constants $\bar{K}$ and $K$ such that*

1. *(Global Lipschitz condition) for all $x, y \in \mathbb{R}^d$ and $t \in [0, T]$*

$$\max\{\|b(x, t) - b(y, t)\|^2, \ \|\sigma(x, t) - \sigma(y, t)\|^2\} \le \bar{K}\|x - y\|^2;$$

2. *(Linear growth condition) for all $x \in \mathbb{R}^d$ and $t \in [0, T]$*

$$\max\{\|b(x, t)\|^2, \ \|\sigma(x, t)\|^2\} \le K(1 + \|x\|^2).$$

*Then, there exists a unique solution $X_t$ to the corresponding SDE, and $X_t \in \mathcal{M}^2([0, T], \mathbb{R}^d)$.*

**Numerical approximation.** Often, SDEs are solved numerically. The simplest algorithm to provide a sample path $(\hat{x}_k)_{k \ge 0}$ for $X_t$, so that $X_{k\Delta t} \cong \hat{x}_k$ for some small $\Delta t$ and for all $k\Delta t \le M$ is called Euler-Maruyama (Algorithm 1). For more details on this integration method and its approximation properties, the reader can check Mao (2007).

---

**Algorithm 1** Euler-Maruyama Integration Method for SDEs

---

**input** The drift $b$, the volatility $\sigma$, and the initial condition $x_0$.
    Fix a stepsize $\Delta t$;
    Initialize $\hat{x}_0 = x_0$;
    $k = 0$;
    **while** $k \le \left\lfloor \frac{T}{\Delta t} \right\rfloor$ **do**
        Sample some $d$-dimensional Gaussian noise $Z_k \sim \mathcal{N}(0, I_d)$;
        Compute $\hat{x}_{k+1} = \hat{x}_k + \Delta t\, b(\hat{x}_k, k\Delta t) + \sqrt{\Delta t}\, \sigma(\hat{x}_k, k\Delta t)Z_k$;
        $k = k + 1$;
    **end while**
**output** The approximated sample path $(\hat{x}_k)_{0 \le k \le \left\lfloor \frac{T}{\Delta t} \right\rfloor}$.

---

## C   THEORETICAL FRAMEWORK - WEAK APPROXIMATION

In this section, we introduce the theoretical framework used in the paper, together with its assumptions and notations.

First of all, many proofs will use Taylor expansions in powers of $\eta$. For ease of notation, we introduce the shorthand that whenever we write $\mathcal{O}\left(\eta^\alpha\right)$, we mean that there exists a function $K(x) \in G$ such that the error terms are bounded by $K(x)\eta^\alpha$. For example, we write

$$b(x + \eta) = b_0(x) + \eta b_1(x) + \mathcal{O}\left(\eta^2\right)$$

to mean: there exists $K \in G$ such that

$$|b(x + \eta) - b_0(x) - \eta b_1(x)| \le K(x)\eta^2.$$

Additionally, we introduce the following shorthand:

- A multi-index is $\alpha = (\alpha_1, \alpha_2, \ldots, \alpha_n)$ such that $\alpha_j \in \{0, 1, 2, \ldots\}$;
- $|\alpha| := \alpha_1 + \alpha_2 + \cdots + \alpha_n$;
- $\alpha! := \alpha_1! \alpha_2! \cdots \alpha_n!$;
- For $x = (x_1, x_2, \ldots, x_n) \in \mathbb{R}^n$, we define $x^\alpha := x_1^{\alpha_1} x_2^{\alpha_2} \cdots x_n^{\alpha_n}$;
- For a multi-index $\beta$, $\partial_\beta^{|\beta|} f(x) := \frac{\partial^{|\beta|}}{\partial_{x_1}^{\beta_1} \partial_{x_2}^{\beta_2} \cdots \partial_{x_n}^{\beta_n}} f(x)$;
- We also denote the partial derivative with respect to $x_i$ by $\partial_{e_i}$.

**Definition C.1** (G Set). Let $G$ denote the set of continuous functions $\mathbb{R}^d \to \mathbb{R}$ of at most polynomial growth, i.e. $g \in G$ if there exists positive integers $\nu_1, \nu_2 > 0$ such that $|g(x)| \le \nu_1 \left(1 + |x|^{2\nu_2}\right)$, for all $x \in \mathbb{R}^d$.

**Definition C.2** ($\mathcal{C}_b^k\left(\mathbb{R}^n, \mathbb{R}\right)$). $\mathcal{C}_b^k\left(\mathbb{R}^n, \mathbb{R}\right)$ denotes the space of functions whose $k$-th derivatives are bounded.

The next results are inspired by Theorem 1 of Li et al. (2017) and are derived under some regularity assumption on the function $f$.

## C.1 ASSUMPTIONS

In general, we assume some regularity in the loss function.

**Assumption C.3.** Assume that the following conditions on $f, f_i \in \mathcal{C}_b^8\left(\mathbb{R}^n, \mathbb{R}\right)$, and their gradients are satisfied:

- $\nabla f, \nabla f_i$ satisfy a Lipschitz condition: there exists $L > 0$ such that

$$|\nabla f(u) - \nabla f(v)| + \sum_{i=1}^n |\nabla f_i(u) - \nabla f_i(v)| \le L|u - v|;$$

- $f, f_i$ and its partial derivatives up to order 7 belong to $G$;
- $\nabla f, \nabla f_i$ satisfy a growth condition: there exists $M > 0$ such that

$$|\nabla f(x)| + \sum_{i=1}^n |\nabla f_i(x)| \le M(1 + |x|).$$

Regarding the gradient noise, each optimizer has its mild assumptions which are weaker or in line with the literature.

**SignSGD**

1. The gradient noise $Z(x)$ admits a strictly positive density function $g_x$ for all $x$ and require that $g : \mathbb{R}^n \times \mathbb{R}^n \to [0, \infty)$ s.t. $(x, y) \mapsto g_x(y)$ is in $C^8(\mathbb{R}^n \times \mathbb{R}^n)$ such that all partial derivatives of $g$ up to order 8 are integrable with respect to $y$ and s.t. their $L^1$-norms are uniformly bounded in $x$. This assumption covers Gaussian and Student's t, thus being *more general than the literature*. Indeed, the Gaussianity of the noise is commonly assumed:

Among others, see Ahn et al. (2012); Chen et al. (2014); Mandt et al. (2016); Stephan et al. (2017); Zhu et al. (2019); Wu et al. (2020); Xie et al. (2021), while Jastrzebski et al. (2018) offers an intuitive justification as well;

2. For all compact sets $K$

$$\sup_{x \in K} |g(x, \cdot)| \in L^1(\mathbb{R}^n),$$

which of course covers the Gaussian case, *thus being more general than the literature*.

3. The functions in Eq. 18 to be in $G$, which, as we show below, covers Gaussian and Student's t, *thus being more general than the literature*.

**Adam(W) and RMSprop(W)**

1. In line with Malladi et al. (2022), we assume that $\sqrt{\Sigma}(x)$ is: In $G$ together with its derivatives, Lipschitz, bounded, and satisfy Affine Growth;

2. The term $(\nabla f(x))^2$ to be Lipschitz and of affine growth, which is a consequence of assuming bounded gradients as often done in the literature on the convergence of RMSprop and Adam: Among many, see (Luo et al., 2019; Défossez et al., 2022; Guo et al., 2021; Huang et al., 2021) together with the discussion in Section 2.1 of Shi and Li (2021).

**Remark** All the assumptions above are *in line with or more general than those commonly found in the literature*. In line with *Remark 11* of the seminal paper Li et al. (2019), we observe that while some of these assumptions might seem strong, loss functions in applications have inward pointing gradients for sufficiently large $x$. Therefore, we could simply modify the loss to satisfy the assumptions above.

Regarding the drift and diffusion coefficients, we highlight that many papers in the literature following this framework do not check for their regularity before applying the approximation theorems Hu et al. (2019); An et al. (2020); Zhu and Ying (2021); Cui et al. (2020); Maulén Soto (2021); Wang and Mao (2022); Compagnoni et al. (2023; 2024); Li et al. (2017). At first sight, it would seem that not even the seminal paper Li et al. (2019) checks these conditions carefully. However, a deeper investigation shows that they are restricting their analysis to compact sets to leverage the regularity and convergence properties of mollifiers: The assumption regarding the compactness of the domain is not highlighted nor assumed in any part of the paper. Therefore, we conclude that, willingly or not, most papers implicitly make these assumptions.

## C.2 TECHNICAL RESULTS

In this subsection, we provide some results that will be instrumental in the derivation of the SDEs.

**Lemma C.4.** *Assume the existence of a probability density $g_x$ of the gradient noise $Z(x)$ for all $x$ and require that $g : \mathbb{R}^n \times \mathbb{R}^n \to [0, \infty)$; $(x, y) \mapsto g_x(y)$ is in $C^8(\mathbb{R}^n \times \mathbb{R}^n)$ such that all partial derivatives of $g$ up to order 8 are integrable with respect to $y$ and such that their $L^1-$norms are uniformly bounded in $x$. Further, let $f \in C^8(\mathbb{R}^n)$ and $h : \mathbb{R}^n \to \mathbb{R}$ be a bounded Borel measurable function. Define the function $k$ by*

$$k(x) = \mathbb{E}\left[h(\nabla f_\gamma(x))\right].$$

*Then there exists a version $\widehat{k}$ of $k$ with $\widehat{k} \in C_b^7(\mathbb{R}^n)$.*

*Proof.* Let $\varphi$ be smooth and compactly supported. Then for all multi indices $\beta$ with $|\beta| \leq 8$, substitution, Fubini's theorem, and integration by parts imply that

$$
\begin{aligned}
\int_{\mathbb{R}^n} k(x) \partial_\beta^{|\beta|} \varphi(x) dx &= \int_{\mathbb{R}^n} \mathbb{E}\left[h(\nabla f_\gamma(x)) \partial_\beta^{|\beta|} \varphi(x) dx\right. \\
&= \int_{\mathbb{R}^n} \int_{\mathbb{R}^n} h(y) g_x(y - \nabla f(x)) dy \partial_\beta^{|\beta|} \varphi(x) dx \\
&= (-1)^{|\beta|} \int_{\mathbb{R}^n} \int_{\mathbb{R}^n} h(y) \partial_\beta^{|\beta|} (g_x(y - \nabla f(x))) dy \varphi(x) dx.
\end{aligned}
$$

So

$$\int_{\mathbb{R}^n} h(y) \partial_\beta^{|\beta|}(g_x(y - \nabla f(x))) dy$$

is a weak derivative $\partial_\beta^{|\beta|} k$ of $k$ on any bounded open set. For compact sets $K$ we obtain that

$$\int_K \left| \int_{\mathbb{R}^n} h(y) \partial_\beta^{|\beta|}(g_x(y - \nabla f(x))) dy \right|^p dx$$

$$\leq \quad \|h\|_\infty^p \lambda^n(K) \left( \sup_{x \in \mathbb{R}^n} \int_{\mathbb{R}^n} \left| \partial_\beta^{|\beta|}(g_x(y - \nabla f(x))) \right| dy \right)^p < \infty$$

for all $p \geq 2$ because of our assumptions on $g$ and $f$ and substitution ($\lambda^n$ Lebesgue measure). So it follows from Sobolev embeddings with respect to Hölder spaces that for all bounded and open sets $\Omega$ there exists a version $\widehat{k}$ of $k$ such that $\widehat{k} \in C^7(\Omega)$. The latter version can be extended to $\Omega = \mathbb{R}^n$, which we also denote by $\widehat{k}$. Since $\partial_\beta^{|\beta|} k$ is bounded for $|\beta| \leq 8$, we conclude that $\widehat{k} \in C_b^7(\mathbb{R}^n)$. $\quad\square$

**Lemma C.5.** *Assuming that for all compact sets $K$*

$$\sup_{x \in K} |g(x, \cdot)| \in L^1(\mathbb{R}^n),$$

*and the positivity of the density functions, we have that for $m = 1, \ldots, 7$ that*

$$\left\| \partial_{j_1} \ldots \partial_{j_m} A^{1/2}(x) \right\| \leq C l_m(x), \tag{17}$$

*where the function $l_m(x)$ is defined as*

$$l_m(x) \quad := \quad \sum_{r=0}^{m-1} \left( \frac{1}{m(x) + s(x)(n-1)^{1/2}} \left( 1 + \frac{2s(x)(n-1)^{1/2}}{m(x) - s(x)(n-1)^{-1/2}} \right) \right)^{-(r+1/2)}$$

$$\times \max_{|\beta| \leq m} \left\| \partial_\beta^{|\beta|} A(x) \right\|^{r+1}. \tag{18}$$

*Proof.* To prove this, we need the fact that the Fréchet derivatives of the square root function $\varphi$ can be represented as follows (see Theorem 1.1 in Del Moral and Niclas (2018)):

$$\nabla \varphi(A)[H] = \int_0^\infty e^{-t\varphi(A)} H e^{-t\varphi(A)} dt,$$

and higher derivatives of order $m \geq 2$ are given by

$$\nabla^m \varphi(A)[H, \ldots, H] \quad = \quad -\nabla \varphi(A) \left[ \sum_{p+q=m-2} \frac{m!}{(p+1)!(q+1)!} (\nabla^{p+1} \varphi(A)[H, \ldots, H]) \right.$$

$$\left. \times (\nabla^{q+1} \varphi(A)[H, \ldots, H]) \right] \tag{19}$$

for all $A \in \mathbb{S}$ and symmetric $n \times n$ matrices $H$. Moreover, we have the following estimate for $m \geq 0$:

$$\left\| \nabla^{m+1} \varphi(A) \right\| \leq (\sqrt{n})^m (m+1)! C_m 2^{-2(m+1)} \lambda_{\min}(A)^{-(m+1/2)}, \tag{20}$$

where $\lambda_{\min}(A) > 0$ is the smallest eigenvalue of $A$ and $C_m := \frac{1}{m+1} \binom{2m}{m}$.

We find that $\partial_l A^{1/2}(x) = \nabla \varphi(A(x))[\partial_l A(x)]$ and

$$\partial_j \partial_l A^{1/2}(x) = \nabla^2 \varphi(A(x))[\partial_j A(x), \partial_l A(x)] + \nabla \varphi(A(x))[\partial_j \partial_l A(x)].$$

Thus, it follows from Eq. (20) that

$$\left\| \partial_l A^{1/2}(x) \right\| \leq C \lambda_{\min}(A(x))^{-1/2} \left\| \partial_l A(x) \right\|,$$

and

$$\left\| \partial_j \partial_l A^{1/2}(x) \right\| \quad \leq \quad C_1 \lambda_{\min}(A(x))^{-(1+1/2)} \left\| \partial_j A(x) \right\| \left\| \partial_l A(x) \right\|$$

$$+ C_2 \lambda_{\min}(A(x))^{-1/2} \left\| \partial_j \partial_l A(x) \right\|.$$

More generally, for $m = 1, \ldots, 7$,

$$\left\| \partial_{j_1} \ldots \partial_{j_m} A^{1/2}(x) \right\| \leq C_m \left\{ \sum_{r=0}^{m-1} \lambda_{\min}(A(x))^{-(r+1/2)} \right.$$
$$\left. \times \max_{|\beta| \leq m} \left\| \partial_\beta^{|\beta|} A(x) \right\|^{r+1} \right\}. \qquad (21)$$

Let us now provide a lower bound for $\lambda_{\min}(A(x))$ in terms of $tr(A(x))$ and $tr((A(x))^2)$. Define

$$s^2(x) = n^{-1} \left( tr((A(x))^2) - \frac{(tr(A(x)))^2}{n} \right), \quad m(x) = \frac{tr(A(x))}{n}.$$

Then, from Corollary 2.1, Corollary 2.2, and Theorem 2.1 in Wolkowicz and Styan (1980), we obtain

$$\frac{1}{\lambda_{\min}(A(x))} \leq \frac{1}{\lambda_{\max}(A(x))} \left( 1 + \frac{2s(x)(n-1)^{1/2}}{m(x) - s(x)(n-1)^{-1/2}} \right)$$
$$\leq \frac{1}{m(x) + s(x)(n-1)^{1/2}} \left( 1 + \frac{2s(x)(n-1)^{1/2}}{m(x) - s(x)(n-1)^{-1/2}} \right).$$

Therefore, from Eq. (21), we have for $m = 1, \ldots, 7$ that

$$\left\| \partial_{j_1} \ldots \partial_{j_m} A^{1/2}(x) \right\| \leq C l_m(x), \qquad (22)$$

where the function $l_m(x)$ is defined as

$$l_m(x) := \sum_{r=0}^{m-1} \left( \frac{1}{m(x) + s(x)(n-1)^{1/2}} \left( 1 + \frac{2s(x)(n-1)^{1/2}}{m(x) - s(x)(n-1)^{-1/2}} \right) \right)^{-(r+1/2)}$$
$$\times \max_{|\beta| \leq m} \left\| \partial_\beta^{|\beta|} A(x) \right\|^{r+1}. \qquad (23)$$

$\square$

The following results are key to guarantee that an SDE is a weak approximation of an optimizer.

**Lemma C.6** (Lemma 1 Li et al. (2017)). *Let $0 < \eta < 1$. Consider a stochastic process $X_t, t \geq 0$ satisfying the SDE*

$$dX_t = b(X_t) dt + \sqrt{\eta} \sigma(X_t) dW_t$$

*with $X_0 = x \in \mathbb{R}^d$ and $b, \sigma$ together with their derivatives belong to $G$. Define the one-step difference $\Delta = X_\eta - x$, and indicate the $i$-th component of $\Delta$ with $\Delta_i$. Then we have*

1. $\mathbb{E}\Delta_i = b_i \eta + \frac{1}{2} \left[ \sum_{j=1}^d b_j \partial_{e_j} b_i \right] \eta^2 + \mathcal{O}(\eta^3) \quad \forall i = 1, \ldots, d;$
2. $\mathbb{E}\Delta_i \Delta_j = \left[ b_i b_j + \sigma \sigma_{(ij)}^T \right] \eta^2 + \mathcal{O}(\eta^3) \quad \forall i, j = 1, \ldots, d;$
3. $\mathbb{E} \prod_{j=1}^s \Delta_{(i_j)} = \mathcal{O}(\eta^3)$ *for all $s \geq 3, i_j = 1, \ldots, d$.*

*All functions above are evaluated at $x$.*

**Theorem C.7** (Theorem 2 and Lemma 5, Mil'shtein (1986)). *Let Assumption C.3 hold and let us define $\bar{\Delta} = x_1 - x$ to be the increment in the discrete-time algorithm, and indicate the $i$-th component of $\bar{\Delta}$ with $\bar{\Delta}_i$. If in addition there exists $K_1, K_2, K_3, K_4 \in G$ so that*

1. $\left| \mathbb{E}\Delta_i - \mathbb{E}\bar{\Delta}_i \right| \leq K_1(x)\eta^2, \quad \forall i = 1, \ldots, d;$

2. $\left| \mathbb{E}\Delta_i\Delta_j - \mathbb{E}\bar{\Delta}_i\bar{\Delta}_j \right| \leq K_2(x)\eta^2, \quad \forall i, j = 1, \ldots, d;$

3. $\left| \mathbb{E}\prod_{j=1}^{s}\Delta_{i_j} - \mathbb{E}\prod_{j=1}^{s}\bar{\Delta}_{i_j} \right| \leq K_3(x)\eta^2, \quad \forall s \geq 3, \quad \forall i_j \in \{1, \ldots, d\};$

4. $\mathbb{E}\prod_{j=1}^{3}\left| \bar{\Delta}_{i_j} \right| \leq K_4(x)\eta^2, \quad \forall i_j \in \{1, \ldots, d\}.$

*Then, there exists a constant $C$ so that for all $k = 0, 1, \ldots, N$ we have*

$$\left| \mathbb{E}g\left(X_{k\eta}\right) - \mathbb{E}g\left(x_k\right) \right| \leq C\eta.$$

## C.3 LIMITATIONS

Modeling of discrete-time algorithms using SDEs relies on Assumption C.3. As noted by Li et al. (2021), the approximation can fail when the stepsize $\eta$ is large or if certain conditions on $\nabla f$ and the noise covariance matrix are not met. Although these issues can be addressed by increasing the order of the weak approximation, we believe that the primary purpose of SDEs is to serve as simplification tools that enhance our intuition: We would not benefit significantly from added complexity.

## C.4 FORMAL DERIVATION - SIGNSGD

In this subsection, we provide the first formal derivation of an SDE model for SignSGD. Let us consider the stochastic process $X_t \in \mathbb{R}^d$ defined as the solution of

$$dX_t = -(1 - 2\mathbb{P}(\nabla f_\gamma(X_t) < 0))dt + \sqrt{\eta}\sqrt{\bar{\Sigma}(X_t)}dW_t, \tag{24}$$

where

$$\bar{\Sigma}(x) = \mathbb{E}[\xi_\gamma(x)\xi_\gamma(x)^\top], \tag{25}$$

and $\xi_\gamma(x) := \text{sign}(\nabla f_\gamma(x)) - 1 + 2\mathbb{P}(\nabla f_\gamma(x) < 0)$ the noise in the sample $\text{sign}(\nabla f_\gamma(x))$. The following theorem guarantees that such a process is a 1-order SDE of the discrete-time algorithm of SignSGD

$$x_{k+1} = x_k - \eta\,\text{sign}\left(f_{\gamma_k}(x_k)\right), \tag{26}$$

with $x_0 \in \mathbb{R}^d$, $\eta \in \mathbb{R}^{>0}$ is the step size, the mini-batches $\{\gamma_k\}$ are modelled as i.i.d. random variables uniformly distributed on $\{1, \cdots, N\}$, and of size $B \geq 1$.

Before proceeding, we ensure that the SDE admits a unique solution and that its coefficients are sufficiently regular.

**Lemma C.8.** *The drift term $b(x) := -(1 - 2\mathbb{P}(\nabla f_\gamma(x) < 0))$ is Lipschitz, satisfies affine growth, and belongs to the space $G$ together with its derivatives.*

*Proof.* Since we are assuming that the gradient noise has a smooth and bounded probability density function,[7] the drift can be rewritten in terms of the CDF $F_Z(x)$ of the noise as $b(x) := 2F_Z(-\nabla f(x)) - 1$, whose derivative is $-2F_Z'(-\nabla f(x))\nabla^2 f(x)$. Since the density function and the Hessian of $f$ are bounded, we conclude that the derivative is bounded. Therefore, the drift is Lipschitz and as regular as $\nabla f$, meaning that each entry is in $G$, together with its derivatives. Finally, since it is bounded, it has affine growth. $\quad\square$

**Lemma C.9.** *The diffusion coefficient $\sqrt{\bar{\Sigma}}$ satisfies the affine growth condition.*

*Proof.* Since it is bounded, the result follows immediately. $\quad\square$

---

[7]This is commonly assumed in the literature. Among others, Ahn et al. (2012); Chen et al. (2014); Mandt et al. (2016); Stephan et al. (2017); Zhu et al. (2019); Wu et al. (2020); Xie et al. (2021) assume that it is Gaussian, while Jastrzebski et al. (2018) offers an intuitive justification.

**Lemma C.10.** *Let us assume the same assumptions as Lemma C.4. Additionally, assume that*

$$\sup_{x \in K} |g(x, \cdot)| \in L^1(\mathbb{R}^n)$$

*for all compact sets $K$. Then the entries of $\overline{\Sigma}$ in Eq. 25 are in $C_b^7(\mathbb{R}^n)$.*

*Proof.* By the definition of $\overline{\Sigma}$ in terms of the sign-function and dominated convergence, from the additional assumption on $g$, it follows that $\overline{\Sigma}$ is continuous. So Lemma C.4 entails that the entries of $\overline{\Sigma}$ are in $C_b^7(\mathbb{R}^n)$. □

**Lemma C.11.** *Under the assumption that*

$$g(x, y) > 0, \tag{27}$$

*the covariance matrix $\overline{\Sigma}$ is positive definite.*

*Proof.* For $y = (y_1, \ldots, y_n)^T$, observe that

$$\left(\overline{\Sigma}(x)y, y\right) = \sum_{i,j=1}^n y_i \mathbb{E}\left[\xi_\gamma^i(x)\xi_\gamma^j(x)\right] y_j = \mathbb{E}\left[\left(\sum_{i=1}^n \xi_\gamma^i(x)y_i\right)^2\right].$$

Using the definition of $\xi_\gamma$ and the positivity of the density $g$, we can argue by contradiction and see that for $y \neq 0$, the right-hand side of the equation must be strictly greater than zero for all $x$. Therefore, $\overline{\Sigma}(x) \in \mathbb{S}$ for all $x$, where $\mathbb{S}$ denotes the open set of positive definite matrices in the space of symmetric $n \times n$ matrices. □

**Corollary C.12.** *Since $\overline{\Sigma}$ is positive definite and its entries are in $C_b^7(\mathbb{R}^n)$, $\sqrt{\overline{\Sigma}}$ is Lipschitz.*

*Proof.* The function

$$\varphi : \mathbb{S} \to \mathbb{S}, \quad A \mapsto \sqrt{A}$$

has Fréchet derivatives of any order on $\mathbb{S}$ (see e.g. Del Moral and Niclas (2018)). Therefore, $\overline{\Sigma}^{1/2} \in C^7(\mathbb{R}^n)$, and since $\overline{\Sigma} \in C_b^7(\mathbb{R}^n)$, $\overline{\Sigma}^{1/2}$ is Lipschitz continuous (see Proposition 6.2 in Ikeda and Watanabe (2014)). □

**Proposition C.13.** *Assume the conditions of Lemma C.5 and assume that the functions $l_m(x)$ for $m = 1, \ldots, 7$ in Eq. (18) are of polynomial growth. Then $\overline{\Sigma}^{1/2} \in G$ together with its derivatives.*

**Corollary C.14.** *If the noise $Z(x) \sim \mathcal{N}(0, \Sigma)$ or $Z(x) \sim t_\nu(0, \Sigma)$, then $\overline{\Sigma}^{1/2} \in G$ together with its derivatives.*

*Proof.* With the definition of $\Xi(x)$ given in Lemma 3.10, the function $K(x) := \sqrt{1 - 4\Xi(x)^2}$ is in $G$ together with its derivative: It is easy to verify that all the derivatives of $K(x)$ are bounded even in the case $\nu = 1$, which is the most pathological one. Therefore, $\sqrt{\overline{\Sigma}}(x)$ is in $G$ together with its derivatives. □

*Remark* C.15. Based on the above results, we have that under mild assumptions on the noise structures (see Sec. C.1) that cover and generalize the well-accepted Gaussianity, e.g. covering Student's t as well, the SDE of SignSGD admits a unique solution and its coefficients are regular enough to apply Lemma C.6 and Thm. C.7.

---

**Theorem C.16** (Stochastic modified equations). *Let $0 < \eta < 1, T > 0$ and set $N = \lfloor T/\eta \rfloor$. Let $x_k \in \mathbb{R}^d, 0 \leq k \leq N$ denote a sequence of SignSGD iterations defined by Eq. 26. Consider the stochastic process $X_t$ defined in Eq. 24 and fix some test function $g \in G$ and suppose that $g$ and its partial derivatives up to order 6 belong to $G$.*
*Then, under Assumption C.3, there exists a constant $C > 0$ independent of $\eta$ such that for all $k = 0, 1, \ldots, N$, we have*

$$|\mathbb{E}g(X_{k\eta}) - \mathbb{E}g(x_k)| \leq C\eta.$$

*That is, the SDE 24 is an order 1 weak approximation of the SignSGD iterations 26.*

**Lemma C.17.** *Under the assumptions of Theorem C.16, let $0 < \eta < 1$ and consider $x_k, k \geq 0$ satisfying the SignSGD iterations*

$$x_{k+1} = x_k - \eta \operatorname{sign}\left(\nabla f_{\gamma_k}(x_k)\right)$$

*with $x_0 \in \mathbb{R}^d$. From the definition the one-step difference $\bar{\Delta} = x_1 - x$, then we have*

*1. $\mathbb{E}\bar{\Delta}_i = -\left(1 - 2\mathbb{P}\left(\partial_i f_\gamma < 0\right)\right)\eta \quad \forall i = 1, \ldots, d$;*
*2. $\mathbb{E}\bar{\Delta}_i \bar{\Delta}_j = \left(\left(1 - 2\mathbb{P}\left(\partial_i f_\gamma < 0\right)\right)\left(1 - 2\mathbb{P}\left(\partial_j f_\gamma < 0\right)\right) + \bar{\Sigma}_{(ij)}\right)\eta^2 \quad \forall i, j = 1, \ldots, d$;*
*3. $\mathbb{E}\prod_{j=1}^{s} \bar{\Delta}_{i_j} = \mathcal{O}\left(\eta^3\right) \quad \forall s \geq 3, \quad i_j \in \{1, \ldots, d\}$.*

*All the functions above are evaluated at $x$.*

*Proof of Lemma C.17.* First of all, we have that by definition

$$\mathbb{E}\left[x_1^i - x^i\right] = -\eta \mathbb{E}\left[\operatorname{sign}\left(\partial_i f_\gamma(x)\right)\right], \tag{28}$$

which implies

$$\mathbb{E}\bar{\Delta}_i = -\left(1 - 2\mathbb{P}\left(\partial_i f_\gamma(x) < 0\right)\right)\eta \quad \forall i = 1, \ldots, d. \tag{29}$$

Second, we have that by definition

$$\mathbb{E}\left[(x_1 - x)(x_1 - x)^\top\right] = \mathbb{E}\left[(x_1 - x)\right]\mathbb{E}\left[(x_1 - x)^\top\right] + \tag{30}$$

$$\mathbb{E}\Big[\left(\operatorname{sign}\left(\nabla f_\gamma(x)\right) - 1 + 2\mathbb{P}\left(\nabla f_\gamma(x) < 0\right)\right) \tag{31}$$

$$\left(\operatorname{sign}\left(\nabla f_\gamma(x)\right) - 1 + 2\mathbb{P}\left(\nabla f_\gamma(x) < 0\right)\right)^\top\Big]\eta^2, \tag{32}$$

which implies that

$$\mathbb{E}\bar{\Delta}_i \bar{\Delta}_j = \left(1 - 2\mathbb{P}\left(\partial_i f_\gamma < 0\right)\right)\left(1 - 2\mathbb{P}\left(\partial_j f_\gamma < 0\right)\right)\eta^2 + \bar{\Sigma}_{(ij)}\eta^2 \quad \forall i, j = 1, \ldots, d. \tag{33}$$

Finally, by definition

$$\mathbb{E}\prod_{j=1}^{s} \bar{\Delta}_{i_j} = \mathcal{O}\left(\eta^3\right) \quad \forall s \geq 3, \quad i_j \in \{1, \ldots, d\}, \tag{34}$$

which concludes our proof. $\square$

*Proof of Theorem C.16.* To prove this result, all we need to do is check the conditions in Theorem C.7. As we apply Lemma C.6, we make the following choices:

- $b(x) = -(1 - 2\mathbb{P}\left(\nabla f_\gamma(x) < 0\right))$;

- $\sigma(x) = \sqrt{\bar{\Sigma}(x)}.$

First of all, we notice that $\forall i = 1, \ldots, d$, it holds that

- $\mathbb{E}\bar{\Delta}_i \overset{\text{1. Lemma C.17}}{=} -\left(1 - 2\mathbb{P}\left(\partial_i f_\gamma(x) < 0\right)\right)\eta$;

- $\mathbb{E}\Delta_i \overset{\text{1. Lemma C.6}}{=} -\left(1 - 2\mathbb{P}\left(\partial_i f_\gamma(x) < 0\right)\right)\eta + \mathcal{O}\left(\eta^2\right).$

Therefore, we have that for some $K_1(x) \in G$,

$$\left| \mathbb{E}\Delta_i - \mathbb{E}\bar{\Delta}_i \right| \le K_1(x)\eta^2, \quad \forall i = 1, \dots, d. \tag{35}$$

Additionally, we notice that $\forall i, j = 1, \dots, d$, it holds that

- $\mathbb{E}\bar{\Delta}_i\bar{\Delta}_j \overset{\text{2. Lemma C.17}}{=} \left(1 - 2\mathbb{P}\left(\partial_i f_\gamma(x) < 0\right)\right)\left(1 - 2\mathbb{P}\left(\partial_j f_\gamma(x) < 0\right)\right)\eta^2 + \bar{\Sigma}_{(ij)}(x)\eta^2;$

- $\mathbb{E}\Delta_i\Delta_j \overset{\text{2. Lemma C.6}}{=} \left(\left(1 - 2\mathbb{P}\left(\partial_i f_\gamma(x) < 0\right)\right)\left(1 - 2\mathbb{P}\left(\partial_j f_\gamma(x) < 0\right)\right) + \bar{\Sigma}_{(ij)}(x)\right)\eta^2 + \mathcal{O}\left(\eta^3\right).$

Therefore, we have that for some $K_2(x) \in G$,

$$\left| \mathbb{E}\Delta_i\Delta_j - \mathbb{E}\bar{\Delta}_i\bar{\Delta}_j \right| \le K_2(x)\eta^2, \quad \forall i, j = 1, \dots, d. \tag{36}$$

Additionally, we notice that $\forall s \ge 3, \forall i_j \in \{1, \dots, d\}$, it holds that

- $\mathbb{E}\prod_{j=1}^{s}\bar{\Delta}_{i_j} \overset{\text{3. Lemma C.17}}{=} \mathcal{O}\left(\eta^3\right);$

- $\mathbb{E}\prod_{j=1}^{s}\Delta_{i_j} \overset{\text{3. Lemma C.6}}{=} \mathcal{O}\left(\eta^3\right).$

Therefore, we have that for some $K_3(x) \in G$,

$$\left| \mathbb{E}\prod_{j=1}^{s}\Delta_{i_j} - \mathbb{E}\prod_{j=1}^{s}\bar{\Delta}_{i_j} \right| \le K_3(x)\eta^2. \tag{37}$$

Additionally, for some $K_4(x) \in G, \forall i_j \in \{1, \dots, d\}$,

$$\mathbb{E}\prod_{j=1}^{3}\left| \bar{\Delta}_{(i_j)} \right| \overset{\text{3. Lemma C.17}}{\le} K_4(x)\eta^2. \tag{38}$$

*Remark* C.18. Remembering Remark C.15, and thanks to Eq. 35, Eq. 36, Eq. 37, and Eq. 38, the thesis follows from Lemma C.6 and Thm. C.7.

$\square$

In all the following results, the reader will notice that all the drifts, diffusion terms, and noise assumptions are selected to guarantee that the SDE we derived for SignSGD is indeed a 1 weak approximation for SignSGD even without the mollification argument used in Li et al. (2019) to handle the regularity issues.

**Corollary C.19.** *Let us take the same assumptions of Theorem C.16, and that the stochastic gradient is $\nabla f_\gamma(x) = \nabla f(x) + Z$ such that $Z \sim \mathcal{N}(0, \Sigma)$ that does not depend on $x$. Then, the following SDE provides a 1 weak approximation of the discrete update of SignSGD*

$$dX_t = -Erf\left(\frac{\Sigma^{-\frac{1}{2}}\nabla f(X_t)}{\sqrt{2}}\right)dt + \sqrt{\eta}\sqrt{I_d - \text{diag}\left(Erf\left(\frac{\Sigma^{-\frac{1}{2}}\nabla f(X_t)}{\sqrt{2}}\right)\right)^2}dW_t, \tag{39}$$

*where the error function $Erf(x)$ and the square are applied component-wise, and $\Sigma = \text{diag}\left(\sigma_1^2, \cdots, \sigma_d^2\right)$.*

*Proof of Corollary C.19.* First of all, we observe that

$$1 - 2\mathbb{P}\left(\nabla f_\gamma(x) < 0\right) = 1 - 2\mathbb{P}\left(\nabla f(x) + \Sigma^{\frac{1}{2}}Z < 0\right) = 1 - 2\Phi\left(-\Sigma^{-\frac{1}{2}}\nabla f(x)\right), \tag{40}$$

where $\Phi$ is the cumulative distribution function of the standardized normal distribution. Remembering that

$$\Phi(x) = \frac{1}{2}\left(1 + \mathrm{Erf}\left(\frac{x}{\sqrt{2}}\right)\right),\tag{41}$$

we have that

$$1 - 2\mathbb{P}\left(\nabla f_\gamma(x) < 0\right) = 1 - 2\frac{1}{2}\left(1 + \mathrm{Erf}\left(-\frac{\Sigma^{-\frac{1}{2}}\nabla f(x)}{\sqrt{2}}\right)\right) = \mathrm{Erf}\left(\frac{\Sigma^{-\frac{1}{2}}\nabla f(x)}{\sqrt{2}}\right).\tag{42}$$

Similarly, one can prove that $\bar{\Sigma}$ defined in 25 becomes

$$\bar{\Sigma} = I_d - \mathrm{diag}\left(\mathrm{Erf}\left(\frac{\Sigma^{-\frac{1}{2}}\nabla f(X_t)}{\sqrt{2}}\right)\right)^2.\tag{43}$$

$\square$

**Corollary C.20.** *Let us take the same assumptions of Theorem C.16, and that the stochastic gradient is $\nabla f_\gamma(x) = \nabla f(x) + \sqrt{\Sigma}Z$ such that $Z \sim t_\nu(0, I_d)$ that does not depend on $x$ and $\nu$ is a positive integer number. Then, the following SDE provides a 1 weak approximation of the discrete update of SignSGD*

$$dX_t = -2\Xi\left(\Sigma^{-\frac{1}{2}}\nabla f(X_t)\right)dt + \sqrt{\eta}\sqrt{I_d - 4\,\mathrm{diag}\left(\Xi\left(\Sigma^{-\frac{1}{2}}\nabla f(X_t)\right)\right)^2}dW_t,\tag{44}$$

*where $\Xi(x)$ is defined as*

$$\Xi(x) := x\frac{\Gamma\left(\frac{\nu+1}{2}\right)}{\sqrt{\pi\nu}\Gamma\left(\frac{\nu}{2}\right)}{}_2F_1\left(\frac{1}{2}, \frac{\nu+1}{2}; \frac{3}{2}; -\frac{x^2}{\nu}\right),\tag{45}$$

*and ${}_2F_1(a, b; c; x)$ is the hypergeometric function. Above, function $\Xi(x)$ and the square are applied component-wise, and $\Sigma = \mathrm{diag}\left(\sigma_1^2, \cdots, \sigma_d^2\right)$.*

*Proof.* First of all, we observe that

$$1 - 2\mathbb{P}\left(\nabla f_\gamma(x) < 0\right) = 1 - 2\mathbb{P}\left(\nabla f(x) + \Sigma^{\frac{1}{2}}Z < 0\right) = 1 - 2F_\nu\left(-\Sigma^{-\frac{1}{2}}\nabla f(x)\right),\tag{46}$$

where $F_\nu(x)$ is the cumulative function of a $t$ distribution with $\nu$ degrees of freedom. Remembering that

$$F_\nu(x) = \frac{1}{2} + \Xi_\nu(x),\tag{47}$$

we have that

$$1 - 2\mathbb{P}\left(\nabla f_\gamma(x) < 0\right) = 1 - 2\left(\frac{1}{2} + \Xi_\nu(-\Sigma^{-\frac{1}{2}}\nabla f(x))\right) = 2\Xi_\nu(\Sigma^{-\frac{1}{2}}\nabla f(x)).\tag{48}$$

Similarly, one can prove that $\bar{\Sigma}$ becomes

$$\bar{\Sigma} = I_d - 4\,\mathrm{diag}\left(\Xi_\nu\left(\Sigma^{-\frac{1}{2}}\nabla f(X_t)\right)\right)^2.\tag{49}$$

$\square$

**Lemma C.21.** *Under the assumptions of Corollary C.19 and signal-to-noise ratio $Y_t := \frac{\Sigma^{-\frac{1}{2}}\nabla f(X_t)}{\sqrt{2}}$,*

1. ***Phase 1:** If $|Y_t| > \frac{3}{2}$, the SDE coincides with the ODE of SignGD:*

$$dX_t = -\mathrm{sign}(\nabla f(X_t))dt;\tag{50}$$

2. ***Phase 2:** If $1 < |Y_t| < \frac{3}{2}$:*

(a) $mY_t + \mathbf{q}^- \le \frac{d\mathbb{E}[X_t]}{dt} \le mY_t + \mathbf{q}^+;$

(b) $\mathbb{P}\left[\|X_t - \mathbb{E}[X_t]\|_2^2 > a\right] \le \frac{\eta}{a}\left(d - \|mY_t + \mathbf{q}^-\|_2^2\right);$

3. **Phase 3:** *If* $|Y_t| < 1$*, the SDE is*

$$dX_t = -\sqrt{\frac{2}{\pi}}\Sigma^{-\frac{1}{2}}\nabla f(X_t)dt + \sqrt{\eta}\sqrt{I_d - \frac{2}{\pi}\,\text{diag}\left(\Sigma^{-\frac{1}{2}}\nabla f(X_t)\right)^2}dW_t. \quad (51)$$

*Proof of Lemma C.21.* Exploiting the regularity of the Erf function, we approximate the SDE in Eq. 39 in three different regions:

1. **Phase 1:** If $|x| > \frac{3}{2}$, $\text{Erf}(x) \sim \text{sign}(x)$. Therefore, if $\left|\frac{\Sigma^{-\frac{1}{2}}\nabla f(X_t)}{\sqrt{2}}\right| > \frac{3}{2}$,

   (a) $\text{Erf}\left(\frac{\Sigma^{-\frac{1}{2}}\nabla f(X_t)}{\sqrt{2}}\right) \sim \text{sign}\left(\frac{\Sigma^{-\frac{1}{2}}\nabla f(X_t)}{\sqrt{2}}\right) = \text{sign}\left(\nabla f(X_t)\right);$

   (b) $\text{Erf}\left(\frac{\Sigma^{-\frac{1}{2}}\nabla f(X_t)}{\sqrt{2}}\right)^2 \sim \text{sign}\left(\frac{\Sigma^{-\frac{1}{2}}\nabla f(X_t)}{\sqrt{2}}\right)^2 = (1,\dots,1).$

   Therefore,

$$dX_t = -\text{Erf}\left(\frac{\Sigma^{-\frac{1}{2}}\nabla f(X_t)}{\sqrt{2}}\right)dt + \sqrt{\eta}\sqrt{I_d - \text{diag}\left(\text{Erf}\left(\frac{\Sigma^{-\frac{1}{2}}\nabla f(X_t)}{\sqrt{2}}\right)\right)^2}dW_t$$
$$\sim -\text{sign}(\nabla f(X_t)); \quad (52)$$

2. **Phase 2:** Let $m$ and $q_1$ are the slope and intercept of the line secant to the graph of $\text{Erf}(x)$ between the points $(1, \text{Erf}(1))$ and $\left(\frac{3}{2}, \text{Erf}\left(\frac{3}{2}\right)\right)$, while $q_2$ is the intercept of the line tangent to the graph of $\text{Erf}(x)$ and slope $m$. If $1 < x < \frac{3}{2}$, we have that

$$mx + q_1 < \text{Erf}(x) < mx + q_2. \quad (53)$$

   Analogously, if $-\frac{3}{2} < x < -1$

$$mx - q_2 < \text{Erf}(x) < mx - q_1. \quad (54)$$

   Therefore, we have that if $1 < \left|\frac{\Sigma^{-\frac{1}{2}}\nabla f(X_t)}{\sqrt{2}}\right| < \frac{3}{2}$, then

   (a)
$$\frac{m}{\sqrt{2}}\Sigma^{-\frac{1}{2}}\nabla f(X_t) + \mathbf{q}^- < \text{Erf}\left(\frac{\Sigma^{-\frac{1}{2}}\nabla f(X_t)}{\sqrt{2}}\right) < \frac{m}{\sqrt{2}}\Sigma^{-\frac{1}{2}}\nabla f(X_t) + \mathbf{q}^+, \quad (55)$$

   where
$$(\mathbf{q}^+)_i := \begin{cases} q_2 & \text{if } \partial_i f(x) > 0 \\ -q_1 & \text{if } \partial_i f(x) < 0, \end{cases} \quad (56)$$

   and
$$(\mathbf{q}^-)_i := \begin{cases} q_1 & \text{if } \partial_i f(x) > 0 \\ -q_2 & \text{if } \partial_i f(x) < 0, \end{cases} \quad (57)$$

   Therefore,
$$-\frac{m}{\sqrt{2}}\Sigma^{-\frac{1}{2}}\nabla f(X_t) - \mathbf{q}^+ \le \frac{d\mathbb{E}[X_t]}{dt} \le -\frac{m}{\sqrt{2}}\Sigma^{-\frac{1}{2}}\nabla f(X_t) - \mathbf{q}^-; \quad (58)$$

   (b) Similar to the above,

$$\left(\frac{m}{\sqrt{2}}\Sigma^{-\frac{1}{2}}\nabla f(X_t) + \mathbf{q}^-\right)^2 \le \text{Erf}\left(\frac{\Sigma^{-\frac{1}{2}}\nabla f(X_t)}{\sqrt{2}}\right)^2 \le \left(\frac{m}{\sqrt{2}}\Sigma^{-\frac{1}{2}}\nabla f(X_t) + \mathbf{q}^+\right)^2.$$

Therefore,

$$\mathbb{P}\left[\|X_t - \mathbb{E}\left[X_t\right]\|_2^2 > a\right] \le \mathbb{P}\left[\exists i \text{ s.t. } |X_t^i - \mathbb{E}\left[X_t^i\right]|^2 > a\right] \tag{59}$$

$$\le \sum_i \mathbb{P}\left[|X_t^i - \mathbb{E}\left[X_t^i\right]| > \sqrt{a}\right]$$

$$\le \frac{\eta}{a} \sum_i \left(1 - \text{Erf}\left(\frac{\Sigma_i^{-\frac{1}{2}}\partial_i f(X_t)}{\sqrt{2}}\right)^2\right) \tag{60}$$

$$< \frac{\eta}{a}\left(d - \|\frac{m}{\sqrt{2}}\Sigma^{-\frac{1}{2}}\nabla f(X_t) + \mathbf{q}^-\|_2^2\right). \tag{61}$$

3. **Phase 3:** If $|x| < 1$, $\text{Erf}(x) \sim \frac{2}{\sqrt{\pi}}x$. Therefore, if $\left|\frac{\Sigma^{-\frac{1}{2}}\nabla f(X_t)}{\sqrt{2}}\right| < 1$,

  (a) $\text{Erf}\left(\frac{\Sigma^{-\frac{1}{2}}\nabla f(X_t)}{\sqrt{2}}\right) \sim \sqrt{\frac{2}{\pi}}\Sigma^{-\frac{1}{2}}\nabla f(X_t)$;

  (b) $\left(\text{Erf}\left(\frac{\Sigma^{-\frac{1}{2}}\nabla f(X_t)}{\sqrt{2}}\right)\right)^2 \sim \frac{2}{\pi}\left(\Sigma^{-\frac{1}{2}}\nabla f(X_t)\right)^2$.

Therefore,

$$dX_t = -\text{Erf}\left(\frac{\Sigma^{-\frac{1}{2}}\nabla f(X_t)}{\sqrt{2}}\right)dt + \sqrt{\eta}\sqrt{I_d - \text{diag}\left(\text{Erf}\left(\frac{\Sigma^{-\frac{1}{2}}\nabla f(X_t)}{\sqrt{2}}\right)\right)^2}dW_t$$

$$\sim -\sqrt{\frac{2}{\pi}}\Sigma^{-\frac{1}{2}}\nabla f(X_t)dt + \sqrt{\eta}\sqrt{I_d - \frac{2}{\pi}\text{diag}\left(\Sigma^{-\frac{1}{2}}\nabla f(X_t)\right)^2}dW_t. \tag{62}$$

$\square$

**Lemma C.22** (Dynamics of Expected Loss). *Let $f$ be $\mu$-strongly convex, $Tr(\nabla^2 f(x)) \le \mathcal{L}_\tau$, and $S_t := f(X_t) - f(X_*)$. Then, during*

  1. *Phase 1, the dynamics will stop before $t_* = 2\sqrt{\frac{S_0}{\mu}}$ because $S_t \le \frac{1}{4}\left(\sqrt{\mu}t - 2\sqrt{S_0}\right)^2$;*

  2. *Phase 2 with $\Delta := \left(\frac{m}{\sqrt{2}\sigma_{max}} + \frac{\eta\mu m^2}{4\sigma_{max}^2}\right)$: $\mathbb{E}[S_t] \le S_0 e^{-2\mu\Delta t} + \frac{\eta}{2}\frac{\left(\mathcal{L}_\tau - \mu d\hat{q}^2\right)}{2\mu\Delta}\left(1 - e^{-2\mu\Delta t}\right)$;*

  3. *Phase 3 with $\Delta := \left(\sqrt{\frac{2}{\pi}}\frac{1}{\sigma_{max}} + \frac{\eta}{\pi}\frac{\mu}{\sigma_{max}^2}\right)$: $\mathbb{E}[S_t] \le S_0 e^{-2\mu\Delta t} + \frac{\eta}{2}\frac{\mathcal{L}_\tau}{2\mu\Delta}\left(1 - e^{-2\mu\Delta t}\right)$.*

*Proof of Lemma C.22.* We prove each point by leveraging the shape of the law of $X_t$ derived in Lemma C.21:

  1. **Phase 1:**
  $$d(f(X_t) - f(X_*)) = -\nabla f(X_t)\,\text{sign}(\nabla f(X_t))dt = -\|\nabla f(X_t)\|_1 dt \le -\|\nabla f(X_t)\|_2 dt \tag{63}$$
  Since $f$ is $\mu - PL$, we have that $-\|\nabla f(X_t)\|_2^2 < -2\mu(f(X_t) - f(X_*))$, which implies that
  $$f(X_t) - f(X_*) \le \frac{1}{4}\left(\sqrt{\mu}t - 2\sqrt{f(X_0) - f(X_*)}\right)^2, \tag{64}$$
  meaning that the dynamics will stop before $t_* = 2\sqrt{\frac{f(X_0) - f(X_*)}{\mu}}$;

  2. **Phase 2:** By applying the Itô Lemma to $f(X_t) - f(X_*)$ and that
  $$\frac{m}{\sqrt{2}}\Sigma^{-\frac{1}{2}}\nabla f(X_t) + \mathbf{q}^- < \text{Erf}\left(\frac{\Sigma^{-\frac{1}{2}}\nabla f(X_t)}{\sqrt{2}}\right) < \frac{m}{\sqrt{2}}\Sigma^{-\frac{1}{2}}\nabla f(X_t) + \mathbf{q}^+, \tag{65}$$

we have that if $\hat{q} := \max(q_1, q_2)$,

$$d(f(X_t) - f(X_*)) \le - \left( \frac{m}{\sqrt{2}} \Sigma^{-\frac{1}{2}} \nabla f(X_t) + \mathbf{q}^- \right)^\top \nabla f(X_t) dt + \mathcal{O}(\text{Noise}) \tag{66}$$

$$+ \frac{\eta}{2} \text{Tr} \left[ \nabla^2 f(X_t) \left( I_d - \text{diag} \left( \frac{m}{\sqrt{2}} \Sigma^{-\frac{1}{2}} \nabla f(X_t) + \mathbf{q}^- \right)^2 \right) \right] dt \tag{67}$$

$$\le - \frac{m}{\sqrt{2}} \frac{1}{\sigma_{\max}} \|\nabla f(X_t)\|_2^2 dt - \hat{q} \|\nabla f(X_t)\|_1 dt + \frac{\eta \mathcal{L}_\tau}{2} dt \tag{68}$$

$$- \frac{\eta \mu}{2} \| \frac{m}{\sqrt{2}} \Sigma^{-\frac{1}{2}} \nabla f(X_t) dt + \mathbf{q}^- \|_2^2 dt + \mathcal{O}(\text{Noise}) \tag{69}$$

$$\le - \frac{m}{\sqrt{2}} \frac{1}{\sigma_{\max}} \|\nabla f(X_t)\|_2^2 dt - \hat{q} \|\nabla f(X_t)\|_1 dt + \frac{\eta \mathcal{L}_\tau}{2} dt \tag{70}$$

$$- \frac{\eta \mu m^2}{4\sigma_{\max}^2} \|\nabla f(X_t)\|_2^2 dt - \frac{\eta \mu d \hat{q}^2}{2} dt - \frac{\sqrt{2} m \hat{q}}{\sigma_{\max}} \|\nabla f(X_t)\|_1 dt \tag{71}$$

$$+ \mathcal{O}(\text{Noise}) \tag{72}$$

$$\le - 2\mu \left( \frac{m}{\sqrt{2}\sigma_{\max}} + \frac{\eta \mu m^2}{4\sigma_{\max}^2} \right) (f(X_t) - f(X_*)) dt \tag{73}$$

$$+ \frac{\eta}{2} \left( \mathcal{L}_\tau - \mu d \hat{q}^2 \right) dt + \mathcal{O}(\text{Noise}), \tag{74}$$

which implies that if $k := 2\mu \left( \frac{m}{\sqrt{2}\sigma_{\max}} + \frac{\eta \mu m^2}{4\sigma_{\max}^2} \right)$,

$$\mathbb{E}[f(X_t) - f(X_*)] \le (f(X_0) - f(X_*))) e^{-kt} + \frac{\eta \left( \mathcal{L}_\tau - \mu d \hat{q}^2 \right)}{2k} \left( 1 - e^{-kt} \right). \tag{75}$$

3. **Phase 3:** By applying the Itô Lemma to $f(X_t) - f(X_*)$, we have that:

$$d(f(X_t) - f(X_*)) = - \sqrt{\frac{2}{\pi}} \nabla f(X_t)^\top \Sigma^{-\frac{1}{2}} \nabla f(X_t) dt + \mathcal{O}(\text{Noise}) \tag{76}$$

$$+ \frac{\eta}{2} \text{Tr} \left( \left( I_d - \frac{2}{\pi} \text{diag} \left( \Sigma^{-\frac{1}{2}} \nabla f(X_t) \right)^2 \right) \nabla^2 f(X_t) \right) dt \tag{77}$$

$$\le - \sqrt{\frac{2}{\pi}} \frac{1}{\sigma_{\max}} \|\nabla f(X_t)\|_2^2 dt + \mathcal{O}(\text{Noise}) \tag{78}$$

$$+ \frac{\eta}{2} \text{Tr} \left( \nabla^2 f(X_t) \right) dt - \frac{\eta}{\pi} \frac{\mu}{\sigma_{\max}^2} \|\nabla f(X_t)\|_2^2 dt \tag{79}$$

$$\le - \left( \sqrt{\frac{2}{\pi}} \frac{1}{\sigma_{\max}} + \frac{\eta}{\pi} \frac{\mu}{\sigma_{\max}^2} \right) \|\nabla f(X_t)\|_2^2 dt \tag{80}$$

$$+ \frac{\eta}{2} Tr(\nabla^2 f(X_t)) dt + \mathcal{O}(\text{Noise}) \tag{81}$$

Since $f$ is $\mu$-Strongly Convex, $f$ is also $\mu$-PL. Therefore, we have

$$d(f(X_t) - f(X_*)) \le - 2\mu \left( \sqrt{\frac{2}{\pi}} \frac{1}{\sigma_{\max}} + \frac{\eta}{\pi} \frac{\mu}{\sigma_{\max}^2} \right) (f(X_t) - f(X_*)) dt \tag{82}$$

$$+ \frac{\eta}{2} Tr(\nabla^2 f(X_t)) dt + \mathcal{O}(\text{Noise}). \tag{83}$$

Therefore,

$$d\mathbb{E}[f(X_t) - f(X_*)] \le - 2\mu \left( \sqrt{\frac{2}{\pi}} \frac{1}{\sigma_{\max}} + \frac{\eta}{\pi} \frac{\mu}{\sigma_{\max}^2} \right) (\mathbb{E}[f(X_t) - f(X_*)]) dt + \frac{\eta}{2} \mathcal{L}_\tau dt, \tag{84}$$

which implies that if $k := 2\mu \left( \sqrt{\frac{2}{\pi}} \frac{1}{\sigma_{\max}} + \frac{\eta}{\pi} \frac{\mu}{\sigma_{\max}^2} \right)$,

$$\mathbb{E}[f(X_t) - f(X_*)] \leq (f(X_0) - f(X_*)))e^{-kt} + \frac{\eta \mathcal{L}_\tau}{2k} \left( 1 - e^{-kt} \right). \tag{85}$$

$\square$

We can weaken the regularity of $f$ from $\mu$-strongly convex to $\mu$-PL: This results in less tight bounds as expected.

**Lemma C.23** (Dynamics of Expected Loss). *Let $f$ be $\mu$-PL, $L$-smooth, and $S_t := f(X_t) - f(X_*)$. Then, during*

1. *Phase 1, the dynamics will stop before $t_* = 2\sqrt{\frac{S_0}{\mu}}$ because $S_t \leq \frac{1}{4} \left( \sqrt{\mu}t - 2\sqrt{S_0} \right)^2$;*

2. *Phase 2 with $\Delta := \frac{m}{\sqrt{2}\sigma_{max}}$: $\mathbb{E}[S_t] \leq S_0 e^{-2\mu\Delta t} + \frac{\eta Ld}{4\mu\Delta} \left( 1 - e^{-2\mu\Delta t} \right)$;*

3. *Phase 3 with $\Delta := \sqrt{\frac{2}{\pi}} \frac{1}{\sigma_{max}}$: $\mathbb{E}[S_t] \leq S_0 e^{-2\mu\Delta t} + \frac{\eta Ld}{4\mu\Delta} \left( 1 - e^{-2\mu\Delta t} \right)$.*

*Proof of Lemma C.22.* We prove each point by leveraging the shape of the law of $X_t$ derived in Lemma C.21:

1. **Phase 1:**

   $$d(f(X_t) - f(X_*)) = -\nabla f(X_t) \operatorname{sign}(\nabla f(X_t))dt = -\|\nabla f(X_t)\|_1 dt \leq -\|\nabla f(X_t)\|_2 dt. \tag{86}$$

   Since $f$ is $\mu - PL$, we have that $-\|\nabla f(X_t)\|_2^2 < -2\mu(f(X_t) - f(X_*))$, which implies that

   $$f(X_t) - f(X_*) \leq \frac{1}{4} \left( \sqrt{\mu}t - 2\sqrt{f(X_0) - f(X_*)} \right)^2, \tag{87}$$

   meaning that the dynamics will stop before $t_* = 2\sqrt{\frac{f(X_0) - f(X_*)}{\mu}}$;

2. **Phase 2:** By applying the Itô Lemma to $f(X_t) - f(X_*)$ and that

   $$\frac{m}{\sqrt{2}} \Sigma^{-\frac{1}{2}} \nabla f(X_t) + \mathbf{q}^- < \operatorname{Erf}\left( \frac{\Sigma^{-\frac{1}{2}} \nabla f(X_t)}{\sqrt{2}} \right) < \frac{m}{\sqrt{2}} \Sigma^{-\frac{1}{2}} \nabla f(X_t) + \mathbf{q}^+, \tag{88}$$

   we have that if $\hat{q} := \max(q_1, q_2)$,

   $$d(f(X_t) - f(X_*)) \leq - \left( \frac{m}{\sqrt{2}} \Sigma^{-\frac{1}{2}} \nabla f(X_t) + \mathbf{q}^- \right)^\top \nabla f(X_t)dt + \mathcal{O}(\text{Noise}) \tag{89}$$

   $$+ \frac{\eta}{2} \operatorname{Tr}\left[ \nabla^2 f(X_t) \left( I_d - \operatorname{diag}\left( \frac{m}{\sqrt{2}} \Sigma^{-\frac{1}{2}} \nabla f(X_t) + \mathbf{q}^- \right)^2 \right) \right] dt \tag{90}$$

   $$\leq - \frac{m}{\sqrt{2}} \frac{1}{\sigma_{\max}} \|\nabla f(X_t)\|_2^2 dt - \hat{q}\|\nabla f(X_t)\|_1 dt + \frac{\eta Ld}{2} dt \tag{91}$$

   $$\leq - 2\mu \frac{m}{\sqrt{2}\sigma_{\max}} (f(X_t) - f(X_*))dt + \frac{\eta Ld}{2} dt + \mathcal{O}(\text{Noise}), \tag{92}$$

   which implies that if $\Delta := \frac{m}{\sqrt{2}\sigma_{\max}}$,

   $$\mathbb{E}[f(X_t) - f(X_*)] \leq (f(X_0) - f(X_*)))e^{-2\mu\Delta t} + \frac{\eta Ld}{4\mu\Delta} \left( 1 - e^{-2\mu\Delta t} \right). \tag{93}$$

3. **Phase 3:** By applying the Itô Lemma to $f(X_t) - f(X_*)$, we have that:

$$d(f(X_t) - f(X_*)) = -\sqrt{\frac{2}{\pi}}\nabla f(X_t)^\top \Sigma^{-\frac{1}{2}}\nabla f(X_t)dt + \mathcal{O}(\text{Noise}) + \frac{\eta L d}{2}dt \quad (94)$$

$$\leq -\sqrt{\frac{2}{\pi}}\frac{1}{\sigma_{\max}}\|\nabla f(X_t)\|_2^2 dt + \mathcal{O}(\text{Noise}) + \frac{\eta L d}{2}dt \quad (95)$$

Since $f$ is $\mu$-PL, we have

$$d(f(X_t) - f(X_*)) \leq -2\mu\sqrt{\frac{2}{\pi}}\frac{1}{\sigma_{\max}}(f(X_t) - f(X_*))dt + \frac{\eta L d}{2}dt + \mathcal{O}(\text{Noise}). \quad (96)$$

Therefore, for $\Delta := \sqrt{\frac{2}{\pi}}\frac{1}{\sigma_{\max}}$,

$$\mathbb{E}[f(X_t) - f(X_*)] \leq (f(X_0) - f(X_*)))e^{-2\mu\Delta t} + \frac{\eta L d}{4\mu\Delta}\left(1 - e^{-2\mu\Delta t}\right). \quad (97)$$

$\square$

We can weaken the regularity of $f$ from $\mu$-PL to $L$-Smooth: Of course, we can only bound the expected norm of the gradient.

**Lemma C.24** (Dynamics of Expected Gradient Norm). *Let $f$ be L-smooth, $\eta_t$ be a learning rate scheduler such that $\lim_{t\to\infty}\frac{\phi_t^2}{\phi_t^1}\overset{t\to\infty}{\to}0$ and $\phi_t^1\overset{t\to\infty}{\to}\infty$, where $\phi_t^i = \int_0^t(\eta_s)^i ds$. Then, during*

1. *Phase 1, $\|\nabla f(X_{\tilde{t}^1})\|_1 \leq \frac{f(X_0) - f(X_*)}{\phi_t^1}\overset{t\to\infty}{\to}0$;*

2. *Phase 2,*

$$\left(\frac{m}{\sqrt{2}}\mathbb{E}\|\nabla f(X_{\tilde{t}^{(1,2)}})\|_2^2 + \hat{q}\sigma_{max}\mathbb{E}\|\nabla f(X_{\tilde{t}^{(2,2)}})\|_1\right) \leq \sigma_{max}\left(\frac{f(X_0) - f(X_*)}{\phi_t^1} + \frac{\eta L d}{2}\frac{\phi_t^2}{\phi_t^1}\right)\overset{t\to\infty}{\to}0;$$

3. *Phase 3, $\mathbb{E}\|\nabla f(X_{\tilde{t}^3})\|_2^2 \leq \sqrt{\frac{\pi}{2}}\frac{\sigma_{max}\eta L d}{2}\frac{\phi_t^2}{\phi_t^1} + \sqrt{\frac{\pi}{2}}\sigma_{max}\frac{f(X_0)-f(X_*)}{\phi_t^1}\overset{t\to\infty}{\to}0$;*

*where $\tilde{t}^1$, $\tilde{t}^{(1,2)}$, $\tilde{t}^{(2,2)}$, and $\tilde{t}^3$ are random times with distribution $\frac{\eta_t}{\phi_t^1}$.*

*Proof of Lemma C.22.* We prove each point by leveraging the shape of the law of $X_t$ derived in Lemma C.21:

1. **Phase 1:**

$$d(f(X_t) - f(X_*)) = -\eta_t\nabla f(X_t)\,\text{sign}(\nabla f(X_t))dt = -\eta_t\|\nabla f(X_t)\|_1 dt \quad (98)$$

$$= -\phi_t^1\frac{\eta_t\|\nabla f(X_t)\|_1}{\phi_t^1}dt \quad (99)$$

Therefore, by integrating over time and using the law of the unconscious statistician

$$\|\nabla f(X_{\tilde{t}^1})\|_1 \leq \frac{f(X_0) - f(X_*)}{\phi_t^1}\overset{t\to\infty}{\to}0; \quad (100)$$

2. **Phase 2:** By applying the Itô Lemma to $f(X_t) - f(X_*)$ and that

$$\frac{m}{\sqrt{2}}\Sigma^{-\frac{1}{2}}\nabla f(X_t) + \mathbf{q}^- < \text{Erf}\left(\frac{\Sigma^{-\frac{1}{2}}\nabla f(X_t)}{\sqrt{2}}\right) < \frac{m}{\sqrt{2}}\Sigma^{-\frac{1}{2}}\nabla f(X_t) + \mathbf{q}^+, \quad (101)$$

Similar to what we have shown above, we have that

$$d(f(X_t) - f(X_*)) \leq -\frac{m}{\sqrt{2}}\frac{1}{\sigma_{\max}}\eta_t\|\nabla f(X_t)\|_2^2 dt - \eta_t\hat{q}\|\nabla f(X_t)\|_1 dt \quad (102)$$

$$+\eta_t^2\frac{\eta L d}{2}dt + \mathcal{O}(\text{Noise}). \quad (103)$$

Therefore, by integrating over time and using the law of the unconscious statistician we have

$$\frac{m}{\sqrt{2}}\mathbb{E}\|\nabla f\left(X_{\tilde{t}^{(1,2)}}\right)\|_2^2 + \hat{q}\sigma_{\max}\mathbb{E}\|\nabla f\left(X_{\tilde{t}^{(2,2)}}\right)\|_1 \leq \frac{\sigma_{\max}}{\phi_t^1}\left(f(X_0) - f(X_*) + \frac{\eta L d\phi_t^2}{2}\right) \overset{t\to\infty}{\to} 0; \tag{104}$$

3. **Phase 3:** By applying the Itô Lemma to $f(X_t) - f(X_*)$, we have that:

$$d(f(X_t) - f(X_*)) \leq -\sqrt{\frac{2}{\pi}}\frac{1}{\sigma_{\max}}\eta_t\|\nabla f(X_t)\|_2^2 dt + \mathcal{O}(\text{Noise}) + \eta_t^2\frac{\eta L d}{2}dt \tag{105}$$

Therefore, by integrating over time and using the law of the unconscious statistician we have

$$\mathbb{E}\|\nabla f\left(X_{\tilde{t}^3}\right)\|_2^2 \leq \sqrt{\frac{\pi}{2}}\frac{\sigma_{\max}\eta L d}{2}\frac{\phi_t^2}{\phi_t^1} + \sqrt{\frac{\pi}{2}}\sigma_{\max}\frac{f(X_0) - f(X_*)}{\phi_t^1} \overset{t\to\infty}{\to} 0. \tag{106}$$

$\square$

**Lemma C.25.** *Under the assumptions of Lemma 3.5, for any step size scheduler $\eta_t$ such that*

$$\int_0^\infty \eta_s ds = \infty \text{ and } \lim_{t\to\infty}\eta_t = 0 \implies \mathbb{E}[f(X_t) - f(X_*)] \overset{t\to\infty}{\to} 0. \tag{107}$$

*Proof of Lemma C.25.* For any scheduler $\eta_k$ used in

$$x_{k+1} = x_k - \eta\eta_k \operatorname{sign}\left(f_{\gamma_k}(x_k)\right), \tag{108}$$

the SDE of Phase 3 is

$$dX_t = -\sqrt{\frac{2}{\pi}}\Sigma^{-\frac{1}{2}}\nabla f(X_t)\eta_t dt + \sqrt{\eta}\eta_t\sqrt{I_d - \frac{2}{\pi}\operatorname{diag}\left(\Sigma^{-\frac{1}{2}}\nabla f(X_t)\right)^2}dW_t. \tag{109}$$

Therefore, analogously to the calculations in Lemma C.22, we have that

$$\mathbb{E}[f(X_t) - f(X_*)] \leq \frac{f(X_0) - f(X_*) + \frac{\eta\mathcal{L}_\tau}{2}\int_0^t e^{2\mu\int_0^s\left(\sqrt{\frac{2}{\pi}}\frac{1}{\sigma_{\max}}\eta_l + \frac{\eta}{\pi}\frac{\mu}{\sigma_{\max}^2}\eta_l^2\right)dl}\eta_s^2 ds}{e^{2\mu\int_0^t\left(\sqrt{\frac{2}{\pi}}\frac{1}{\sigma_{\max}}\eta_s + \frac{\eta}{\pi}\frac{\mu}{\sigma_{\max}^2}\eta_s^2\right)ds}}. \tag{110}$$

Therefore, using l'Hôpital's rule we have that

$$\int_0^\infty \eta_s ds = \infty \text{ and } \lim_{t\to\infty}\eta_t = 0 \implies \mathbb{E}[f(X_t) - f(X_*)] \overset{t\to\infty}{\to} 0. \tag{111}$$

$\square$

**Lemma C.26.** *Let $H = \operatorname{diag}(\lambda_1, \ldots, \lambda_d)$ and $M_t := e^{-2\left(\sqrt{\frac{2}{\pi}}\Sigma^{-\frac{1}{2}}H + \frac{\eta}{\pi}\Sigma^{-1}H^2\right)t}$. Then,*

1. $\mathbb{E}[X_t] = e^{-\sqrt{\frac{2}{\pi}}\Sigma^{-\frac{1}{2}}Ht}X_0$;

2. $Var[X_t] = \left(M_t - e^{-2\sqrt{\frac{2}{\pi}}\Sigma^{-\frac{1}{2}}Ht}\right)X_0^2 + \frac{\eta}{2}\left(\sqrt{\frac{2}{\pi}}I_d + \frac{\eta}{\pi}H\Sigma^{-\frac{1}{2}}\right)^{-1}H^{-1}\Sigma^{\frac{1}{2}}\left(I_d - M_t\right).$

*Proof of Lemma C.26.* The proof is banal: The expected value derivation leverages the martingale property of the Brownian motion while that of the variance uses the Itô Isomerty. $\square$

**Lemma C.27.** *Let $H = \operatorname{diag}(\lambda_1, \ldots, \lambda_d)$. Then, $\mathbb{E}\left[\frac{X_t^\top H X_t}{2}\right]$ is equal to*

$$\sum_{i=1}^d \frac{\lambda_i(X_0^i)^2}{2}e^{-2\lambda_i\left(\sqrt{\frac{2}{\pi}}\frac{1}{\sigma_i} + \frac{\lambda_i\eta}{\pi\sigma_i^2}\right)t} + \frac{\eta}{4\left(\sqrt{\frac{2}{\pi}}\frac{1}{\sigma_i} + \frac{\lambda_i\eta}{\pi\sigma_i^2}\right)}\left(1 - e^{-2\lambda_i\left(\sqrt{\frac{2}{\pi}}\frac{1}{\sigma_i} + \frac{\lambda_i\eta}{\pi\sigma_i^2}\right)t}\right). \tag{112}$$

*Proof of Lemma C.27.* Since the matrix $H$ is diagonal, we focus on a single component. We apply the Itô Lemma to $\frac{\lambda_i(X_t^i)^2}{2}$:

$$d\left(\frac{\lambda_i(X_t^i)^2}{2}\right) = -2\sqrt{\frac{2}{\pi}}\frac{\lambda_i}{\sigma_i}\frac{\lambda_i(X_t^i)^2}{2}dt + \frac{\eta\lambda_i}{2}dt - \frac{2\lambda_i^2\eta}{\pi\sigma_i^2}\frac{\lambda_i(X_t^i)^2}{2}dt + \mathcal{O}(\text{Noise}), \quad (113)$$

which implies that

$$\mathbb{E}\left[\frac{\lambda_i(X_t^i)^2}{2}\right] = \frac{\lambda_i(X_0^i)^2}{2}e^{-2\left(\sqrt{\frac{2}{\pi}}\frac{\lambda_i}{\sigma_i}+\frac{\lambda_i^2\eta}{\pi\sigma_i^2}\right)t} + \frac{\eta}{4\left(\sqrt{\frac{2}{\pi}}\frac{1}{\sigma_i}+\frac{\lambda_i\eta}{\pi\sigma_i^2}\right)}\left(1 - e^{-2\left(\sqrt{\frac{2}{\pi}}\frac{\lambda_i}{\sigma_i}+\frac{\lambda_i^2\eta}{\pi\sigma_i^2}\right)t}\right). \quad (114)$$

Therefore,

$$\mathbb{E}\left[\frac{X_t^\top H X_t}{2}\right] = \sum_{i=1}^d \frac{\lambda_i(X_0^i)^2}{2}e^{-2\lambda_i\left(\sqrt{\frac{2}{\pi}}\frac{1}{\sigma_i}+\frac{\lambda_i\eta}{\pi\sigma_i^2}\right)t} + \frac{\eta}{4\left(\sqrt{\frac{2}{\pi}}\frac{1}{\sigma_i}+\frac{\lambda_i\eta}{\pi\sigma_i^2}\right)}\left(1 - e^{-2\lambda_i\left(\sqrt{\frac{2}{\pi}}\frac{1}{\sigma_i}+\frac{\lambda_i\eta}{\pi\sigma_i^2}\right)t}\right). \quad (115)$$

$\square$

**Lemma C.28.** *Under the assumptions of Corollary C.20, where $\nabla f_\gamma(x) = \nabla f(x) + \sqrt{\Sigma}Z$, we have that the dynamics of SignSGD in **Phase 3** is:*

$$dX_t = -\sqrt{\frac{1}{2}}\Sigma^{-\frac{1}{2}}\nabla f(X_t)dt + \sqrt{\eta}\sqrt{I_d - \frac{1}{2}\text{diag}\left(\Sigma^{-\frac{1}{2}}\nabla f(X_t)\right)^2}dW_t. \quad (116)$$

*Proof of lemma C.28.* We apply Eq. 44 with $\nu = 2$ and linearly approximate $\Xi(x)$ as $|x| < 1$, where $2\Xi(x) \sim \frac{x}{\sqrt{2}}$. $\square$

## C.5 ALTERNATIVE NOISE ASSUMPTIONS

In this subsection, we report the consequences of assuming different noise structures. We do not provide the proofs as they mimic those of Corollary C.19 and Lemma C.21. We validate our results in Figure 7.

**Assumption from (Ziyin et al., 2021)** As per Eq. (16) in Corollary 2 of (Ziyin et al., 2021), we take $\Sigma := \sigma^2 f(x_*)\nabla^2 f(x_*)$, where we added the constant $\sigma^2$ as a parameter to control the scale of the noise and $f(x_*) > 0$. Under this assumption, we have that for $Y_t := \frac{\nabla^2 f(x_*)^{-\frac{1}{2}}\nabla f(X_t)}{\sqrt{2\nabla f(x_*)\sigma}}$ and $\mathcal{S}_d(X_t) := \mathbb{E}_\gamma[(\text{sign}(\nabla f_\gamma(X_t)))(\text{sign}(\nabla f_\gamma(X_t))^\top]$, Corollary C.19 becomes:

$$dX_t = -\text{Erf}(Y_t)dt + \sqrt{\eta}\sqrt{\mathcal{S}_d(X_t) - \text{Erf}(Y_t)\text{Erf}(Y_t)^\top}dW_t. \quad (117)$$

As a consequence, Lemma C.21 becomes:

**Lemma C.29.** *Let $f$ be $\mu$-strongly convex, $Tr(\nabla^2 f(x)) \leq \mathcal{L}_\tau$, $\lambda_{max}$ be the largest eigenvalue of $\nabla^2 f(x_*)$, and $S_t := f(X_t) - f(x_*)$. Then, during*

1. *Phase 1, the loss will reach $0$ before $t_* = 2\sqrt{\frac{S_0}{\mu}}$ because $S_t \leq \frac{1}{4}\left(\sqrt{\mu}t - 2\sqrt{S_0}\right)^2$;*

2. *Phase 2 with $\Delta := \left(\frac{m}{\sqrt{2f(x_*)}\sigma_{max}\sqrt{\lambda_{max}}} + \frac{\eta\mu m^2}{4f(x_*)\sigma_{max}^2\lambda_{max}}\right)$: $\mathbb{E}[S_t] \leq S_0 e^{-2\mu\Delta t} + \frac{\eta}{2}\frac{(\mathcal{L}_\tau - \mu d\hat{q}^2)}{2\mu\Delta}\left(1 - e^{-2\mu\Delta t}\right)$;*

3. *Phase 3 with $\Delta := \left(\sqrt{\frac{2}{\pi}}\frac{1}{\sqrt{f(x_*)}\sigma_{max}\sqrt{\lambda_{max}}} + \frac{\eta}{\pi}\frac{\mu}{f(x_*)\sigma_{max}^2\lambda_{max}}\right)$: $\mathbb{E}[S_t] \leq S_0 e^{-2\mu\Delta t} + \frac{\eta}{2}\frac{\mathcal{L}_\tau}{2\mu\Delta}\left(1 - e^{-2\mu\Delta t}\right)$.*

**Assumption from (Wojtowytsch, 2024)** (Wojtowytsch, 2024) discusses two possible assumptions on $\Sigma$: $\|\Sigma(x)\| \leq Cf(x)$ and $\|\Sigma(x)\| \leq Cf(x)\left[1 + |x|^2\right]$. As Section 2.4, they ultimately use $\Sigma = Cf(x)I_d$. Therefore, we take $\Sigma := \sigma^2 f(x)I_d$, where we changed the constant to $\sigma^2$ to maintain consistency with the rest of our paper. Under this assumption, we have that for $Y_t := \frac{\nabla f(X_t)}{\sqrt{2f(x)\sigma}}$, Corollary C.19 becomes:

$$dX_t = -\operatorname{Erf}(Y_t)\,dt + \sqrt{\eta}\sqrt{I_d - \operatorname{diag}(\operatorname{Erf}(Y_t))^2}\,dW_t. \tag{118}$$

As a consequence, Lemma C.21 becomes:

**Lemma C.30.** *Let $f$ be $\mu$-strongly convex, $Tr(\nabla^2 f(x)) \leq \mathcal{L}_\tau$, and $S_t := f(X_t) - f(X_*)$. Then, during*

1. *Phase 1, the loss will reach $0$ before $t_* = 2\sqrt{\frac{S_0}{\mu}}$ because $S_t \leq \frac{1}{4}\left(\sqrt{\mu}t - 2\sqrt{S_0}\right)^2$;*

2. *Phase 2 with $\beta := \frac{\eta}{2}\left(\mathcal{L}_\tau - \mu d\hat{q}^2 - \frac{m^2\mu^2}{\sigma^2}\right)$ and $\alpha := \frac{\sqrt{2}m\mu}{\sigma}$,*

$$\mathbb{E}[S_t] \leq \frac{\beta^2\left(\mathcal{W}\left(\frac{(\beta+\sqrt{S_0}\alpha)}{\beta}\exp\left(-\frac{\alpha^2 t - 2\sqrt{S_0}\alpha}{2\beta} - 1\right)\right) + 1\right)^2}{\alpha^2} \overset{t\to\infty}{\to} \frac{\beta^2}{\alpha^2}; \tag{119}$$

3. *Phase 3 with $\beta := \eta\left(\frac{\mathcal{L}_\tau}{2} - \frac{2\mu^2}{\pi\sigma^2}\right)$ and $\alpha := 2\sqrt{\frac{2}{\pi}}\frac{\mu}{\sigma}$,*

$$\mathbb{E}[S_t] \leq \frac{\beta^2\left(\mathcal{W}\left(\frac{(\beta+\sqrt{S_0}\alpha)}{\beta}\exp\left(-\frac{\alpha^2 t - 2\sqrt{S_0}\alpha}{2\beta} - 1\right)\right) + 1\right)^2}{\alpha^2} \overset{t\to\infty}{\to} \frac{\beta^2}{\alpha^2}, \tag{120}$$

*where $\mathcal{W}$ is the Lambert $\mathcal{W}$ function.*

**Assumption from (Wu et al., 2022)** (Wu et al., 2022) proposes a novel structure of $\Sigma$ as being aligned with the Fisher Information Matrix and proportional to the loss function. Consistently with this, we take $\Sigma := \sigma^2 f(x)\nabla^2 f(x)$, where we changed the constants to $\sigma^2$ to maintain consistency with the rest of our paper. Under this assumption, we have that for $Y_t := \frac{(\nabla^2 f(X_t))^{-\frac{1}{2}}\nabla f(X_t)}{\sqrt{2\nabla f(x)\sigma}}$ and $\mathcal{S}_d(X_t) := \mathbb{E}_\gamma[(\operatorname{sign}(\nabla f_\gamma(X_t)))(\operatorname{sign}(\nabla f_\gamma(X_t))^\top]$, Corollary C.19 becomes:

$$dX_t = -\operatorname{Erf}(Y_t)\,dt + \sqrt{\eta}\sqrt{\mathcal{S}_d(X_t) - \operatorname{Erf}(Y_t)\operatorname{Erf}(Y_t)^\top}\,dW_t. \tag{121}$$

As a consequence, Lemma C.21 becomes:

**Lemma C.31.** *Let $f$ be $\mu$-strongly convex, $L$-smooth, $Tr(\nabla^2 f(x)) \leq \mathcal{L}_\tau$, and $S_t := f(X_t) - f(X_*)$. Then, during*

1. *Phase 1, the loss will reach $0$ before $t_* = 2\sqrt{\frac{S_0}{\mu}}$ because $S_t \leq \frac{1}{4}\left(\sqrt{\mu}t - 2\sqrt{S_0}\right)^2$;*

2. *Phase 2 with $\beta := \frac{\eta}{2}\left(\mathcal{L}_\tau - \mu d\hat{q}^2 - \frac{m^2\mu^2}{\sigma^2 L}\right)$ and $\alpha := \frac{\sqrt{2}m\mu}{\sqrt{L}\sigma}$,*

$$\mathbb{E}[S_t] \leq \frac{\beta^2\left(\mathcal{W}\left(\frac{(\beta+\sqrt{S_0}\alpha)}{\beta}\exp\left(-\frac{\alpha^2 t - 2\sqrt{S_0}\alpha}{2\beta} - 1\right)\right) + 1\right)^2}{\alpha^2} \overset{t\to\infty}{\to} \frac{\beta^2}{\alpha^2}; \tag{122}$$

3. *Phase 3 with $\beta := \eta\left(\frac{\mathcal{L}_\tau}{2} - \frac{2\mu^2}{\pi\sigma^2 L}\right)$ and $\alpha := 2\sqrt{\frac{2}{\pi}}\frac{\mu}{\sqrt{L}\sigma}$,*

$$\mathbb{E}[S_t] \leq \frac{\beta^2\left(\mathcal{W}\left(\frac{(\beta+\sqrt{S_0}\alpha)}{\beta}\exp\left(-\frac{\alpha^2 t - 2\sqrt{S_0}\alpha}{2\beta} - 1\right)\right) + 1\right)^2}{\alpha^2} \overset{t\to\infty}{\to} \frac{\beta^2}{\alpha^2}, \tag{123}$$

*where $\mathcal{W}$ is the Lambert $\mathcal{W}$ function.*

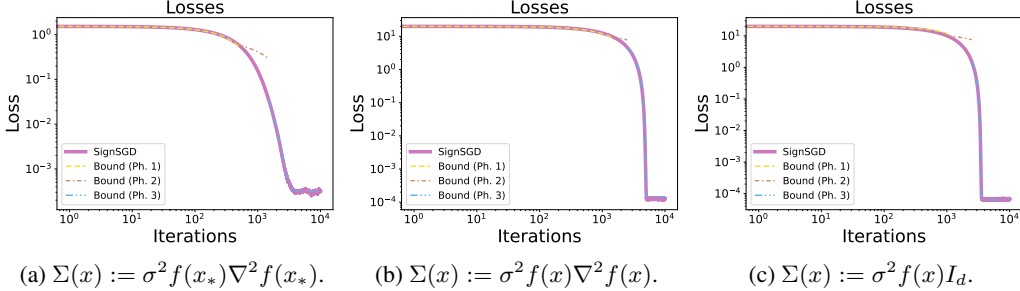

(a) $\Sigma(x) := \sigma^2 f(x_*)\nabla^2 f(x_*)$.    (b) $\Sigma(x) := \sigma^2 f(x)\nabla^2 f(x)$.    (c) $\Sigma(x) := \sigma^2 f(x)I_d$.

Figure 7: Empirical validation of the bounds for SignSGD derived for the noise structure: $\Sigma(x) := \sigma^2 f(x_*)\nabla^2 f(x_*)$ (left), $\Sigma(x) := \sigma^2 f(x)\nabla^2 f(x)$ (center), $\Sigma(x) := \sigma^2 f(x)I_d$ (right). Each empirical validation has been carried out on strongly convex quadratic loss functions. The experiments are averaged over 500 runs.

**Assumption from (Paquette et al., 2024)**   In this section, we adopt the notation of (Paquette et al., 2024). As per Eq. 5 of (Paquette et al., 2024), the loss function can be rewritten as

$$f(\theta) = \frac{1}{2}\langle D(W\theta - b), (W\theta - b)\rangle, \quad \text{where } D = \text{diag}\left(j^{-2\alpha}\right) \in \mathbb{R}^{v \times v}. \tag{124}$$

Without loss of generality, we define $\phi := W\theta - b$, which implies that

$$f(\theta) = \frac{\phi^\top D\phi}{2}, \quad \text{where } D = \text{diag}\left(j^{-2\alpha}\right) \in \mathbb{R}^{v \times v}, \text{ where }, 1 \le j \le v. \tag{125}$$

The stochastic gradient is unbiased and its covariance is the well-known $B\Sigma(\phi) = (\phi^\top D\phi)D + D\phi\phi^\top D = 2f(\phi)D + \nabla f(\phi)\nabla f(\phi)^\top$, where $B$ is the batch size. Under this assumption, we have that for $Y_t := \frac{\sqrt{B}(\Sigma(\phi_t))^{-\frac{1}{2}}\nabla f(\phi_t)}{\sqrt{2}}$ and $\mathcal{S}(\phi_t) = \mathbb{E}[(Sign(\nabla f_\gamma(\phi_t)))(Sign(\nabla f_\gamma(\phi_t))^\top]$, Corollary C.19 becomes:

$$d\phi_t = -Erf\left(Y_t\right)dt + \sqrt{\eta}\sqrt{\mathcal{S}(\phi_t) - Erf\left(Y_t\right)Erf\left(Y_t\right)^\top}dW_t. \tag{126}$$

As a consequence, Lemma C.21 becomes:

**Lemma C.32.** *Let $f$ be as above, $f_t := f(\phi_t)$, and $\mathcal{L}_\tau := Tr(D)$. Let $\mu$ be the minimum eigenvalue of $D$, and $L$ be its maximum one. Then, during*

1. *Phase 1, the loss will reach 0 before $t_* = 2\sqrt{\frac{S_0}{\mu}}$ because $f_t \le \frac{1}{4}\left(\sqrt{\mu}t - 2\sqrt{f_0}\right)^2$;*

2. *Phase 2 with $\beta := \frac{\eta}{2}\mathcal{L}_\tau$ and $\alpha := \frac{m\mu\sqrt{B}}{\sqrt{2L}}$,*

$$\mathbb{E}[S_t] \le \frac{\beta^2\left(\mathcal{W}\left(\frac{(\beta+\sqrt{S_0}\alpha)}{\beta}\exp\left(-\frac{\alpha^2 t - 2\sqrt{S_0}\alpha}{2\beta} - 1\right)\right) + 1\right)^2}{\alpha^2} \overset{t\to\infty}{\Rightarrow} \frac{\beta^2}{\alpha^2}; \tag{127}$$

3. *Phase 3 with $\beta := \frac{\eta}{2}\mathcal{L}_\tau$ and $\alpha := \sqrt{\frac{2}{\pi}}\frac{\mu\sqrt{B}}{\sqrt{L}}$;*

$$\mathbb{E}[S_t] \le \frac{\beta^2\left(\mathcal{W}\left(\frac{(\beta+\sqrt{S_0}\alpha)}{\beta}\exp\left(-\frac{\alpha^2 t - 2\sqrt{S_0}\alpha}{2\beta} - 1\right)\right) + 1\right)^2}{\alpha^2} \overset{t\to\infty}{\Rightarrow} \frac{\beta^2}{\alpha^2}, \tag{128}$$

*where $\mathcal{W}$ is the Lambert $\mathcal{W}$ function.*

See Figure 8 for an empirical validation.

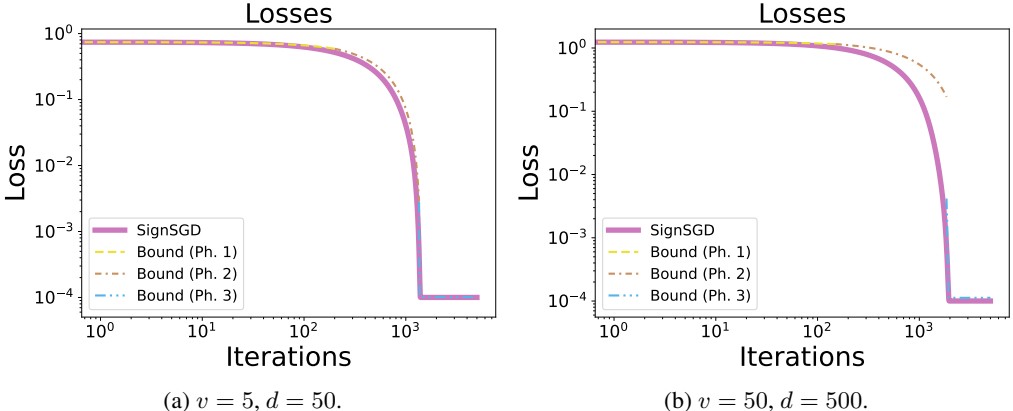

Figure 8: Empirical validation of the bounds for SignSGD derived in Lemma C.32: In both experiments, $\alpha = 0.25$, $\beta = 2$, $\eta = 0.001$, $B = 256$, $N = 10000$, and trajectories are averaged over 500 runs.

## C.6 FORMAL DERIVATION - RMSPROP

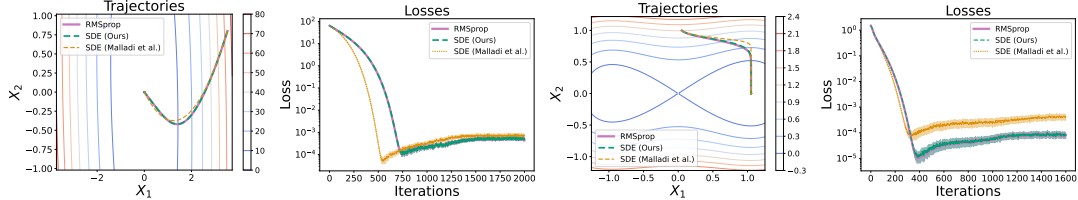

Figure 9: The first two subfigures on the left compare our SDE, that from Malladi et al. (2022), and RMSprop in terms of trajectories and $f(x)$, respectively, for a convex quadratic function. The other subfigures do the same for an embedded saddle, and one can observe that our SDE tracks RMSprop more faithfully.

In this subsection, we provide our formal derivation of an SDE model for RMSprop. Let us consider the stochastic process $L_t := (X_t, V_t) \in \mathbb{R}^d \times \mathbb{R}^d$ defined as the solution of

$$dX_t = -P_t^{-1}(\nabla f(X_t)dt + \sqrt{\eta}\Sigma(X_t)^{\frac{1}{2}}dW_t) \tag{129}$$

$$dV_t = \rho((\nabla f(X_t))^2 + \text{diag}(\Sigma(X_t)) - V_t)dt, \tag{130}$$

where $\beta = 1 - \eta\rho$, $\rho = \mathcal{O}(1)$, and $P_t := \text{diag}\left(V_t\right)^{\frac{1}{2}} + \epsilon I_d$.

*Remark* C.33. We observe that the term in blue is the only difference w.r.t. the SDE derived in (Malladi et al., 2022) (see Theorem D.2): This is extremely relevant when the gradient size is not negligible. Figure 9 shows the comparison between our SDE, the one derived in (Malladi et al., 2022), and RMSprop itself: It is clear that even on simple landscapes, our SDE tracks the algorithm more faithfully. Importantly, one can observe that the SDE derived in (Malladi et al., 2022) is only slightly less accurate than ours at the end of the dynamics: As we show in Lemma C.37, Theorem D.2 is a corollary of Theorem C.34 when $\nabla f(x) = \mathcal{O}(\sqrt{\eta})$, e.g. it only describes the dynamics where the gradient is vanishing. In Figure 10, we compare the two SDEs in question with RMSprop on an MLP, a CNN, a ResNet, and a Transformer: Our SDE exhibits a superior description of the dynamics.

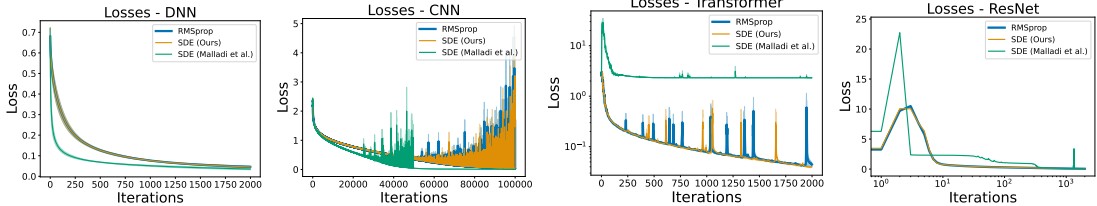

Figure 10: We compare our SDE, that from Malladi et al. (2022), and RMSprop in terms of $f(x)$: The first is an MLP on the Breast Cancer dataset, the second a CNN on MNIST, the third a Transformer on MNIST, and the last a ResNet on CIFAR-10: One can observe that our SDE tracks RMSprop more faithfully.

The following theorem guarantees that such a process is a 1-order SDE of the discrete-time algorithm of RMSprop

$$x_{k+1} = x_k - \eta \frac{\nabla f_{\gamma_k}(x_k)}{\sqrt{v_{k+1}} + \epsilon I_d} \tag{131}$$

$$v_{k+1} = \beta v_k + (1 - \beta) \left(\nabla f_{\gamma_k}(x_k)\right)^2 \tag{132}$$

with $(x_0, v_0) \in \mathbb{R}^d \times \mathbb{R}^d$, $\eta \in \mathbb{R}^{>0}$ is the step size, $\beta = 1 - \rho\eta$ for $\rho = \mathcal{O}(1)$, the mini-batches $\{\gamma_k\}$ are modelled as i.i.d. random variables uniformly distributed on $\{1, \cdots, N\}$, and of size $B \geq 1$.

> **Theorem C.34** (Stochastic modified equations). *Let $0 < \eta < 1, T > 0$ and set $N = \lfloor T/\eta \rfloor$. Let $l_k := (x_k, v_k) \in \mathbb{R}^d \times \mathbb{R}^d, 0 \leq k \leq N$ denote a sequence of RMSprop iterations defined by Eq. 131. Consider the stochastic process $L_t$ defined in Eq. 129 and fix some test function $g \in G$ and suppose that $g$ and its partial derivatives up to order 6 belong to $G$. Then, under Assumption C.3 and $\rho = \mathcal{O}(1)$ there exists a constant $C > 0$ independent of $\eta$ such that for all $k = 0, 1, \ldots, N$, we have*
>
> $$|\mathbb{E}g\left(L_{k\eta}\right) - \mathbb{E}g\left(l_k\right)| \leq C\eta.$$
>
> *That is, the SDE 129 is an order 1 weak approximation of the RMSprop iterations 131.*

*Proof.* The proof is virtually identical to that of Theorem C.16. Therefore, we only report the key steps necessary to conclude the thesis. First of all, we observe that since $\beta = 1 - \eta\rho$

$$v_{k+1} - v_k = -\eta\rho \left(v_k - \left(\nabla f_{\gamma_k}(x_k)\right)^2\right). \tag{133}$$

Then,

$$\frac{1}{\sqrt{v_{k+1}}} = \sqrt{\frac{v_k}{v_{k+1}} \frac{1}{v_k}} = \sqrt{\frac{v_{k+1} + \mathcal{O}(\eta)}{v_{k+1}} \frac{1}{v_k}} = \sqrt{1 + \frac{\mathcal{O}(\eta)}{v_{k+1}}} \sqrt{\frac{1}{v_k}} \sim \sqrt{\frac{1}{v_k}}(1 + \mathcal{O}(\eta)). \tag{134}$$

Therefore, we work with the following algorithm as all the approximations below only carry an additional error of order $\mathcal{O}(\eta^2)$, which we can ignore. Therefore, we have that

$$x_{k+1} - x_k = -\eta \frac{\nabla f_{\gamma_k}(x_k)}{\sqrt{v_k} + \epsilon I_d} \tag{135}$$

$$v_k - v_{k-1} = -\eta\rho \left(v_{k-1} - \left(\nabla f_{\gamma_{k-1}}(x_{k-1})\right)^2\right). \tag{136}$$

Therefore, if $\nabla f_{\gamma_j}(x_j) = \nabla f(x_j) + Z_j(x_j)$, $\mathbb{E}[Z_j(x_j)] = 0$, and $Cov(Z_j(x_j)) = \Sigma(x_j)$

1. $\mathbb{E}[x_{k+1} - x_k] = -\eta \operatorname{diag}(v_k + \epsilon I_d)^{-\frac{1}{2}} \nabla f(x_k)$ ;

2. $\mathbb{E}[v_k - v_{k-1}] = \eta\rho \left[\left(\nabla f(x_{k-1})\right)^2 + \operatorname{diag}(\Sigma(x_k)) - v_{k-1}\right]$ .

Then, we have that if $\Phi_k := \frac{\nabla f(x_k)}{\sqrt{v_k} + \epsilon I_d} - \frac{\nabla f_{\gamma_k}(x_k)}{\sqrt{v_k} + \epsilon I_d}$

1.

$$\mathbb{E}[(x_{k+1} - x_k)(x_{k+1} - x_k)^\top] = \mathbb{E}[(x_{k+1} - x_k)]\mathbb{E}[(x_{k+1} - x_k)]^\top \tag{137}$$

$$+ \eta^2 \mathbb{E}\left[(\Phi_k)(\Phi_k)^\top\right] \tag{138}$$

$$= \mathbb{E}[(x_{k+1} - x_k)]\mathbb{E}[(x_{k+1} - x_k)]^\top \tag{139}$$

$$+ \eta^2 (\mathrm{diag}(v_k) + \epsilon I_d)^{-1} \Sigma(x_k); \tag{140}$$

2. $\mathbb{E}[(v_k - v_{k-1})(v_k - v_{k-1})^\top] = \mathbb{E}[(v_k - v_{k-1})]\mathbb{E}[(v_k - v_{k-1})]^\top + \mathcal{O}(\rho\eta^2);$

3. $\mathbb{E}[(x_{k+1} - x_k)(v_k - v_{k-1})^\top] = \mathbb{E}[(x_{k+1} - x_k)]\mathbb{E}[(v_k - v_{k-1})^\top] + 0.$

*Remark* C.35. Let us remember that by assumption, $\nabla f(x)$ and $\sqrt{\Sigma}(x)$ are Lipschitz, grow at most affinely, and are in $G$ together with their derivative. Therefore, the drift and diffusion terms of the SDE governing $X_t$ are the ratio between regular functions and a uniformly lower bounded process. Therefore, they are in turn regular, modulo dividing by $\sqrt{V_t + \epsilon_V^2} + \epsilon$ s.t. $\epsilon_V^2 \sim 0$ rather than by $\sqrt{V_t} + \epsilon$ (See Bock and Weiß (2021) as they experimentally verify that this has no impact on the performance of the optimizer). Regarding the ODE governing $V_t$, $\Sigma(X_t)$ is Lipschitz because $\sqrt{\Sigma}(X_t)$ is bounded and Lipschitz. Additionally, it is smooth, and with affine growth. On top of this, we need the term $(\nabla f(x))^2$ to be Lipschitz and of affine growth, which is a consequence of assuming bounded gradients as often done in the literature on the convergence of RMSprop and Adam: Among many, see (Luo et al., 2019; Défossez et al., 2022; Guo et al., 2021; Huang et al., 2021) together with the discussion in Section 2.1 of Shi and Li (2021). Alternatively, exactly as done in Theorem 9 of Li et al. (2019), one can regularize the drifts and the diffusion terms with mollifiers on a sufficiently large compact Reddi et al. (2018), which automatically implies that drift and diffusion coefficients satisfy all necessary regularity conditions. Importantly, one needs to then send the mollification parameter $\epsilon$ to 0 to conclude our statement. Therefore, we the SDE of RMSprop for $P_t := \mathrm{diag}\,(V_t)^{\frac{1}{2}} + \epsilon I_d$ is

$$dX_t = -P_t^{-1}(\nabla f(X_t)dt + \sqrt{\eta}\Sigma(X_t)^{\frac{1}{2}}dW_t) \tag{141}$$

$$dV_t = \rho(((\nabla f(X_t))^2 + \mathrm{diag}(\Sigma(X_t)) - V_t))dt. \tag{142}$$

$\square$

*Remark* C.36. In all the following results, the reader will notice that all the drifts, diffusion terms, and noise assumptions are selected to guarantee that the SDE we derived for RMSprop is indeed a 1 weak approximation for RMSprop even without the mollification argument. Importantly, our analysis of RMSprop focuses on its behavior at convergence, i.e. $(\nabla f(x))^2 = \mathcal{O}(\eta)$. Therefore, there is no need to assume bounded gradients or a compact domain.

**Lemma C.37.** *If $(\nabla f(x))^2 = \mathcal{O}(\eta)$, Theorem D.2 is a Corollary of Theorem C.34.*

*Proof.* In the proof of Theorem C.34, one drops the term $\eta(\nabla f(x))^2$ as it is of order $\eta^2$. $\square$

**Corollary C.38.** *Under the assumptions of Theorem C.34 with $\Sigma(x) = \sigma^2 I_d$, $\tilde{\eta} = \kappa\eta$, $\tilde{B} = B\delta$, and $\tilde{\rho} = \alpha\rho$,*

$$dX_t = \kappa \,\mathrm{diag}(V_t)^{-\frac{1}{2}}\left(-\nabla f(X_t)dt + \frac{1}{\sqrt{\delta}}\sqrt{\frac{\eta}{B}}\sigma I_d dW_t\right) \tag{143}$$

$$dV_t = \alpha\rho\left((\nabla f(X_t))^2 + \frac{\sigma^2}{B\delta}\mathbf{1} - V_t\right)dt. \tag{144}$$

**Lemma C.39** (Scaling Rule at Convergence). *Under the assumptions of Corollary C.38, $f$ is $\mu$-strongly convex, $Tr(\nabla^2 f(x)) \leq \mathcal{L}_\tau$, and $(\nabla f(x))^2 = \mathcal{O}(\eta)$, the asymptotic dynamics of the iterates of RMSprop satisfies the classic scaling rule $\kappa = \sqrt{\delta}$ because*

$$\mathbb{E}[f(X_t) - f(X_*)] \overset{t\to\infty}{\leq} \frac{\eta\sigma\mathcal{L}_\tau}{4\mu\sqrt{B}}\frac{\kappa}{\sqrt{\delta}}. \tag{145}$$

*By enforcing that the speed of $V_t$ matches that of $X_t$, one needs $\tilde{\rho} = \kappa\rho$, which implies $\tilde{\beta} = 1 - \kappa(1 - \beta)$.*

*Proof of Lemma C.39.* In order to recover the scaling of $\beta$, we enforce that the rate at which $V_t$ converges to its limit matches the speed of $X_t$: We need $\tilde{\rho} = \kappa\rho$, which recovers the novel scaling $\tilde{\beta} = 1 - \kappa(1 - \beta)$. Additionally, since $(\nabla f(x))^2 = \mathcal{O}(\eta)$ we have that

$$dX_t = \kappa \operatorname{diag}(V_t)^{-\frac{1}{2}}\left(-\nabla f(X_t)dt + \frac{1}{\sqrt{\delta}}\sqrt{\frac{\eta}{B}}\sigma I_d dW_t\right) \tag{146}$$

$$dV_t = \kappa\rho\left(\frac{\sigma^2}{B\delta}\mathbf{1} - V_t\right)dt. \tag{147}$$

Therefore, $V_t \overset{t\to\infty}{\to} \frac{\sigma^2}{B\delta}\mathbf{1}$, meaning that under these conditions:

$$dX_t = -\frac{\sqrt{B\delta}\kappa}{\sigma}\nabla f(X_t)dt + \kappa\sqrt{\eta}I_d dW_t, \tag{148}$$

which satisfies the following for $\mu$-strongly convex functions

$$d\mathbb{E}[f(X_t) - f(X_*)] \leq -2\kappa\mu\frac{\sqrt{B\delta}}{\sigma}\mathbb{E}[f(X_t) - f(X_*)]dt + \frac{\kappa^2\eta\mathcal{L}_\tau}{2}dt, \tag{149}$$

meaning that $\mathbb{E}[f(X_t) - f(X_*)] \overset{t\to\infty}{\leq} \frac{\eta\sigma\mathcal{L}_\tau}{4\mu\sqrt{B}}\frac{\kappa}{\sqrt{\delta}}$.

Since the asymptotic the loss is $\frac{\eta}{2}\frac{\mathcal{L}_\tau\sigma}{2\mu\sqrt{B}}\frac{\kappa}{\sqrt{\delta}}$ does not depend on $\kappa$ and $\delta$ if $\frac{\kappa}{\sqrt{\delta}} = 1$, we recover the classic scaling rule. $\square$

**Remark:** Under the same conditions, SGD satisfies

$$dX_t = -\kappa\nabla f(X_t)dt + \kappa\frac{1}{\sqrt{\delta}}\sqrt{\frac{\eta}{B}}\sigma I_d dW_t \tag{150}$$

and therefore

$$\mathbb{E}[f(X_t) - f(X_*)] \leq (f(X_0) - f(X_*))e^{-2\mu\kappa t} + \frac{\eta}{2}\frac{\mathcal{L}_\tau\sigma^2}{2\mu B}\frac{\kappa}{\delta}\left(1 - e^{-2\mu\kappa t}\right), \tag{151}$$

meaning that asymptotically the loss is $\frac{\eta}{2}\frac{\mathcal{L}_\tau\sigma^2}{2\mu B}\frac{\kappa}{\delta}$ which does not depend on $\kappa$ and $\delta$ if $\frac{\kappa}{\delta} = 1$.

**Lemma C.40.** *For $f(x) := \frac{x^\top Hx}{2}$, the stationary distribution of RMSprop is $(\mathbb{E}[X_\infty]], Cov(X_\infty)) = \left(0, \frac{\eta}{2}\Sigma^{\frac{1}{2}}H^{-1}\right)$.*

*Proof.* As $(\nabla f(x))^2 = \mathcal{O}(\eta)$ and $t \to \infty$, we have

$$dX_t = -\Sigma^{-\frac{1}{2}}HX_t dt + \sqrt{\eta}I_d dW_t \tag{152}$$

which implies that

$$X_t = e^{-\Sigma^{-\frac{1}{2}}Ht}\left(X_0 + \sqrt{\eta}\int_0^t e^{\Sigma^{-\frac{1}{2}}Hs}dW_s\right). \tag{153}$$

The thesis follows from the martingale property of Brownian motion and the Itô isometry. $\square$

## C.7 RMSPROPW

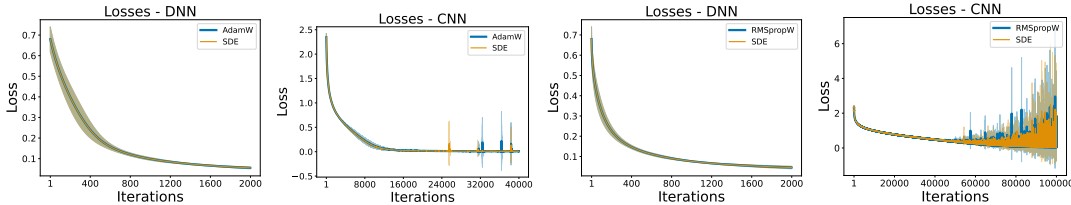

Figure 11: The first two represent the comparison between AdamW and its SDE in terms of $f(x)$. The other two do the same for RMSpropW. In both cases, the first is an MLP on the Breast Cancer Dataset and the second a CNN on MNIST: Our SDEs match the respective optimizers.

In this subsection, we derive the SDE of RMSpropW defined as

$$x_{k+1} = x_k - \eta \frac{\nabla f_{\gamma_k}(x_k)}{\sqrt{v_{k+1}} + \epsilon I_d} - \eta\theta x_k \tag{154}$$

$$v_{k+1} = \beta v_k + (1-\beta)\left(\nabla f_{\gamma_k}(x_k)\right)^2 \tag{155}$$

with $(x_0, v_0) \in \mathbb{R}^d \times \mathbb{R}^d$, $\eta \in \mathbb{R}^{>0}$ is the step size, $\beta = 1 - \rho\eta$ for $\rho = \mathcal{O}(1)$, $\theta > 0$, the mini-batches $\{\gamma_k\}$ are modelled as i.i.d. random variables uniformly distributed on $\{1, \cdots, N\}$, and of size $B \geq 1$.

**Theorem C.41.** *Under the same assumptions as Theorem C.34, the SDE of RMSpropW is*

$$dX_t = -P_t^{-1}(\nabla f(X_t)dt + \sqrt{\eta}\Sigma(X_t)^{\frac{1}{2}}dW_t) - \theta X_t dt \tag{156}$$

$$dV_t = \rho((\nabla f(X_t))^2 + \text{diag}(\Sigma(X_t)) - V_t))dt, \tag{157}$$

*where $\beta = 1 - \eta\rho$, $\rho = \mathcal{O}(1)$, $\theta > 0$, and $P_t := \text{diag}\left(V_t\right)^{\frac{1}{2}} + \epsilon I_d$.*

*Proof.* The proof is the same as the of Theorem C.34 and the only difference is that $\eta\theta x_k$ is approximated with $\theta X_t dt$. □

Figure 4 and Figure 11 validate this result on a variety of architectures and datasets.

*Remark* C.42. See Remark C.35 and Remark C.36 for a discussion on the regularity of the SDE derived in Theorem C.41.

**Corollary C.43.** *Under the assumptions of Theorem C.41 with $\Sigma(x) = \sigma^2 I_d$, $\tilde{\eta} = \kappa\eta$, $\tilde{B} = B\delta$, and $\tilde{\rho} = \alpha\rho$, and $\tilde{\theta} = \xi\theta$,*

$$dX_t = \kappa\,\text{diag}(V_t)^{-\frac{1}{2}}\left(-\nabla f(X_t)dt + \frac{1}{\sqrt{\delta}}\sqrt{\frac{\eta}{B}}\sigma I_d dW_t\right) - \xi\theta\kappa X_t dt \tag{158}$$

$$dV_t = \alpha\rho\left((\nabla f(X_t))^2 + \frac{\sigma^2}{B\delta}\mathbf{1} - V_t\right)dt. \tag{159}$$

**Lemma C.44** (Scaling Rule at Convergence)**.** *Under the assumptions of Corollary C.43, $f$ is $\mu$-strongly convex and $L$-smooth, $Tr(\nabla^2 f(x)) \leq \mathcal{L}_\tau$, $X_* = 0$, and $(\nabla f(x))^2 = \mathcal{O}(\eta)$, the asymptotic dynamics of the iterates of RMSpropW satisfies the novel scaling rule if $\kappa = \sqrt{\delta}$ and $\xi = \kappa$ because*

$$\mathbb{E}[f(X_t) - f(X_*)] \overset{t\to\infty}{\leq} \frac{\eta\mathcal{L}_\tau\sigma L}{2}\frac{\kappa}{2\mu\sqrt{B\delta}L + \sigma\xi\theta(L+\mu)}. \tag{160}$$

*By enforcing that the speed of $V_t$ matches that of $X_t$, one needs $\tilde{\rho} = \kappa\rho$, which implies $\tilde{\beta} = 1 - \kappa(1-\beta)$.*

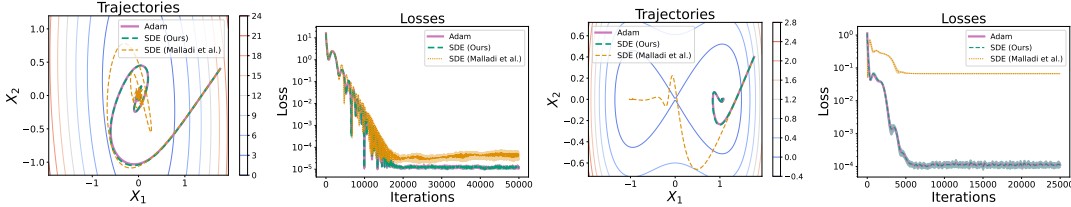

Figure 12: The first two on the left compare our SDE, that from Malladi et al. (2022), and Adam in terms of trajectories and $f(x)$, respectively, for a convex quadratic function. The others do the same for an embedded saddle: Our SDE is shown here to track Adam more faithfully.

*Proof of Lemma C.44.* In order to recover the scaling of $\beta$, we enforce that the rate at which $V_t$ converges to its limit matches the speed of $X_t$: We need $\tilde{\rho} = \kappa\rho$, which recovers the novel scaling $\tilde{\beta} = 1 - \kappa(1 - \beta)$. Additionally, since $(\nabla f(x))^2 = \mathcal{O}(\eta)$ we have that

$$dX_t = \kappa\,\mathrm{diag}(V_t)^{-\frac{1}{2}}\left(-\nabla f(X_t)dt + \frac{1}{\sqrt{\delta}}\sqrt{\frac{\eta}{B}}\sigma I_d dW_t\right) - \kappa\xi\theta X_t dt \tag{161}$$

$$dV_t = \kappa\rho\left(\frac{\sigma^2}{B\delta}\mathbf{1} - V_t\right)dt. \tag{162}$$

Therefore, $V_t \overset{t\to\infty}{\to} \frac{\sigma^2}{B\delta}\mathbf{1}$, meaning that under these conditions:

$$dX_t = -\frac{\sqrt{B\delta}\kappa}{\sigma}\nabla f(X_t)dt + \kappa\sqrt{\eta}I_d dW_t - \kappa\xi\theta X_t dt, \tag{163}$$

which satisfies the following for $\mu$-strongly convex and $L$-smooth functions

$$d\mathbb{E}[f(X_t) - f(X_*)] \leq \kappa\left(2\mu\frac{\sqrt{B\delta}}{\sigma} + \xi\theta\left(1 + \frac{\mu}{L}\right)\right)\mathbb{E}[f(X_t) - f(X_*)]dt + \frac{\kappa^2\eta\mathcal{L}_\tau}{2}dt, \tag{164}$$

meaning that $\mathbb{E}[f(X_t) - f(X_*)] \overset{t\to\infty}{\leq} \frac{\eta\mathcal{L}_\tau\sigma L}{2}\frac{\kappa}{2\mu\sqrt{B\delta}L + \sigma\xi\theta(L+\mu)}$.

Since the asymptotic the loss $\frac{\eta\mathcal{L}_\tau\sigma L}{2}\frac{\kappa}{2\mu\sqrt{B\delta}L + \sigma\xi\theta(L+\mu)}$ does not depend on $\kappa$ and $\delta$ and $\xi$ if $\kappa = \xi = \sqrt{\delta}$, we recover the novel scaling rule. $\qquad\square$

A similar result can be derived for a general $X_*$: The final expression is very convoluted and brings marginally negligible added value.

**Lemma C.45.** *For* $f(x) := \frac{x^\top H x}{2}$, *the stationary distribution of RMSpropW is* $(\mathbb{E}[X_\infty], Cov(X_\infty)) = \left(0, \frac{\eta}{2}(H\Sigma^{-\frac{1}{2}} + \theta I_d)^{-1}\right)$.

*Proof.* As $(\nabla f(x))^2 = \mathcal{O}(\eta)$ and $t \to \infty$, we have

$$dX_t = -\Sigma^{-\frac{1}{2}}H X_t dt + \sqrt{\eta}I_d dW_t - \theta X_t dt \tag{165}$$

which implies that

$$X_t = e^{-(\Sigma^{-\frac{1}{2}}H + \gamma I_d)t}\left(X_0 + \sqrt{\eta}\int_0^t e^{(\Sigma^{-\frac{1}{2}}H + \theta I_d)s}dW_s\right). \tag{166}$$

The thesis follows from the martingale property of Brownian motion and the Itô isometry. $\qquad\square$

## C.8 FORMAL DERIVATION - ADAM

In this subsection, we provide our formal derivation of an SDE model for Adam. Let us consider the stochastic process $L_t := (X_t, M_t, V_t) \in \mathbb{R}^d \times \mathbb{R}^d \times \mathbb{R}^d$ defined as the solution of

$$dX_t = -\frac{\sqrt{\iota_2(t)}}{\iota_1(t)} P_t^{-1}(M_t + \textcolor{purple}{\eta\rho_1 (\nabla f(X_t) - M_t)})dt \tag{167}$$

$$dM_t = \rho_1 (\nabla f(X_t) - M_t) dt + \sqrt{\eta}\rho_1 \Sigma^{1/2}(X_t) dW_t \tag{168}$$

$$dV_t = \rho_2 \left((\nabla f(X_t))^2 + \text{diag}(\Sigma(X_t)) - V_t\right) dt, \tag{169}$$

where $\beta_i = 1 - \eta\rho_i$, $\iota_i(t) = 1 - e^{-\rho_i t}$, $\rho_1 = \mathcal{O}(\eta^{-\zeta})$ s.t. $\zeta \in (0,1)$, $\rho_2 = \mathcal{O}(1)$, $t > t_0$, and $P_t = \text{diag} \sqrt{V_t} + \epsilon\sqrt{\iota_2(t)}I_d$.

*Remark C.46.* The terms in purple and in blue are the two differences w.r.t. that of (Malladi et al., 2022) which is reported in Theorem D.5. The first appears because we assume realistic values of $\beta_1$ while the second appears because we allow the gradient size to be non-negligible. For two simple landscapes, Figure 12 compares our SDE and that of Malladi et al. (2022) with Adam: In both cases, the first part of the dynamics is perfectly represented only by our SDE. While the discrepancy between the SDE of (Malladi et al., 2022) and Adam is asymptotically negligible in the convex setting, we observe that in the non-convex case, it converges to a different local minimum than ours and of Adam. Finally, Theorem D.5 is a corollary of ours when $(\nabla f(x))^2 = \mathcal{O}(\eta)$ and $\rho_1 = \mathcal{O}(1)$: It only describes the dynamics where the gradient to noise ratio is vanishing and only for unrealistic values of $\beta_1 = 1 - \eta\rho_1$. In Figure 13, we compare the dynamics of our SDE, that of Malladi et al. (2022), and Adam on an MLP, a CNN, a ResNet, and a Transformer. One can observe that our SDE captures the Adam's dynamics more accurately. Details on these experiments are in Appendix F.

The following theorem guarantees that such a process is a 1-order SDE of the discrete-time algorithm of Adam

$$v_{k+1} = \beta_2 v_k + (1 - \beta_2)(\nabla f_{\gamma_k}(x_k))^2 \tag{170}$$

$$m_{k+1} = \beta_1 m_k + (1 - \beta_1)\nabla f_{\gamma_k}(x_k) \tag{171}$$

$$\hat{m}_k = m_k \left(1 - \beta_1^k\right)^{-1} \tag{172}$$

$$\hat{v}_k = v_k \left(1 - \beta_2^k\right)^{-1} \tag{173}$$

$$x_{k+1} = x_k - \eta\frac{\hat{m}_{k+1}}{\sqrt{\hat{v}_{k+1}} + \epsilon I_d}, \tag{174}$$

with $(x_0, m_0, v_0) \in \mathbb{R}^d \times \mathbb{R}^d \times \mathbb{R}^d$, $\eta \in \mathbb{R}^{>0}$ is the step size, $\beta_i = 1 - \rho_i\eta$ for $\rho_1 = \mathcal{O}(\eta^{-\zeta})$ s.t. $\zeta \in (0,1)$, $\rho_2 = \mathcal{O}(1)$, the mini-batches $\{\gamma_k\}$ are modelled as i.i.d. random variables uniformly distributed on $\{1, \cdots, N\}$, and of size $B \geq 1$.

**Theorem C.47** (Stochastic modified equations). *Let $0 < \eta < 1, T > 0$ and set $N = \lfloor T/\eta \rfloor$. Let $l_k := (x_k, m_k, v_k) \in \mathbb{R}^d \times \mathbb{R}^d \times \mathbb{R}^d, 0 \leq k \leq N$ denote a sequence of Adam iterations defined by Eq. 170. Consider the stochastic process $L_t$ defined in Eq. 167 and fix some test function $g \in G$ and suppose that $g$ and its partial derivatives up to order 6 belong to $G$. Then, under Assumption C.3 $\rho_1 = \mathcal{O}(\eta^{-\zeta})$ s.t. $\zeta \in (0,1)$, while $\rho_2 = \mathcal{O}(1)$, there exists a constant $C > 0$ independent of $\eta$ such that for all $k = 0, 1, \ldots, N$, we have*

$$|\mathbb{E}g(L_{k\eta}) - \mathbb{E}g(l_k)| \leq C\eta.$$

*That is, the SDE 167 is an order 1 weak approximation of the Adam iterations 170 for $t > t_0$.*

*Proof.* The proof is virtually identical to that of Theorem C.16. Therefore, we only report the key steps necessary to conclude the thesis. First of all, we observe that since $\beta_1 = 1 - \eta\rho_1$

$$v_{k+1} - v_k = -\eta\rho_1 \left(v_k - (\nabla f_{\gamma_k}(x_k))^2\right). \tag{175}$$

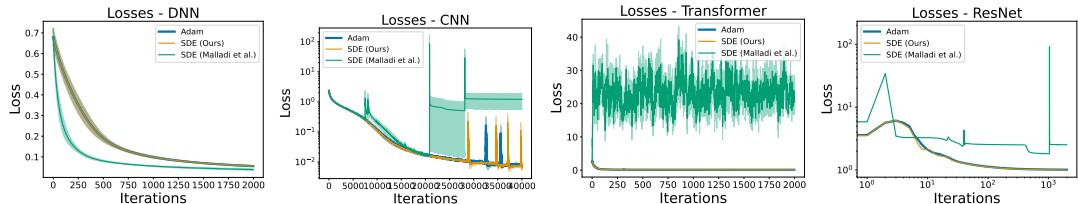

Figure 13: We compare our SDE, that from Malladi et al. (2022), and Adam in terms of $f(x)$: The first is an MLP on the Breast Cancer dataset, the second a CNN on MNIST, the third a Transformer on MNIST, and the last a ResNet on CIFAR-10: One can observe that our SDE matches the algorithm more accurately.

Then,

$$\frac{1}{\sqrt{v_{k+1}}} = \sqrt{\frac{v_k}{v_{k+1}} \frac{1}{v_k}} = \sqrt{\frac{v_{k+1} + \mathcal{O}(\eta)}{v_{k+1}} \frac{1}{v_k}} = \sqrt{1 + \frac{\mathcal{O}(\eta)}{v_{k+1}}} \sqrt{\frac{1}{v_k}} \sim \sqrt{\frac{1}{v_k}}(1 + \mathcal{O}(\eta)). \quad (176)$$

Therefore, we work with the following algorithm as all approximations only carry an additional error of order $\mathcal{O}(\eta^2)$, which we can ignore. Therefore, we have that

$$v_k - v_{k-1} = -\eta\rho_2 \left( v_{k-1} - \left( \nabla f_{\gamma_{k-1}}(x_{k-1}) \right)^2 \right) \quad (177)$$

$$m_{k+1} - m_k = -\eta\rho_1 \left( m_k - \nabla f_{\gamma_k}(x_k) \right) \quad (178)$$

$$\hat{m}_k = m_k \left( 1 - \beta_1^k \right)^{-1} \quad (179)$$

$$\hat{v}_k = v_k \left( 1 - \beta_1^k \right)^{-1} \quad (180)$$

$$x_{k+1} - x_k = -\frac{\eta}{\sqrt{v_k} + \epsilon I_d} \frac{\sqrt{1 - (1 - \eta\rho_2)^k}}{1 - (1 - \eta\rho_1)^{k+1}} (m_k + \eta\rho_1(\nabla f_{\gamma_k}(x_k) - m_k)). \quad (181)$$

Therefore, if $\nabla f_{\gamma_j}(x_j) = \nabla f(x_j) + Z_j(x_j)$ and $\mathbb{E}[Z_j(x_j)] = 0$, and $Cov(Z_j(x_j)) = \Sigma(x_j)$, we have that

1. $\mathbb{E}[v_k - v_{k-1}] = \eta\rho_2 \left[ (\nabla f(x_{k-1}))^2 + \text{diag}(\Sigma(x_k)) - v_{k-1} \right]$;

2. $\mathbb{E}[m_{k+1} - m_k] = \eta\rho_1 \left[ \nabla f(x_k) - m_k \right]$;

3. $\mathbb{E}[x_{k+1} - x_k] = -\frac{\eta}{\sqrt{v_k} + \epsilon I_d} \frac{\sqrt{1 - (1 - \eta\rho_2)^k}}{1 - (1 - \eta\rho_1)^{k+1}} (m_k + \eta\rho_1(\nabla f(x_k) - m_k))$.

Then, we have

1. $\mathbb{E}[(x_{k+1} - x_k)(x_{k+1} - x_k)^\top] = \mathbb{E}[(x_{k+1} - x_k)]\mathbb{E}[(x_{k+1} - x_k)]^\top + \mathcal{O}(\eta^4\rho_1^2)$;

2. $\mathbb{E}[(x_{k+1} - x_k)(m_k - m_{k-1})^\top] = \mathbb{E}[(x_{k+1} - x_k)]\mathbb{E}[(m_k - m_{k-1})]^\top + 0$;

3. $\mathbb{E}[(x_{k+1} - x_k)(v_k - v_{k-1})^\top] = \mathbb{E}[(x_{k+1} - x_k)]\mathbb{E}[(v_k - v_{k-1})]^\top + 0$;

4. $\mathbb{E}[(v_k - v_{k-1})(v_k - v_{k-1})^\top] = \mathbb{E}[(v_k - v_{k-1})]\mathbb{E}[(v_k - v_{k-1})]^\top + \mathcal{O}(\eta^2\rho_2^2)$;

5. $\mathbb{E}[(m_k - m_{k-1})(m_k - m_{k-1})^\top] = \mathbb{E}[(m_k - m_{k-1})]\mathbb{E}[(m_k - m_{k-1})]^\top + \eta^2\rho_1^2\Sigma(x_{k-1})$;

6. $\mathbb{E}[(v_k - v_{k-1})(m_k - m_{k-1})^\top] = \mathbb{E}[(v_k - v_{k-1})]\mathbb{E}[(m_k - m_{k-1})]^\top + \mathcal{O}(\eta^2\rho_1\rho_2)$.

Since in real-world applications, $\rho_1 = \mathcal{O}(\eta^{-\zeta})$ s.t. $\zeta \in (0,1)$, while $\rho_2 = \mathcal{O}(1)$, we have

$$dX_t = -\frac{\sqrt{\iota_2(t)}}{\iota_1(t)} P_t^{-1}(M_t + \eta\rho_1 \left(\nabla f\left(X_t\right) - M_t\right))dt \tag{182}$$

$$dM_t = \rho_1 \left(\nabla f\left(X_t\right) - M_t\right) dt + \sqrt{\eta}\rho_1 \Sigma^{1/2}\left(X_t\right) dW_t \tag{183}$$

$$dV_t = \rho_2 \left(\left(\nabla f(X_t)\right)^2 + \operatorname{diag}\left(\Sigma\left(X_t\right)\right) - V_t\right) dt. \tag{184}$$

where $\beta_i = 1 - \eta\rho_i$, $\iota_i(t) = 1 - e^{-\rho_i t}$, $t > t_0$, and $P_t = \operatorname{diag}\sqrt{V_t} + \epsilon\sqrt{\iota_2(t)}I_d$. $\qquad\square$

*Remark C.48.* See Remark C.35 and Remark C.36 for a discussion on the regularity of the SDE derived in Theorem C.47.

**Corollary C.49.** *Under the assumptions of Theorem C.47 with $\Sigma(x) = \sigma^2 I_d$, $\tilde{\eta} = \kappa\eta$, $\tilde{B} = B\delta$, $\tilde{\rho}_1 = \alpha_1\rho_1$, and $\tilde{\rho}_2 = \alpha_2\rho_2$*

$$dX_t = -\kappa\frac{\sqrt{\iota_2(t)}}{\iota_1(t)} P_t^{-1}(M_t + \eta\alpha_1\rho_1 \left(\nabla f\left(X_t\right) - M_t\right))dt \tag{185}$$

$$dM_t = \alpha_1\rho_1 \left(\nabla f\left(X_t\right) - M_t\right) dt + \sqrt{\eta}\alpha_1\rho_1 \frac{\sigma}{\sqrt{B\delta}} I_d dW_t \tag{186}$$

$$dV_t = \alpha_2\rho_2 \left(\left(\nabla f(X_t)\right)^2 + \frac{\sigma^2}{B\delta}I_d - V_t\right) dt. \tag{187}$$

**Lemma C.50.** *Under the assumptions of Corollary C.49, $f$ is $\mu$-strongly convex, $\operatorname{Tr}(\nabla^2 f(x)) \leq \mathcal{L}_\tau$, and $(\nabla f(x))^2 = \mathcal{O}(\eta)$, the asymptotic dynamics of the iterates of Adam satisfies the classic scaling rule $\kappa = \sqrt{\delta}$ because $\mathbb{E}[f(X_t)] \overset{t\to\infty}{\leq} \frac{\eta\sigma\mathcal{L}_\tau}{4\sqrt{B}}\frac{\kappa}{\sqrt{\delta}}$. To enforce that the speed of $M_t$ and $V_t$ match that of $X_t$, one needs $\tilde{\rho}_i = \kappa\rho_i$, which implies $\tilde{\beta}_i = 1 - \kappa(1 - \beta_i)$.*

*Proof.* First of all, we need to ensure that the relative speeds of $X_t$, $M_t$, and $V_t$ match. Therefore, we select $\alpha_i = \kappa$, which recovers the scaling rules for $\tilde{\beta}_i = 1 - \kappa(1 - \beta_i)$. Then, recalling that $(\nabla f(x))^2 = \mathcal{O}(\eta)$, we have that as $t \to \infty$, $V_t \to \frac{\sigma^2}{B\delta}$, and $M_t \to \nabla f(X_t)$ with high probability. Therefore,

$$dX_t = -\kappa\frac{\sqrt{B\delta}}{\sigma}\nabla f(X_t)dt \tag{188}$$

$$dM_t = \kappa\sqrt{\eta}\rho_1 \frac{\sigma}{\sqrt{B\delta}}dW_t \tag{189}$$

$$dV_t = 0. \tag{190}$$

Therefore, if $H(X_t, V_t) := f(X_t) + \frac{\mathcal{L}_\tau \delta B}{\rho_1^2\sigma^2}\frac{\|M_t\|_2^2}{2}$ and $\xi \in (0,1)$ we have that by Itô's lemma,

$$dH(X_t, V_t) = -(\nabla f(X_t))^\top \left(\kappa\frac{\sqrt{B\delta}}{\sigma}\nabla f(X_t)\right) dt + \left(\frac{\mathcal{L}_\tau \delta B}{\rho_1^2\sigma^2}M_t\right) \kappa\sqrt{\eta}\rho_1 \frac{\sigma}{\sqrt{B\delta}}dW_t \tag{191}$$

$$+ \frac{1}{2}\left(\frac{\mathcal{L}_\tau \delta B}{\rho_1^2\sigma^2}\right) \kappa^2\eta\rho_1^2 \frac{\sigma^2}{B\delta}dt \tag{192}$$

$$= -\left(\kappa\frac{\sqrt{B\delta}}{\sigma}\right) \|\nabla f(X_t)\|_2^2 dt + \text{Noise} + \frac{\kappa^2\eta\mathcal{L}_\tau}{2}dt \tag{193}$$

$$= -\left(\kappa\frac{\sqrt{B\delta}}{\sigma}\right) \left(\xi\|\nabla f(X_t)\|_2^2 + (1-\xi)\|\nabla f(X_t)\|_2^2\right) dt + \text{Noise} + \frac{\kappa^2\eta\mathcal{L}_\tau}{2}dt \tag{194}$$

$$\leq -2\kappa\mu\frac{\sqrt{B\delta}}{\sigma}\xi \left(f(X_t) + \frac{1-\xi}{\mu\xi}\frac{\|\nabla f(X_t)\|_2^2}{2}\right) dt + \text{Noise} + \frac{\kappa^2\eta\mathcal{L}_\tau}{2}dt. \tag{195}$$

Let us now select $\xi$ such that $\frac{1-\xi}{\mu\xi} = \frac{\mathcal{L}_\tau \delta B}{\rho_1^2 \sigma^2}$, this means that $\xi = \frac{\sigma^2 \rho_1^2}{\sigma^2 \rho_1^2 + \mu \mathcal{L}_\tau \sigma B} \in (0,1)$ and $\frac{1}{\xi} = 1 + \mu \frac{\mathcal{L}_\tau \delta B}{\rho_1^2 \sigma^2}$. Since $M_t \to \nabla f(X_t)$, we have that

$$dH(X_t, V_t) \leq -2\kappa\mu \frac{\sqrt{B\delta}}{\sigma} \xi H(X_t, V_t) dt + \frac{\kappa^2 \eta \mathcal{L}_\tau}{2} dt + \text{Noise}. \tag{196}$$

Therefore,

$$\frac{\mathbb{E}[f(X_t)]}{\xi} = \left(1 + \mu \frac{\mathcal{L}_\tau \delta B}{\rho_1^2 \sigma^2}\right) \mathbb{E}[f(X_t)] \leq \mathbb{E}[H(X_t, V_t)] \stackrel{t\to\infty}{\leq} \frac{1}{\xi} \frac{\eta\sigma\mathcal{L}_\tau}{4\mu\sqrt{B}} \frac{\kappa}{\sqrt{\delta}}, \tag{197}$$

which implies that

$$\mathbb{E}[f(X_t)] \stackrel{t\to\infty}{\leq} \frac{\eta\sigma\mathcal{L}_\tau}{4\mu\sqrt{B}} \frac{\kappa}{\sqrt{\delta}}. \tag{198}$$

Analogously,

$$\mathbb{E}[f(X_t) - f(X_*)] \stackrel{t\to\infty}{\leq} \frac{\eta\sigma\mathcal{L}_\tau}{4\mu\sqrt{B}} \frac{\kappa}{\sqrt{\delta}}. \tag{199}$$

which gives the square root scaling rule. $\qquad\square$

**Lemma C.51.** *Under the assumptions of Corollary C.49, $f(x) = \frac{x^\top H x}{2}$ s.t. $H = \operatorname{diag}(\lambda_1, \cdots, \lambda_d)$ and $(\nabla f(x))^2 = \mathcal{O}(\eta)$, the dynamics of Adam implies that $f(X_t) \to \frac{\eta\sigma d}{4\sqrt{B}} \frac{\kappa}{\sqrt{\delta}}$.*

*Proof.* Recalling that $(\nabla f(x))^2 = \mathcal{O}(\eta)$, we have that as $t \to \infty$, $V_t \to \frac{\sigma^2}{B\delta}$, and $M_t \to \lambda X_t$ with high probability. Therefore, in the one-dimensional case

$$dX_t = -\kappa \frac{\sqrt{B\delta}}{\sigma} \lambda X_t dt \tag{200}$$

$$dM_t = \kappa\sqrt{\eta}\rho_1 \frac{\sigma}{\sqrt{B\delta}} dW_t \tag{201}$$

$$dV_t = 0. \tag{202}$$

Therefore, if $H(X_t, V_t) := \frac{\lambda X_t^2}{2} + \frac{\lambda\delta B}{\rho_1^2 \sigma^2} \frac{M_t^2}{2}$,[8] we have that by Itô's lemma,

$$dH(X_t, V_t) = -(\lambda X_t)\left(\kappa \frac{\sqrt{B\delta}}{\sigma} \lambda X_t\right) dt + \left(\frac{\lambda\delta B}{\rho_1^2 \sigma^2} M_t\right) \kappa\sqrt{\eta}\rho_1 \frac{\sigma}{\sqrt{B\delta}} dW_t \tag{203}$$

$$+ \frac{1}{2}\left(\frac{\lambda\delta B}{\rho_1^2 \sigma^2}\right) \kappa^2 \eta \rho_1^2 \frac{\sigma^2}{B\delta} dt \tag{204}$$

$$= -2\kappa\lambda \frac{\sqrt{B\delta}}{\sigma} f(X_t) dt + \frac{\kappa^2 \eta \rho_1^2 \sigma^2}{2B\delta} \frac{\lambda\delta B}{\rho_1^2 \sigma^2} dt + \text{Noise}. \tag{205}$$

$$= -2\kappa\lambda \frac{\sqrt{B\delta}}{\sigma} f(X_t) dt + \frac{\kappa^2 \eta \lambda}{2} dt + \text{Noise}. \tag{206}$$

Once again, since $M_t \to \lambda X_t$, we have that

$$H(X_t, V_t) = \frac{\lambda X_t^2}{2} + \frac{\lambda\delta B}{\rho_1^2 \sigma^2} \frac{M_t^2}{2} \to \frac{\lambda X_t^2}{2} + \lambda \frac{\lambda\delta B}{\rho_1^2 \sigma^2} \frac{\lambda X_t^2}{2} = \left(1 + \lambda \frac{\lambda\delta B}{\rho_1^2 \sigma^2}\right) \frac{\lambda X_t^2}{2} =: Kf(X_t). \tag{207}$$

Therefore,

$$Kd\mathbb{E}[f(X_t)] = -2\kappa\lambda \frac{\sqrt{B\delta}}{\sigma} \mathbb{E}[f(X_t)] dt + \frac{\kappa^2 \eta \lambda}{2} dt, \tag{208}$$

which implies that $\mathbb{E}[f(X_t)] \to \frac{\eta\sigma}{4\sqrt{B}} \frac{\kappa}{\sqrt{\delta}}$, which also gives the square root scaling rule. The generalization to $d$ dimension is analogous and one needs to sum across all the dimensions. $\qquad\square$

---

[8] Inspired by (Barakat and Bianchi, 2021)

**Lemma C.52.** *Let* $f(x) := \frac{x^\top H x}{2}$ *where* $H = \mathrm{diag}(\lambda_1, \ldots, \lambda_d)$. *The stationary distribution of Adam is* $(\mathbb{E}[X_\infty]], Cov(X_\infty)) = \left(0, \frac{\eta}{2} \Sigma^{\frac{1}{2}} H^{-1}\right)$.

*Proof.* The expected value follows immediately from the fact that

$$dX_t = -\Sigma^{-\frac{1}{2}} X_t dt \tag{209}$$

For the covariance, we focus on the one-dimensional case. We define $H(X_t, V_t) := \frac{X_t^2}{2} + \frac{\lambda^2}{2\sigma^2 \rho_1^2} \frac{M_t^2}{2}$. With the same arguments as Lemma C.51, we have

$$d(X_t)^2 = -\frac{\lambda}{\sigma} X_t^2 dt + \frac{\eta}{2} dt + \text{Noise}, \tag{210}$$

which implies that

$$\mathbb{E}[X_t^2] \overset{t \to 0}{\to} \frac{\eta}{2} \frac{\sigma}{\lambda}. \tag{211}$$

The thesis follows by applying the same logic to multiple dimensions. $\square$

## C.9 ADAMW

In this subsection, we derive the SDE of AdamW defined as defined as

$$v_{k+1} = \beta_2 v_k + (1 - \beta_2) \left(\nabla f_{\gamma_k}(x_k)\right)^2 \tag{212}$$

$$m_{k+1} = \beta_1 m_k + (1 - \beta_1) \nabla f_{\gamma_k}(x_k) \tag{213}$$

$$\hat{m}_k = m_k \left(1 - \beta_1^k\right)^{-1} \tag{214}$$

$$\hat{v}_k = v_k \left(1 - \beta_2^k\right)^{-1} \tag{215}$$

$$x_{k+1} = x_k - \eta \frac{\hat{m}_{k+1}}{\sqrt{\hat{v}_{k+1}} + \epsilon I_d} - \eta \theta x_k \tag{216}$$

with $(x_0, m_0, v_0) \in \mathbb{R}^d \times \mathbb{R}^d \times \mathbb{R}^d$, $\eta \in \mathbb{R}^{>0}$ is the step size, $\beta_i = 1 - \rho_i \eta$ for $\rho_1 = \mathcal{O}(\eta^{-\varsigma})$ s.t. $\varsigma \in (0, 1)$, $\rho_2 = \mathcal{O}(1)$, $\theta > 0$, the mini-batches $\{\gamma_k\}$ are modelled as i.i.d. random variables uniformly distributed on $\{1, \cdots, N\}$, and of size $B \geq 1$.

**Theorem C.53.** *Under the same assumptions as Theorem C.47, the SDE of AdamW is*

$$dX_t = -\frac{\sqrt{\iota_2(t)}}{\iota_1(t)} P_t^{-1}(M_t + \eta \rho_1 \left(\nabla f(X_t) - M_t\right)) dt - \theta X_t dt \tag{217}$$

$$dM_t = \rho_1 \left(\nabla f(X_t) - M_t\right) dt + \sqrt{\eta} \rho_1 \Sigma^{1/2}(X_t) dW_t \tag{218}$$

$$dV_t = \rho_2 \left((\nabla f(X_t))^2 + \mathrm{diag}\left(\Sigma(X_t)\right) - V_t\right) dt. \tag{219}$$

*where* $\beta_i = 1 - \eta \rho_i$, $\theta > 0$, $\iota_i(t) = 1 - e^{-\rho_i t}$, $t > t_0$, *and* $P_t = \mathrm{diag} \sqrt{V_t} + \epsilon \sqrt{\iota_2(t)} I_d$.

*Proof.* The proof is the same as the of Theorem C.47 and the only difference is that $\eta \theta x_k$ is approximated with $\theta X_t dt$. $\square$

Figure 4 and Figure 11 validate this result on a variety of architectures and datasets.

*Remark C.54.* See Remark C.35 and Remark C.36 for a discussion on the regularity of the SDE derived in Theorem C.53.

**Corollary C.55.** *Under the assumptions of Theorem C.53 with* $\Sigma(x) = \sigma^2 I_d$, $\tilde{\eta} = \kappa \eta$, $\tilde{B} = B\delta$, $\tilde{\rho}_1 = \alpha_1 \rho_1$, $\tilde{\theta} = \xi \theta$, *and* $\tilde{\rho}_2 = \alpha_2 \rho_2$

$$dX_t = -\kappa \frac{\sqrt{\iota_2(t)}}{\iota_1(t)} P_t^{-1}(M_t + \eta \alpha_1 \rho_1 \left(\nabla f(X_t) - M_t\right)) dt - \kappa \xi \theta X_t dt \tag{220}$$

$$dM_t = \alpha_1 \rho_1 \left(\nabla f(X_t) - M_t\right) dt + \sqrt{\eta} \alpha_1 \rho_1 \frac{\sigma}{\sqrt{B\delta}} I_d dW_t \tag{221}$$

$$dV_t = \alpha_2 \rho_2 \left((\nabla f(X_t))^2 + \frac{\sigma^2}{B\delta} I_d - V_t\right) dt. \tag{222}$$

**Lemma C.56** (Scaling Rule at Convergence). *Under the assumptions of Corollary C.55, $f$ is $\mu$-strongly convex and $L$-smooth, $Tr(\nabla^2 f(x)) \leq \mathcal{L}_\tau$, $X_* = 0$, and $(\nabla f(x))^2 = \mathcal{O}(\eta)$, the asymptotic dynamics of the iterates of AdamW satisfies the novel scaling rule if $\kappa = \sqrt{\delta}$ and $\xi = \kappa$ because*

$$\mathbb{E}[f(X_t) - f(X_*)] \overset{t\to\infty}{\leq} \frac{\eta\mathcal{L}_\tau \sigma L}{2} \frac{\kappa}{2\mu\sqrt{B\delta}L + \sigma\xi\theta(L+\mu)} \tag{223}$$

*By enforcing that the speed of $V_t$ matches that of $X_t$, one needs $\tilde{\rho} = \kappa\rho$, which implies $\tilde{\beta}_i = 1 - \kappa(1 - \beta_i)$.*

*Proof.* The proof is the same as Lemma C.50 where we also use $L$-smoothness as in Lemma C.44. $\square$

A similar result can be derived for a general $X_*$: The final expression is very convoluted and brings marginally negligible added value.

**Lemma C.57.** *For $f(x) := \frac{x^\top H x}{2}$, the stationary distribution of AdamW is $(\mathbb{E}[X_\infty]], Cov(X_\infty)) = \left(0, \frac{\eta}{2}(H\Sigma^{-\frac{1}{2}} + \theta I_d)^{-1}\right)$.*

*Proof.* The proof is the same as Lemma C.52. $\square$

Finally, we prove a generalization of Lemma C.56 to the $L$-smooth case.

**Lemma C.58.** *Let $f$ be $L$-smooth, $\eta_t$ be a learning rate scheduler such that $\lim_{t\to\infty} \frac{\phi_t^2}{\phi_t^1} \overset{t\to\infty}{\to} 0$ and $\phi_t^1 \overset{t\to\infty}{\to} \infty$, where $\phi_t^i = \int_0^t (\eta_s)^i ds$. Then*

$$\mathbb{E}\|\nabla f(X_{\tilde{t}})\|_2^2 \leq \left(f(X_0) - f(X_*) + \frac{\mathcal{L}_\tau \delta B}{\rho_1^2 \sigma^2}\frac{\|M_0\|_2^2}{2} + \frac{\phi_t^2 \eta \kappa^2 \mathcal{L}_\tau}{2}\right) \frac{\sigma}{\kappa\sqrt{\delta B}}\frac{1}{\phi_t^1} \overset{t\to\infty}{\to} 0, \tag{224}$$

*where $\tilde{t}$ is a random time with distribution $\frac{\eta_t}{\phi_t^1}$.*

*Proof.* The proof is the same as Lemma C.24. $\square$

## D   SDEs FROM THE LITERATURE

**Theorem D.1** (Original Malladi's Statement). *Let $\sigma_0 := \sigma\eta$, $\epsilon_0 := \epsilon\eta$, and $c_2 := \frac{1-\beta}{\eta^2}$. Define the state of the SDE as $L_t = (X_t, u_t)$ and the dynamics as*

$$dX_t = -P_t^{-1}\left(\nabla f(X_t) dt + \sigma_0 \Sigma^{1/2}(X_t) dW_t\right) \tag{225}$$

$$du_t = c_2 \left(\mathrm{diag}(\Sigma(X_t)) - u_t\right) dt \tag{226}$$

*where $P_t := \sigma_0 \mathrm{diag}(u_t)^{1/2} + \epsilon_0 I_d$.*

**Theorem D.2** (Informal Statement of Theorem C.2 Malladi et al. (2022)). *Under sufficient regularity conditions and $\nabla f(x) = \mathcal{O}(\sqrt{\eta})$, the following SDE is an order 1 weak approximation of RMSprop:*

$$dX_t = -P_t^{-1}(\nabla f(X_t)dt + \sqrt{\eta}\Sigma(X_t)^{\frac{1}{2}}dW_t) \tag{227}$$

$$dV_t = \rho(\mathrm{diag}(\Sigma(X_t)) - V_t))dt, \tag{228}$$

*where $\beta = 1 - \eta\rho$, $\rho = \mathcal{O}(1)$, and $P_t := \mathrm{diag}(V_t)^{\frac{1}{2}} + \epsilon I_d$.*

**Lemma D.3.** *Theorem D.1 and Theorem D.2 are equivalent.*

*Proof.* It follows applying time rescaling $t := \eta\xi$ and observing that $W_t = W_{\eta\xi} = \sqrt{\eta}W_\xi$. $\square$

**Theorem D.4** (Original Malladi's Statement). *Let $c_1 := (1 - \beta_1)/\eta^2, c_2 := (1 - \beta_2)/\eta^2$ and define $\sigma_0, \epsilon_0$ in Theorem D.1. Let $\iota_1(t) := 1 - \exp(-c_1 t)$ and $\iota_2(t) := 1 - \exp(-c_2 t)$. Define the state of the SDE as $L_t = (X_t, m_t, u_t)$ and the dynamics as*

$$dX_t = -\frac{\sqrt{\iota_2(t)}}{\iota_1(t)} P_t^{-1} m_t dt \tag{229}$$

$$dm_t = c_1 (\nabla f(X_t) - m_t) dt + \sigma_0 c_1 \Sigma^{1/2}(X_t) dW_t, \tag{230}$$
$$du_t = c_2 (\text{diag}(\Sigma(X_t)) - u_t) dt, \tag{231}$$

*where $P_t := \sigma_0 \text{diag}(u_t)^{1/2} + \epsilon_0 \sqrt{\iota_2(t)} I_d$.*

**Theorem D.5** (Informal Statement of Theorem D.2 Malladi et al. (2022)). *Under sufficient regularity conditions and $\nabla f(x) = \mathcal{O}(\sqrt{\eta})$, the following SDE is an order $1$ weak approximation of Adam:*

$$dX_t = -\frac{\sqrt{\iota_2(t)}}{\iota_1(t)} P_t^{-1} M_t dt \tag{232}$$

$$dM_t = \rho_1 (\nabla f(X_t) - M_t) dt + \sqrt{\eta} \rho_1 \Sigma^{1/2}(X_t) dW_t \tag{233}$$
$$dV_t = \rho_2 (\text{diag}(\Sigma(X_t)) - V_t) dt. \tag{234}$$

*where $\beta_i = 1 - \eta \rho_i$, $\iota_i(t) = 1 - e^{-\rho_i t}$, $\rho_i = \mathcal{O}(1)$, and $P_t = \text{diag}\sqrt{V_t} + \epsilon\sqrt{\iota_2(t)} I_d$.*

**Lemma D.6.** *Theorem D.4 and Theorem D.5 are equivalent.*

*Proof.* It follows applying time rescaling $t := \eta \xi$ and observing that $W_t = W_{\eta \xi} = \sqrt{\eta} W_\xi$. $\qquad \square$

## E  SDE CANNOT BE DERIVED NOR USED NAIVELY

In this section, we provide a gentle introduction to the meaning of deriving an SDE model for an optimizer and discuss how SDEs have been used to derive scaling rules. To aid the intuition of the reader, we informally derive an SDE for SGD with learning rate $\eta$, mini-batches $\gamma_B$ of size $B$, and starting point $x_0 = x$, which we dub SGD$^{(\eta,B)}$. The iterates are given by:

$$x_{k+1} = x_k - \eta \nabla f_{\gamma_k^B}(x_k) \tag{235}$$

which for $U_k := \sqrt{\eta}(\nabla f(x_k) - \nabla f_{\gamma_k^B}(x_k))$, we rewrite as

$$x_k - \eta \nabla f(x_k) + \sqrt{\eta} U_k, \tag{236}$$

where $\mathbb{E}[U_k] = 0$ and $Cov(U_k) = \frac{\eta}{B}\Sigma(x_k) = \frac{\eta}{B}\frac{1}{n}\sum_{i=0}^n (\nabla f(x_k) - \nabla f_i(x_k))(\nabla f(x) - \nabla f_i(x_k))^\top$. If we now consider the SDE

$$dX_t = -\nabla f(X_t) dt + \sqrt{\frac{\eta}{B}}\Sigma(X_t)^{\frac{1}{2}} dW_t, \tag{237}$$

its Euler-Maruyama discretization with pace $\Delta t = \eta$ and $Z_k \sim \mathcal{N}(0, I_d)$ is

$$X_{k+1} = X_k - \eta \nabla f(X_k) + \sqrt{\eta}\sqrt{\frac{\eta}{B}}\Sigma(X_t)^{\frac{1}{2}} Z_k. \tag{238}$$

Since the Eq. 235 and Eq. 238 share the first two moments, it is reasonable that by identifying $t = k\eta$, the SDE in Eq. 237 is a good model to describe the iterates of SGD in Eq. 235.

Informally, we need a "good model", which is an SDE that is close to the real optimizer. This is formalized in the following definition which comes from the field of numerical analysis of SDEs (see Mil'shtein (1986)) and bounds the disparity between the the discrete and the continuous process.

**Definition E.1** (Weak Approximation). A continuous-time stochastic process $\{X_t\}_{t\in[0,T]}$ is an order $\alpha$ weak approximation (or $\alpha$-order SDE) of a discrete stochastic process $\{x_k\}_{k=0}^{\lfloor T/\eta \rfloor}$ if for every polynomial growth function $g$, there exists a positive constant $C$, independent of the stepsize $\eta$, such that $\max_{k=0,\ldots,\lfloor T/\eta \rfloor} |\mathbb{E}g(x_k) - \mathbb{E}g(X_{k\eta})| \le C\eta^\alpha$.

To see if an SDE satisfies such a definition, one has to check that for $\bar{\Delta} = x_1 - x$ and $\Delta = X_\eta - x$,

1. $\left|\mathbb{E}\Delta_i - \mathbb{E}\bar{\Delta}_i\right| = \mathcal{O}(\eta^2), \quad \forall i = 1, \ldots, d;$

2. $\left|\mathbb{E}\Delta_i\Delta_j - \mathbb{E}\bar{\Delta}_i\bar{\Delta}_j\right| = \mathcal{O}(\eta^2), \quad \forall i, j = 1, \ldots, d.$

**Example:** Let us prove that the SDE in Eq. 237 is a valid approximation of SGD$^{(\eta,B)}$: The first condition is easily verified. Coming to the second condition we have that

1. $\mathbb{E}\Delta_i\Delta_j = \eta^2\partial_i f(x)\partial_j f(x) + \frac{\eta^2}{B}\Sigma(x);$

2. $\mathbb{E}\bar{\Delta}_i\bar{\Delta}_j = \eta^2\partial_i f(x)\partial_j f(x) + \frac{\eta^2}{B}\Sigma(x) + \mathcal{O}(\eta^3);$

whose difference is of order $\eta^3$ and thus satisfies the condition. However, we observe that if the scale of the noise is too small w.r.t. $\eta$, i.e. $\Sigma(x) = \mathcal{O}(\eta^\alpha)$ for $\alpha \geq 0$, then the **simplest** SDE model describing SGD$^{(\eta,B)}$ is the ODE $dX_t = -\nabla f(X_t)dt$ as in that case

1. $\mathbb{E}\Delta_i\Delta_j = \eta^2\partial_i f(x)\partial_j f(x) + \mathcal{O}(\eta^{2+\alpha});$

2. $\mathbb{E}\bar{\Delta}_i\bar{\Delta}_j = \eta^2\partial_i f(x)\partial_j f(x) + \mathcal{O}(\eta^2),$

whose difference is also of order $\eta^2$. Much differently, if $\Sigma(x) = \mathcal{O}(\eta^{-\alpha})$ for $\alpha > 0$, the simplest model is the SDE in Eq. 237. We highlight that *simplest* does not mean *best*: The SDE is more accurate than the ODE even in a regime with low noise, but this observation serves as a provocation. One has to pay attention when deriving SDEs: Some models are more realistic than others.

Let us dig deeper into this thought as we derive **two** SDEs for SGD with learning rate $\tilde{\eta} := \kappa\eta$ and batch size $\tilde{B} := \delta B$ for $\kappa > 1$ and $\delta > 1$, which we dub SGD$^{(\tilde{\eta},\tilde{B})}$. The first is derived considering that the learning rate is $\tilde{\eta}$ and carries an error of order $\mathcal{O}(\tilde{\eta})$ w.r.t. SGD$^{(\tilde{\eta},\tilde{B})}$

$$dX_t = -\nabla f(X_t)dt + \sqrt{\frac{\tilde{\eta}}{\tilde{B}}}\Sigma(X_t)^{\frac{1}{2}}dW_t = -\nabla f(X_t)dt + \sqrt{\frac{\eta\kappa}{B\delta}}\Sigma(X_t)^{\frac{1}{2}}dW_t. \tag{239}$$

The second one instead is derived considering $\eta$ as the learning rate and $\kappa$ as a constant "scheduler". Consistently with (Li et al., 2017), the SDE which carries an error of order $\mathcal{O}(\eta)$ w.r.t. SGD$^{(\tilde{\eta},\tilde{B})}$ is

$$dX_t = -\kappa\nabla f(X_t)dt + \kappa\sqrt{\frac{\eta}{B\delta}}\Sigma(X_t)^{\frac{1}{2}}dW_t. \tag{240}$$

While they both are valid models, there are three reasons why one should prefer the latter:

1. It fully reflects the fact that a larger learning rate results in a faster and noisier dynamics;

2. It has intrinsically less error than the other;

3. It is consistent with the optimizer in that there is no combination of $\kappa$ and $\delta$ that can ever leave the dynamics unchanged.

### E.1 DERIVING SCALING RULES

Jastrzebski et al. (2018) observed that only the ratio between $\eta$ and $B$ matters in determining the dynamics of Eq. 238. Therefore, they argue that for $\kappa = \delta$ the SDE for SGD$^{(\kappa\eta,\delta B)}$ coincides with that of SGD$^{(\eta,B)}$ and that this implies that the path properties of the optimizers are the same. On the contrary, the path of SGD$^{(\eta,B)}$ strongly depends on the hyperparameters: The speed and volatility of the dynamics are driven by $\eta$, and no choice of $B$ can undo this. We remind the reader that the goal of these rules is not to keep the dynamics of the optimizers unaltered, but rather to give a practical way to change a hyperparameter, e.g. $\eta$, and have a principled way to adjust the others, e.g. $B$, such that the performance of the optimizer is preserved. Therefore, we propose deriving scaling rules as we preserve certain relevant quantities of the dynamics such as the convergence bound on the expected loss. To show this quantitatively, we use this rationale to derive the scaling rule of SGD as we aim at preserving the asymptotic loss level.

**Lemma E.2.** *If $f$ is a $\mu$ strongly convex function, $Tr(\nabla^2 f(x)) \leq \mathcal{L}_\tau$ and $\Sigma(x) = \sigma^2 I_d$, then:*

1. *Under the dynamics of Eq. 237 we have:*

$$\mathbb{E}[f(X_t) - f(X_*)] \leq (f(X_0) - f(X_*))e^{-2\mu t} + \frac{\eta}{2}\frac{\mathcal{L}_\tau \sigma^2}{2\mu B}\left(1 - e^{-2\mu t}\right); \qquad (241)$$

2. *Under the dynamics of Eq. 239 we have:*

$$\mathbb{E}[f(X_t) - f(X_*)] \leq (f(X_0) - f(X_*))e^{-2\mu t} + \frac{\eta}{2}\frac{\mathcal{L}_\tau \sigma^2}{2\mu B}\frac{\kappa}{\delta}\left(1 - e^{-2\mu t}\right); \qquad (242)$$

3. *Under the dynamics of Eq. 240 we have:*

$$\mathbb{E}[f(X_t) - f(X_*)] \leq (f(X_0) - f(X_*))e^{-2\mu\kappa t} + \frac{\eta}{2}\frac{\mathcal{L}_\tau \sigma^2}{2\mu B}\frac{\kappa}{\delta}\left(1 - e^{-2\mu\kappa t}\right). \qquad (243)$$

The first bound implies that the asymptotic limit of the expected loss for $\text{SGD}^{(\eta,B)}$ is $\frac{\eta}{2}\frac{\mathcal{L}_\tau \sigma^2}{2\mu B}$. The last two bounds predict that the asymptotic loss level for $\text{SGD}^{(\tilde{\eta},\tilde{B})}$ is $\frac{\eta}{2}\frac{\mathcal{L}_\tau \sigma^2}{2\mu B}\frac{\kappa}{\delta}$. Since the objective of the scaling rule is to find $\kappa$ and $\delta$ such that $\text{SGD}^{(\tilde{\eta},\tilde{B})}$ achieves the same loss level as $\text{SGD}^{(\eta,B)}$, we recover the linear scaling rule setting $\kappa = \delta$. However, only the last bound can correctly capture the fact that the dynamics of $\text{SGD}^{(\tilde{\eta},\tilde{B})}$ is $\kappa$ times faster than that of $\text{SGD}^{(\eta,B)}$.

We thus conclude that:

1. Eq. 240 is a better model for $\text{SGD}^{(\tilde{\eta},\tilde{B})}$ as it represents the dynamics more accurately;
2. Maintaining the shape of the SDE does not preserve the path properties of the optimizer;
3. Deriving a scaling rule uniquely from the SDE might lead to the wrong conclusions in the general case.

*Remark* E.3. In Malladi et al. (2022), Theorem 5.3 proposes a formal derivation of a scaling rule for RMSprop. Following the perspective of Jastrzebski et al. (2018), the authors suggest that a scaling rule that leaves their SDE unchanged would also preserve the dynamics of the RMSprop iterates. We note, however, that an SDE is defined not only by the equation that governs the dynamics but also by its initial condition (see Karatzas and Shreve (2014), Section 5). Although the scaling rule in question leaves the differential equation unchanged, it modifies the initial condition of the process $u_t$, and hence, the overall SDE is altered. This observation suggests that the claim and proof in Malladi et al. (2022) may need to be revisited.

Furthermore, the rule appears to be valid only in the vicinity of convergence, as the corresponding SDE is applicable primarily in that regime. Additionally, Lemma E.2 provides concrete evidence that maintaining the form of the SDE does not necessarily imply that the path properties of the optimizer are preserved. We offer these clarifications with the hope of contributing constructively to the discussion.

# F EXPERIMENTS

In this section, we provide the modeling choices and instructions to replicate our experiments. The code is implemented in Python 3 (Van Rossum and Drake, 2009) mainly using Numpy (Harris et al., 2020), scikit-learn (Pedregosa et al., 2011), and JAX (Bradbury et al., 2018).

## F.1 SIGNSGD: SDE VALIDATION (FIGURE 1)

In this subsection, we describe the experiments we run to produce Figure 1: The loss dynamics of SignSGD and that of our SDE match on average.

**DNN on Breast Cancer Dataset (Dua and Graff, 2017)** This paragraph refers to the *left* of Figure 1. The DNN has 10 dense layers with 20 neurons each activated with a ReLu. We minimize the binary cross-entropy loss. We run SignSGD for 50000 epochs as we calculate the full gradient and inject it with Gaussian noise $Z \sim \mathcal{N}(0, \sigma^2 I_d)$ where $\sigma = 1$. The learning rate is $\eta = 0.001$. Similarly, we integrate the SignSGD SDE (Eq. 7) with Euler-Maruyama (Algorithm 1) with $\Delta t = \eta$. Results are averaged over 3 runs and the shaded areas are the average $\pm$ the standard deviation.

**CNN on MNIST (Deng, 2012)**    This paragraph refers to the *center-left* of Figure 1. The CNN has a $(3, 3, 32)$ convolutional layer with stride 1, followed by a ReLu activation, a $(2, 2)$ max pool layer with stride $(2, 2)$, a $(3, 3, 32)$ convolutional layer with stride 1, a ReLu activation, a $(2, 2)$ max pool layer with stride $(2, 2)$. Then the activations are flattened and passed through a dense layer that compresses them into 128 dimensions, a final ReLu activation, and a final dense layer into the output dimension 10. The output finally goes through a softmax as we minimize the cross-entropy loss. We run SignSGD for 60000 epochs as we calculate the full gradient and inject it with Gaussian noise $Z \sim \mathcal{N}(0, \sigma^2 I_d)$ where $\sigma = 0.4$. The learning rate is $\eta = 0.001$. Similarly, we integrate the SignSGD SDE (Eq. 7) with Euler-Maruyama (Algorithm 1) with $\Delta t = \eta$. Results are averaged over 3 run and the shaded areas are the average $\pm$ the standard deviation.

**Transformer on MNIST**    This paragraph refers to the *center-right* of Figure 1. The Architecture is a scaled-down version of (Dosovitskiy et al., 2021), where the hyperparameters are *patch size*=28, *out features*=10, *width*=48, *depth*=3, *num heads*=6, and *dim ffn*=192. We minimize the cross-entropy loss as we run SignSGD for 5000 epochs as we calculate the full gradient and inject it with Gaussian noise $Z \sim \mathcal{N}(0, \sigma^2 I_d)$ where $\sigma = 1$. The learning rate is $\eta = 0.001$. Similarly, we integrate the SignSGD SDE (Eq. 7) with Euler-Maruyama (Algorithm 1) with $\Delta t = \eta$. Results are averaged over 3 runs and the shaded areas are the average $\pm$ the standard deviation.

**ResNet on CIFAR-10 (Krizhevsky et al., 2009)**    This paragraph refers to the *right* of Figure 1. The ResNet has a $(3, 3, 32)$ convolutional layer with stride 1, followed by a ReLu activation, a second $(3, 3, 32)$ convolutional layer with stride 1, followed by a residual connection from the first convolutional layer, then a $(2, 2)$ max pool layer with stride $(2, 2)$. Then the activations are flattened and passed through a dense layer that compresses them into 128 dimensions, a final ReLu activation, and a final dense layer into the output dimension 10. The output finally goes through a softmax as we minimize the cross-entropy loss. We run SignSGD for 5000 epochs as we calculate the full gradient and inject it with Gaussian noise $Z \sim \mathcal{N}(0, \sigma^2 I_d)$ where $\sigma = 1$. The learning rate is $\eta = 0.001$. Similarly, we integrate the SignSGD SDE (Eq. 7) with Euler-Maruyama (Algorithm 1) with $\Delta t = \eta$. Results are averaged over 3 runs and the shaded areas are the average $\pm$ the standard deviation.

F.2    SIGNSGD: INSIGHTS VALIDATION (FIGURE 2)

In this subsection, we describe the experiments we run to produce Figure 2: We successfully validate them all.

**Phases: Lemma 3.4 and Lemma 3.5**    In this paragraph, we describe how we validated the existence of the phases of SignSGD as predicted in Lemma 3.4 and Lemma 3.5. To produce the *left* of Figure 2), we simulated the *full SDE* (Eq. 24) and the one describing Phase 3 (Eq. 5). The optimized function is $f(x) = \frac{x^\top H x}{2}$ for $H = \mathrm{diag}(1, 2)$, $x_0$ drawn (and fixed for all runs) from a normal distribution $\mathcal{N}(0, 0.01)$, $\eta = 0.001$, and $\Sigma = \sigma^2 I_d$ where $\sigma = 0.1$. We integrate the SDEs with Euler-Maruyama (Algorithm 1) with $\Delta t = \eta$ and for 3000 iterations. Results are averaged over 500 runs and the shaded areas are the average $\pm$ the standard deviation. Clearly, the two SDEs share the same dynamics.

To produce the *center-left* of Figure 2, we repeat the above as $x_0$ drawn (and fixed for all runs) from a normal distribution $\mathcal{N}(0, 1)$. Then, we plot the average loss values together with the theoretical prediction of Phase 1 and Phase 3: They perfectly overlap.

**Stationary distribution: Lemma 3.7**    In this paragraph, we describe how we validated the convergence behavior predicted in Lemma 3.7. To produce the *center-right* of Figure 2), we run SignSGD on $f(x) = \frac{x^\top H x}{2}$ for $H = \mathrm{diag}(1, 2)$, $x_0 = (0.001, 0.001)$, $\eta = 0.001$ and $\Sigma = \sigma^2 I_d$ where $\sigma = 0.1$. We run this for 5000 times and report the evolution of the moments. Then, we add lines representing the theoretical predictions derived in Lemma 3.7: They match.

**Schedulers: Lemma 3.9**    In this paragraph, we describe how we validated the convergence behavior predicted in Lemma 3.9. To produce the *right* of Figure 2, we run SignSGD on $f(x) = \frac{x^\top H x}{2}$ for $H = \mathrm{diag}(1, 2)$, $x_0 = (0.01, 0.01)$, $\eta = 0.01$ and $\Sigma = \sigma^2 I_d$ where $\sigma = 0.1$. We used the scheduler

$\eta_t^\vartheta = \frac{1}{(t+1)^\vartheta}$ for $\vartheta \in \{0.1, 0.5, 1.5\}$. For the first two choices of $\vartheta$, $\eta_t^\vartheta$ satisfies our sufficient condition for the convergence of SignSGD: In the figure, we observe that indeed SignSGD converges to 0 with the same speed as the one predicted in the Lemma. For $\vartheta = 1.5$, we observe that SignSGD does not converge following the theoretical curve because it does not satisfy our sufficient condition. Results are averaged over 500 runs.

### F.3 RMSPROP: SDE VALIDATION (FIGURE 9 AND FIGURE 10)

In this subsection, we describe the experiments we run to produce Figure 9 and Figure 10: The dynamics of our SDE matches that of RMSprop more accurately than the SDE derived in (Malladi et al., 2022).

**Quadratic convex function**    This paragraph refers to the *left* and *center-left* of Figure 9. We optimize the function $f(x) = \frac{x^\top H x}{2}$ where $H = \text{diag}(10, 2)$. We run RMSprop for 2000 epochs as we calculate the full gradient and inject it with Gaussian noise $Z \sim \mathcal{N}(0, \sigma^2 I_d)$ where $\sigma = 0.1$. The learning rate is $\eta = 0.01$, $\beta = 0.99$. Similarly, we integrate our RMSprop SDE (Eq. 129) and that of Malladi (Eq. 227) with Euler-Maruyama (Algorithm 1) with $\Delta t = \eta$. Results are averaged over 500 runs and the shaded areas are the average $\pm$ the standard deviation: Our SDE matches RMSprop more accurately.

**Embedded saddle**    This paragraph refers to the *center-right* and *right* of Figure 9. We optimize the function $f(x) = \frac{x^\top H x}{2} + \frac{1}{4}\lambda \sum_{i=1}^2 x_i^4 - \frac{\xi}{3} \sum_{i=1}^2 x_i^3$ where $H = \text{diag}(-1, 2)$, $\lambda = 1$, and $\xi = 0.1$. We run RMSprop for 1600 epochs as we calculate the full gradient and inject it with Gaussian noise $Z \sim \mathcal{N}(0, \sigma^2 I_d)$ where $\sigma = 0.01$. The learning rate is $\eta = 0.01$, $\beta = 0.99$. Similarly, we integrate our RMSprop SDE (Eq. 129) and that of Malladi (Eq. 227) with Euler-Maruyama (Algorithm 1) with $\Delta t = \eta$. Results are averaged over 500 runs and the shaded areas are the average $\pm$ the standard deviation: Our SDE matches RMSprop more accurately.

**DNN on Breast Cancer Dataset**    This paragraph refers to the *left* of Figure 10. The architecture and loss are the same as used above for SignSGD. We run RMSprop for 2000 epochs as we calculate the full gradient and inject it with Gaussian noise $Z \sim \mathcal{N}(0, \sigma^2 I_d)$ where $\sigma = 10^{-2}$. The learning rate is $\eta = 10^{-4}$, $\beta = 0.9995$. Similarly, we integrate our RMSprop SDE (Eq. 129) and that of Malladi (Eq. 227) with Euler-Maruyama (Algorithm 1) with $\Delta t = \eta$. Results are averaged over 3 runs and the shaded areas are the average $\pm$ the standard deviation: Our SDE matches RMSprop more accurately.

**CNN on MNIST**    This paragraph refers to the *center-left* of Figure 10. The architecture and loss are the same as used above for SignSGD. We run RMSprop for 100000 epochs as we calculate the full gradient and inject it with Gaussian noise $Z \sim \mathcal{N}(0, \sigma^2 I_d)$ where $\sigma = 10^{-2}$. The learning rate is $\eta = 10^{-4}$, $\beta = 0.999$. Similarly, we integrate our RMSprop SDE (Eq. 129) and that of Malladi (Eq. 227) with Euler-Maruyama (Algorithm 1) with $\Delta t = \eta$. Results are averaged over 3 run and the shaded areas are the average $\pm$ the standard deviation: Our SDE matches RMSprop more accurately.

**Transformer on MNIST**    This paragraph refers to the *center-right* of Figure 10. The architecture and loss are the same as used above for SignSGD. We run RMSprop for 2000 epochs as we calculate the full gradient and inject it with Gaussian noise $Z \sim \mathcal{N}(0, \sigma^2 I_d)$ where $\sigma = 10^{-2}$. The learning rate is $\eta = 10^{-3}$, $\beta = 0.995$. Similarly, we integrate our RMSprop SDE (Eq. 129) and that of Malladi (Eq. 227) with Euler-Maruyama (Algorithm 1) with $\Delta t = \eta$. Results are averaged over 3 runs and the shaded areas are the average $\pm$ the standard deviation: Our SDE matches RMSprop more accurately.

**ResNet on CIFAR-10**    This paragraph refers to the *right* of Figure 10. The architecture and loss are the same as used above for SignSGD. We run RMSprop for 500 epochs as we calculate the full gradient and inject it with Gaussian noise $Z \sim \mathcal{N}(0, \sigma^2 I_d)$ where $\sigma = 10^{-4}$. The learning rate is $\eta = 10^{-4}$, $\beta = 0.9999$. Similarly, we integrate our RMSprop SDE (Eq. 129) and that of Malladi (Eq. 227) with Euler-Maruyama (Algorithm 1) with $\Delta t = \eta$. Results are averaged over 3 runs and the shaded areas are the average $\pm$ the standard deviation: Our SDE matches RMSprop more accurately.

## F.4 ADAM: SDE VALIDATION (FIGURE 12 AND FIGURE 13)

In this subsection, we describe the experiments we run to produce Figure 13 and Figure 12: The dynamics of our SDE matches that of Adam more accurately than that derived in (Malladi et al., 2022).

**Quadratic convex function** This paragraph refers to the *left* and *center-left* of Figure 12. We optimize the function $f(x) = \frac{x^\top H x}{2}$ where $H = \text{diag}(10, 2)$. We run Adam for 50000 epochs as we calculate the full gradient and inject it with Gaussian noise $Z \sim \mathcal{N}(0, \sigma^2 I_d)$ where $\sigma = 0.01$. The learning rate is $\eta = 0.001$, $\beta_1 = 0.9$, and $\beta_2 = 0.999$. Similarly, we integrate our Adam SDE (Eq. 167) and that of Malladi (Eq. 232) with Euler-Maruyama (Algorithm 1) with $\Delta t = \eta$. Results are averaged over 500 runs and the shaded areas are the average $\pm$ the standard deviation: Our SDE matches Adam more accurately.

**Embedded saddle** This paragraph refers to the *center-right* and *right* of Figure 12. We optimize the function $f(x) = \frac{x^\top H x}{2} + \frac{1}{4}\lambda \sum_{i=1}^2 x_i^4 - \frac{\xi}{3} \sum_{i=1}^2 x_i^3$ where $H = \text{diag}(-1, 2)$, $\lambda = 1$, and $\xi = 0.1$. We run Adam as we calculate the full gradient and inject it with Gaussian noise $Z \sim \mathcal{N}(0, \sigma^2 I_d)$ where $\sigma = 0.1$. The learning rate is $\eta = 0.001$, $\beta_1 = 0.9$, and $\beta_2 = 0.999$. Similarly, we integrate our Adam SDE (Eq. 167) and that of Malladi (Eq. 232) with Euler-Maruyama (Algorithm 1) with $\Delta t = \eta$. Results are averaged over 500 runs and the shaded areas are the average $\pm$ the standard deviation: Our SDE matches Adam more accurately.

**DNN on Breast Cancer Dataset** This paragraph refers to the *left* of Figure 13. The architecture and loss are the same as used above for SignSGD. We run Adam for 2000 epochs as we calculate the full gradient and inject it with Gaussian noise $Z \sim \mathcal{N}(0, \sigma^2 I_d)$ where $\sigma = 10^{-2}$. The learning rate is $\eta = 10^{-4}$, $\beta_1 = 0.99$, and $\beta_2 = 0.999$. Similarly, we integrate our Adam SDE (Eq. 167) and that of Malladi (Eq. 232) with Euler-Maruyama (Algorithm 1) with $\Delta t = \eta$. Results are averaged over 3 runs and the shaded areas are the average $\pm$ the standard deviation: Our SDE matches Adam more accurately.

**CNN on MNIST** This paragraph refers to the *center-left* of Figure 13. The architecture and loss are the same as used above for SignSGD. We run Adam for 40000 epochs as we calculate the full gradient and inject it with Gaussian noise $Z \sim \mathcal{N}(0, \sigma^2 I_d)$ where $\sigma = 10^{-2}$. The learning rate is $\eta = 10^{-3}$, $\beta_1 = 0.99$, and $\beta_2 = 0.999$. Similarly, we integrate our Adam SDE (Eq. 167) and that of Malladi (Eq. 232) with Euler-Maruyama (Algorithm 1) with $\Delta t = \eta$. Results are averaged over 3 runs and the shaded areas are the average $\pm$ the standard deviation: Our SDE matches Adam more accurately.

**Transformer on MNIST** This paragraph refers to the *center-right* of Figure 13. The architecture and loss are the same as used above for SignSGD. We run Adam for 2000 epochs as we calculate the full gradient and inject it with Gaussian noise $Z \sim \mathcal{N}(0, \sigma^2 I_d)$ where $\sigma = 10^{-2}$. The learning rate is $\eta = 10^{-2}$, $\beta_1 = 0.9$, and $\beta_2 = 0.99$. Similarly, we integrate our Adam SDE (Eq. 167) and that of Malladi (Eq. 232) with Euler-Maruyama (Algorithm 1) with $\Delta t = \eta$. Results are averaged over 3 runs and the shaded areas are the average $\pm$ the standard deviation: Our SDE matches Adam more accurately.

**ResNet on CIFAR-10** This paragraph refers to the *right* of Figure 13. The architecture and loss are the same as used above for SignSGD. We run Adam for 2000 epochs as we calculate the full gradient and inject it with Gaussian noise $Z \sim \mathcal{N}(0, \sigma^2 I_d)$ where $\sigma = 10^{-5}$. The learning rate is $\eta = 10^{-5}$, $\beta_1 = 0.99$, and $\beta_2 = 0.9999$. Similarly, we integrate our Adam SDE (Eq. 167) and that of Malladi (Eq. 232) with Euler-Maruyama (Algorithm 1) with $\Delta t = \eta$. Results are averaged over 3 runs and the shaded areas are the average $\pm$ the standard deviation: Our SDE matches Adam more accurately.

## F.5 RMSPROPW & ADAMW: SDE VALIDATION (FIGURE 3, FIGURE 4)

The settings are exactly the same as those for RMSprop and Adam. The regularization parameter used is always $\theta = 0.01$. We observe that our SDEs match the respective algorithm with a good agreement.

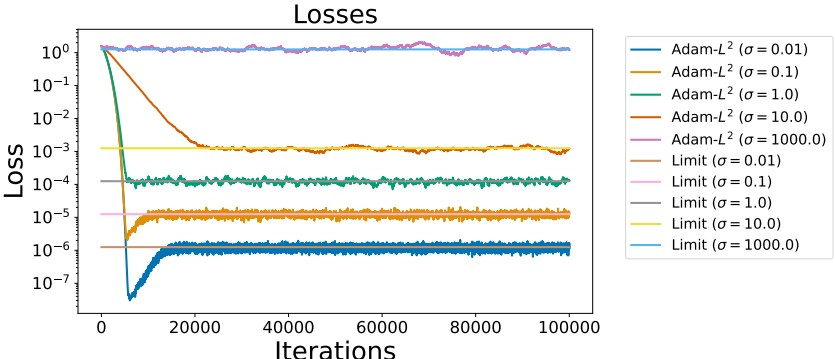

Figure 14: Empirical validation of the bounds for Adam on an $L^2$-regularized loss $f(x) + \frac{\theta \|x\|_2^2}{2}$. For several levels of noise $\sigma$, we find that our theoretical predictions match the experimental results: The loss levels scale linearly in $\sigma$.

## F.6 RMSPROPW & ADAMW: INSIGHTS VALIDATION (FIGURE 5)

In this subsection, we describe the experiments we run to produce Figure 5: The theoretically predicted asymptotic loss value and moments of RMSpropW and AdamW match those empirically found.

**Asymptotic loss & scaling rule of AdamW**  This paragraph refers to the *left* of Figure 5. We optimize the function $f(x) = \frac{x^\top H x}{2}$ where $H = \text{diag}(1,3)$. We run AdamW for 20000 epochs as we calculate the full gradient and inject it with Gaussian noise $Z \sim \mathcal{N}(0, \sigma^2 I_d)$ where $\sigma = 1$. The learning rate is $\eta = 0.001$, $\beta_1 = 0.9$, and $\beta_2 = 0.999$. Experiments are run for both $\theta = 1$ and $\theta = 4$. The rescaled versions of the algorithms *AdamW R* follow the novel scaling rule with $\kappa = 2$. *AdamW NR* follows the scaling rule but not for $\theta$ which is left unchanged. We plot the evolution of the loss values with the theoretical predictions of Lemma C.50: Results are averaged over 500 runs.

**Asymptotic loss & scaling rule of RMSpropW**  This paragraph refers to the *center-left* of Figure 5: The only difference with the previous paragraph is that we use RMSpropW with $\beta = 0.999$.

**AdamW: the role of the $\beta$s**  This paragraph refers to the *center-right* of Figure 5. We optimize the function $f(x) = \frac{x^\top H x}{2} + \frac{1}{4}\lambda \sum_{i=1}^2 x_i^4 - \frac{\xi}{3} \sum_{i=1}^2 x_i^3$ where $H = \text{diag}(-1,2)$, $\lambda = 1$, and $\xi = 0.1$. We run AdamW as we calculate the full gradient and inject it with Gaussian noise $Z \sim \mathcal{N}(0, \sigma^2 I_d)$ where $\sigma = 0.1$. The learning rate is $\eta = 0.001$, $\theta = 0.1$, $\beta_1 \in \{0.99, 0.999\}$, and $\beta_2 \in \{0.992, 0.996, 0.998\}$: Clearly, three combinations go into a minimum and three go into the other. For each minimum, the three optimizers converge to the same asymptotic loss value independently on the values of $\beta_1$ and $\beta_2$. We argue that $\beta_1$, and $\beta_2$ select the basin and the speed of convergence, not the asymptotic loss value: This is consistent with Lemma 3.13.

**Stationary distribution**  This paragraph refers to the *right* of Figure 5. We optimize the function $f(x) = \frac{x^\top H x}{2}$ where $H = \text{diag}(1,3)$. We run Adam for 20000 epochs as we calculate the full gradient and inject it with Gaussian noise $Z \sim \mathcal{N}(0, \sigma^2 I_d)$ where $\sigma = 0.01$. The learning rate is $\eta = 0.001$, $\theta = 4$, $\beta = 0.999$, $\beta_1 = 0.9$, and $\beta_2 = 0.999$. We plot the evolution of the average variances with the theoretical predictions of Lemma C.45 and Lemma 3.14: Results are averaged over 100 runs.

## F.7 EFFECT OF NOISE - VALIDATION (FIGURE 6 AND FIGURE 14)

In this subsection, we describe the experiments run to produce Figure 6 and Figure 14: All bounds on the asymptotic expected loss value for SGD, SignSGD, Adam, and AdamW, and Adam on an $L^2$-regularized loss are perfectly verified.

We optimize the loss $f(x) = \frac{x^\top H x}{2}$ where $H = \mathrm{diag}(1, 1)$ as we run each optimizer for $100000$ iterations with $\eta = 0.01$. We repeat this procedure five times, one for each $\sigma \in \{0.01, 0.1, 1, 10, 100\}$. As we train, we inject noise on the gradient as distributed as $\mathcal{N}(0, \sigma^2 I_d)$. We plot the average loss together with the respective limits predicted by our Lemmas. For each optimizer and each $\sigma$, the average asymptotic loss matches the predicted limit. Therefore, we verify that the loss of SGD scales quadratically in $\sigma$, that of Adam on $f(x)$, Adam on $f(x) + \frac{\theta \|x\|_2^2}{2}$, and SignSGD scales linearly, and that of AdamW is limited in $\sigma$.

## F.8 Increasing weight decay with the batch size (Figure 15 and Table 1)

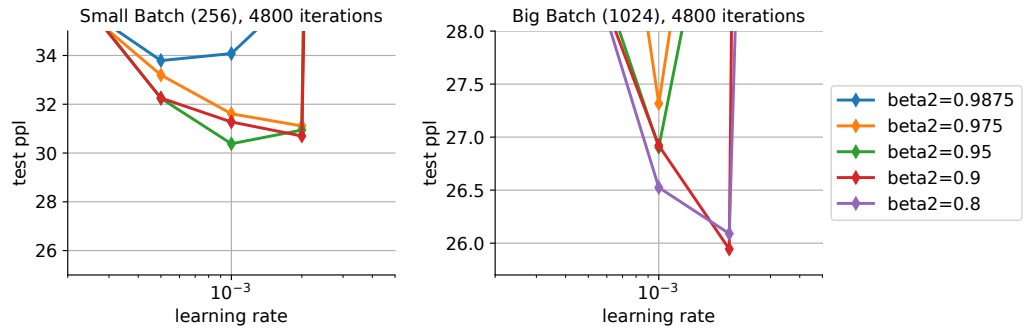

Figure 15: RMSprop Scaling. We first (left plot) tune at a small batch size (256) the learning rate and $\beta_2$ simultaneously and report all test perplexity results after cosine decay at 2.5B tokens (4800 RMSprop steps). For these runs, weight decay is turned off to eliminate confounding factors in this first analysis (see later Table 1). We then report the results for the same tuning but at a batch size of 1024 sequences. One can clearly see that the optimal learning rate shifts up from $1e-3$ to $2e-3$, as predicted by both ours and Malladi et al. (2022) scalings (factor $\delta = 4$). The best run in the base setting was with $\beta_2 = 0.95$. Our rule predicts a scaled up optimal $\beta_2$ with value $1 - \sqrt{\delta}(1 - \beta_2) = 1 - 2 \times 0.5 = 0.9$, while Malladi et al. (2022) predicts $1 - \delta(1 - \beta_2) = 1 - 4 \times 0.5 = 0.8$. The results show that while results are close (biggest effects in learning rate scaling with $\sqrt{\delta}$), our scaling is slightly more performant.

The analysis of Malladi et al. (2022) suggests that, when scaling batch size $B$ by a factor $\delta$, one also has to scale up ($\uparrow$) the learning rate $\eta$ by a factor $\sqrt{\delta}$ and scale down ($\downarrow$) $\beta_1$ to the value $1 - \delta(1 - \beta_1)$ and $\beta_2$ to the value $1 - \delta(1 - \beta_2)$. Our SDE analysis confirms similar rules (Lemma 3.13) but

1. Proposes to scale down *less* $\beta_1$ and $\beta_2$, i.e. as $1 - \sqrt{\delta}(1 - \beta)$.
2. Additionally suggests scaling up the decoupled weight decay parameter $\theta$ by a factor $\sqrt{\delta}$.

We test this in the language modeling setting utilizing a Pythia-like 160M parameter transformer architecture (Biderman et al., 2023) trained on 2.5B and 10B tokens from the SlimPajama dataset. We scale up the batch size by a factor of $\delta = 4$ and keep the same number of iterations – i.e., we have 10B tokens of training in the scaled-up runs. The sequence length in all of our experiments is 2048 tokens. We perform two experiments, and report results in Figure 15 and Table 1:

**Vanilla experiment.** In Figure 15, we set $\beta_1, \theta = 0$ and study in isolation the effects of scaling the learning rate and $\beta_2$ as the batch size increases. The results clearly indicate that scaling up the learning rate is beneficial when increasing the batch size. In addition, they indicate that our scaling of $\beta_2$ might be slightly preferable compared to the one in Malladi et al. (2022).

**Scaling weight decay.** In Table 1 we verify our scaling on AdamW runs ($\beta_1, \theta \neq 0$). We operate in a similar regime as Figure 15 and scale 9 distinct configurations at a small batch size to a big batch size using different strategies. Again, results indicate the effectiveness of our strategy.

**Remark.** While the experiments above show promise, future research is needed in order to better compare and evaluate strategies. In particular, we noticed that in non-pathological settings, it might be beneficial to not scale down $\beta_1$ with the batch size and to perhaps keep the same learning rate.

| Baseline $B = 256, \theta = 0.1$ | (Malladi et al., 2022) $B = 1024, \theta = 0.1$ | (Malladi et al., 2022) $B = 1024, \theta = 0.2$ | This paper, $B = 1024, \theta = 0.2$ |
|---|---|---|---|
| $\beta_1 = 0.975, \beta_2 = 0.9875$ 23.5889 | $\beta_1 = 0.9, \beta_2 = 0.95$ 20.6157 | $\beta_1 = 0.9, \beta_2 = 0.95$ 20.2158 | $\beta_1 = 0.95, \beta_2 = 0.975$ **19.5785** |
| $\beta_1 = 0.975, \beta_2 = 0.975$ 23.7916 | $\beta_1 = 0.9, \beta_2 = 0.9$ 20.0463 | $\beta_1 = 0.9, \beta_2 = 0.9$ 19.7162 | $\beta_1 = 0.95, \beta_2 = 0.95$ **19.3467** |
| $\beta_1 = 0.975, \beta_2 = 0.95$ 23.7461 | $\beta_1 = 0.9, \beta_2 = 0.8$ 20.0237 | $\beta_1 = 0.9, \beta_2 = 0.8$ 19.5846 | $\beta_1 = 0.95, \beta_2 = 0.9$ **19.2398** |
| $\beta_1 = 0.95, \beta_2 = 0.9875$ 23.4310 | $\beta_1 = 0.8, \beta_2 = 0.95$ 22.7750 | $\beta_1 = 0.8, \beta_2 = 0.95$ 21.3613 | $\beta_1 = 0.9, \beta_2 = 0.975$ **20.6354** |
| $\beta_1 = 0.95, \beta_2 = 0.975$ 23.3911 | $\beta_1 = 0.8, \beta_2 = 0.9$ 21.4489 | $\beta_1 = 0.8, \beta_2 = 0.9$ 20.4485 | $\beta_1 = 0.9, \beta_2 = 0.95$ **20.2158** |
| $\beta_1 = 0.95, \beta_2 = 0.95$ 23.4654 | $\beta_1 = 0.8, \beta_2 = 0.8$ 20.5648 | $\beta_1 = 0.8, \beta_2 = 0.8$ 20.3054 | $\beta_1 = 0.9, \beta_2 = 0.9$ **19.7162** |
| $\beta_1 = 0.9, \beta_2 = 0.9875$ 25.0240 | $\beta_1 = 0.6, \beta_2 = 0.95$ 1972.5442 | $\beta_1 = 0.6, \beta_2 = 0.95$ 185.4383 | $\beta_1 = 0.8, \beta_2 = 0.975$ **23.3668** |
| $\beta_1 = 0.9, \beta_2 = 0.975$ 25.1012 | $\beta_1 = 0.6, \beta_2 = 0.9$ 646.8980 | $\beta_1 = 0.6, \beta_2 = 0.9$ 42.6782 | $\beta_1 = 0.8, \beta_2 = 0.95$ **21.3613** |
| $\beta_1 = 0.9, \beta_2 = 0.95$ 23.6411 | $\beta_1 = 0.6, \beta_2 = 0.8$ 145.7700 | $\beta_1 = 0.6, \beta_2 = 0.8$ 24.2124 | $\beta_1 = 0.8, \beta_2 = 0.9$ **20.4485** |

Table 1: We perform 9 AdamW base runs at a batch size of 256 sequences of length 2048 (first column). A total of 4800 steps are performed, for a total of $2.5B$ tokens. For these runs, we always select a learning rate of 0.004 and weight decay $\theta$ of 0.1. We report results (test perplexity) for 9 different combinations of $\beta_1, \beta_2$. In the three right-most columns, we scale each setting according either to Malladi et al. (2022) or according to Lemma 3.13. Since Malladi et al. (2022) does not give prescriptions on the weight decay value $\theta$, we either scale it as we propose or leave it at 0.1. Scaled up runs process $4\times$ the number of tokens, i.e. $10B$ tokens: the algorithm performs the same number of steps but with a batch size of 1024 sequences (factor 4). Test accuracy results indicate that our scaling is more effective in terms of final test perplexity compared to (Malladi et al., 2022).

