# OpenReview forum: "Adaptive Methods through the Lens of SDEs: Theoretical Insights on the Role of Noise"
_ICLR.cc/2025/Conference — ICLR 2025 Poster_

### Official Review · Reviewer_QC2v · 2024-10-30

**Soundness:** 4
**Presentation:** 4
**Contribution:** 3
**Rating:** 8
**Confidence:** 3

**Summary:**

This work derives a novel SDE (of weak approximation order 1, under Gaussian gradient noise Z) for the SignSGD optimizer, and using this SDE, uncovers three phases of dynamics with distinct behaviours. They show theoretically that the effect of noise on SignSGD differs from SGD, and that SignSGD is more resilient to high levels of noise. Further, they show that when Z is heavy-tailed noise, the associated SDEs for SignSGD take a very similar form, further highlighting SignSGD's resilience to high levels of noise. They then derive SDEs (of weak approximation order 1) for AdamW and RMSpropW under much less restrictive assumptions than previous work, and find that decoupled weight decay has an important role in stabilization of dynamics at high noise levels. They also provide empirical validations of their theoretical results.

**Strengths:**

Results are presented very clearly and interpretations are made clear

Use numerical integrators to verify that their derived SDEs match the associated discrete-time optimizers across a diverse range of relevant setups

Technical assumptions appear much less restrictive than other works

**Weaknesses:**

Mostly theoretical, with practical implications of findings underexplored; do the findings tell you anything about how the studied optimizers can be improved, tuned, etc.?

Form of stochastic gradient appears unjustified. Is the noise Z being additive, and distributed as a Gaussian or heavy-tailed, accurate to practice?

**Questions:**

For Lemma 3.4 (line 199), since Y_t \in R^d, is the condition e.g. |Y_t| > 3/2 denoting the absolute value treated element-wise, like |(Y_t)_i| > 3/2, or does |Y_t| denote the norm of Y_t?

---

> ### Author Response · Authors · 2024-11-19
> **Many Thanks**
>
> We thank the reviewer for their very positive review.
>
> We are pleased to hear that the reviewer found our work to provide an **excellently sound** and **good contribution** towards understanding adaptive optimizers and that it was **excellently presented** with a **very clear interpretation of results**.
>
> Here is how we addressed the weaknesses and questions: we included them in the revised PDF.
>
> **W1.** *"Do the findings tell you anything about how the studied optimizers can be improved, tuned, etc.?"*: Our focus was to provide new insights into existing adaptive methods that are known to perform well in practice, even though the reasons for their effectiveness are not yet fully understood. These insights point out some important features of adaptive methods which would be important to preserve while developing new optimizers. While we do not currently have suggestions for developing other optimizers, this is certainly a direction we are actively considering.
>
> **W2.** *"Form of stochastic gradient appears unjustified. Is the noise Z being additive, and distributed as a Gaussian or heavy-tailed, accurate to practice?"*: While we acknowledge that this noise structure is not fully realistic, it is very common in the literature: See Lines 139-142 **of the original submission** for a selected collection of the references, with particular attention to Jastrzebski et al., who offer a justification to back this claim. We highlight that more complex and realistic noise structures were presented in the appendix **of the original submission** with analogous conclusions — see Section C.3 ALTERNATIVE NOISE ASSUMPTIONS.
>
>
> **Q1.** *"For Lemma 3.4 (line 199), since $Y_t \in R^d$, is the condition e.g. $|Y_t| > 3/2$ denoting the absolute value treated element-wise, like $|(Y_t)_i| > 3/2$, or does $|Y_t|$ denote the norm of $Y_t$?"*: It has to be treated element-wise and we clarified it in the updated version.

---

> > ### Comment · Reviewer_QC2v · 2024-11-25
> >
> > Thank you for addressing my questions. I maintain my score of 8.

---

> > > ### Author Response · Authors · 2024-12-02
> > > **Thank You**
> > >
> > > Dear Reviewer QC2v,
> > >
> > > Thank you for your continued support and for maintaining your score of 8. We are grateful for your thoughtful feedback and engagement throughout the review process.
> > >
> > > Best regards,
> > >
> > > The Authors

---

### Official Review · Reviewer_2DqW · 2024-11-03

**Soundness:** 2
**Presentation:** 1
**Contribution:** 2
**Rating:** 6
**Confidence:** 3

**Summary:**

The work is centered on introducing novel SDEs for adaptive optimizers. Under weak assumptions, they derive SDE for SignSGD and compare it against vanilla SGD to show three different phases in the dynamics of SignSGD. They also show an inverse relationship between noise and convergence rate for SGD which is in contrast to the quadratic effect on loss & variance observed for SGD. They also analyze decoupled weight decay and compare it against SignSGD.

**Strengths:**

The paper has introduced A rigorous mathematical framework of SDEs for SignSGD and its comparison against commonly used adaptive optimizers like ADAMW. They also characterize a relationship between noise and convergence rates for SignSGD and SGD. Also, the framework is used to understand decoupled weight decay efficacy and infer the effect of noise on such approaches. Their analysis is followed by relevant experiments.

**Weaknesses:**

Major weakness

1. Line 245-246. The definitions of 'signal' and 'noise' need to be explicitly specified in this context. The current characterization of 'signal' in these lines does not make sense. $Y_t$ line 199 defined signal-to-noise ratio but I don't see what is being referred to in lines 245-246 and what part is being referred to as "large". The entire paragraph here should be rephrased to better connect it to the surrounding content as it currently seems disconnected, or poorly worded.

2. Line 245: "SNR is large, meaning SignSGD behaves like SignGD": This claim is unsupported. You should provide the reasoning or evidence that supports this claim, such as referencing relevant equations, prior results, or additional explanation.

3. Figure 2 referred to in line 249: Hard to understand. Please provide details about the loss function/landscape, dimensionality, and any other relevant parameters used to generate Figure 2.  Lacks description of the setting used to plot figures.

4. In general, (not commenting on technical soundness), the paper is very hard to follow and poorly structured.
- Need to provide brief intuitive explanations before each lemma, highlighting its significance and key takeaways in order to understand what the authors want to convey.
- In section 3, authors should be more rigorous about introducing notation. There are unnecessary references to related work mentions which should be pushed somewhere else.  A comprehensive introduction of notation in the upcoming lemmas or some useful background should be introduced. Since the main argument is around novel SDEs, more introductory material on SDEs and their relevance to optimization algorithms before presenting the main results.

5. Section 3.1.1 - authors should ensure all new notation is properly introduced before it is used in lemmas, either by adding definitions as terms first appear or by including a notation section at the beginning of 3.1.1.

6. Line 412: Decoupled weight decay.
- Clarify why decoupled weight decay is considered key in this context
- Provide a more explicit link between their analysis and the claim about stabilization at high noise levels. Currently, the authors' analysis to support this is not evident.
- Include specific equations or results that demonstrate the stabilization effect of decoupled weight decay

7. The authors should include key experimental results in the main paper, selecting the most important figures or tables that support their main claims, while potentially moving more detailed results to an appendix if space is an issue.

Not sure how to comment on the rigor of mathematical soundness of SDEs analysis but overall it was very hard to follow what the authors wanted to show/convey.

**Questions:**

I am not quite sure what is the key message (takeaway) of SDE analysis for decoupled weight decay analysis. They are known to be more effective than their vanilla counterparts. Is there a key message here that helps improve our understanding of adaptive optimizers? Also, is the analysis valid only for convex quadratics in this case?

---

> ### Author Response · Authors · 2024-11-19
> **Many Thanks - Addressed Weaknesses and Questions**
>
> We thank the reviewer for their review.
>
> We regret that our contribution was considered only fair and poorly presented, as **this contrasts starkly** with feedback from all three other reviewers who deemed **our paper "well-written", "rich in valuable results for practitioners", and "excellently presented with clear interpretations of the findings"**. We however take this feedback seriously: We appreciate the points raised and outline below how we address these weaknesses and questions, which we incorporated into the revised PDF.
>
> Respectfully, we do not believe the weaknesses highlighted by the reviewer to be **substantial**: They can be addressed through rewording or additional clarifications. **None of the mentioned concerns compromise the validity, generality, or impact of our analysis.**
> We have already updated the PDF incorporating the feedback of the reviewer (see discussion below) and are happy to discuss further with the reviewer if some concerns remain.
>
> Here is how we addressed the weaknesses and questions: we included them in the revised PDF.
>
> ## Weaknesses
>
> **W1.** While this informal definition was **inspired by  Bernstein et al. (2018)** (who first provided an extended theoretical analysis of SignSGD), introducing the SNR was intended to support the reader's intuition and was not meant as a formal concept. We have reworded the mentioned paragraphs accordingly.
>
> **W2.** **On the contrary**, this **claim** is in fact **supported**: Eq. 4 shows that if the SNR is large (e.g., the gradient is much larger than its noise), the SDE of SignSGD becomes the classical ODE of SignGD. The left and center-left sides of Fig. 2 visually support this: Thank you for pointing this out --- We have reworded the mentioned paragraph accordingly.
>
> **W3.** While we initially omitted certain details to avoid overloading the paper with technical information that could affect readability, we have added some level of detail to the caption.
>
> **W4 and W5.** The relevance of SDEs in optimization was **already** discussed in a dedicated (large) paragraph in the Related Works while Appendix B **already** provided introductory material on SDEs. To address your point regarding the notation, we have expanded the dedicated paragraph.
>
> Regarding the interpretation of our results, we find this comment puzzling. As highlighted by other Reviewers, our results were **already** complemented by a paragraph where we interpret them, even in dedicated “Remarks”, or “Conclusions” in gray boxes, specifically **providing brief intuitive explanations**. For instance, Reviewer QC2v stated that "**Results are presented very clearly and interpretations are made clear**". Below is a guide indicating where each result was interpreted. All mentioned lines refer to the **original submission**:
>
> 1. Lemma 3.4: This result is meant to break down and make Corollary 3.3 and Theorem 3.2 more digestible: Its interpretation was in lines 230-234 and visually supported in Fig. 1 and Fig. 2 (left).
>
> 2. Lemma 3.5 and Lemma 3.6: Their interpretation was in lines 245-250 together with the **Remark** in lines 254-261. Visually supported in Fig. 2 (center-left).
>
> 3. Lemma 3.7 and Lemma 3.8: Their interpretation was in lines 277-279. Visually supported in Fig. 2 (center-right).
>
> 4. Lemma 3.9: Its interpretation was in lines 280-283. Visually supported in Fig. 2 (right).
>
> 5. **We summarized Section 3** with a **Conclusion** to collect the findings of the section.
>
> 6. Lemma 3.10 and Corollary 3.11 had a dedicated **Conclusion** where we interpreted them in lines 317-322.
>
> 7. Lemma 3.13: We introduced it in lines 362-370 and interpreted it in lines 406-409. Visually supported in Fig. 5.
>
> 8. Lemma 3.14: We interpreted it in the **Conclusion** in lines 447-453. Visually supported in Fig. 5 (right).
>
> We now added a dedicated paragraph to Theorem 3.12, which we recognized was not properly interpreted, but visually well-supported in Fig. 3 and 4.
>
>
> **W6.**
> 1. Let us recall that AdamW is defined as Adam + **decoupled** weight decay: Since we study such an optimizer, we need to study how decoupled weight decay affects the dynamics of AdamW as decoupled weight decay **plays a fundamental role in modern LLMs** [1,2].
>
> 2. **On the contrary**, this **claim** is, in fact, **supported**: Eq. 13 shows that the bound on the expected loss of AdamW is itself bounded with respect to the noise level $\sigma$. This was **clearly stated** in lines 415, 442, 449, and 479, and we have further clarified this point in the updated version.
>
> 3. As mentioned, this was provided in Eq. 13. In contrast, Eq. 14 showed that Adam on the $L^2$-regularized loss function does not enjoy such a stabilization: See Lines 412 to 420 of the **original submission**.
>
> [1] D'Angelo, Francesco, et al. "Why Do We Need Weight Decay in Modern Deep Learning?." NeurIPS 2024.
>
> [2] Xiao, Lechao. "Rethinking Conventional Wisdom in Machine Learning: From Generalization to Scaling.", 2024

---

> > ### Author Response · Authors · 2024-11-19
> > **Many Thanks - Continuation**
> >
> > ## Continuation of Rebuttal on Weaknesses
> >
> > **W7.** We find this point rather puzzling. **Each of our results is supported by a specific figure illustrating it**. We conducted experiments across a range of architectures and datasets, including quadratic functions, embedded saddles, MLPs, Transformers, ResNets, and CNNs: We have already **mapped** each result to its illustrating figure as we replied to **W4 and W5**. Could you please clarify if there is a particular result that you feel lacks visual support?
> >
> > ## Questions
> >
> > **Q1**. *"I am not quite sure what is the key message (takeaway) of SDE analysis for decoupled weight decay analysis. They are known to be more effective than their vanilla counterparts."*:
> > While AdamW and RMSpropW are indeed **experimentally** known to be more effective than their vanilla counterparts, it is currently **theoretically** unclear why. The key message of the SDE analysis for decoupled weight decay is that it elucidates **how these optimizers are more resilient to high gradient noise, thanks to the stabilizing effect of decoupled weight decay**. By understanding the role of this hyperparameter, practitioners can make more informed decisions when tuning it, potentially leading to better performance, especially in noisy environments common in deep learning. Other reviewers **have recognized** this as an **important** contribution, and we would greatly appreciate it if the reviewer could reconsider their evaluation. We are happy to clarify specific points in the writing if the reviewer would like to share additional feedback.
> >
> > **Q2.** *"Is there a key message here that helps improve our understanding of adaptive optimizers?"*
> > Yes, our work demonstrates that **adaptive optimizers can better handle high levels of noise**. For SignSGD, this is because its performance depends linearly on the noise level, unlike SGD, which has a quadratic dependence (therefore the same level of noise can be more detrimental to its convergence). For Adam, we show that its behavior with respect to noise is similar to SignSGD, while for AdamW, the decoupled weight decay further strengthens this resilience. **As recognized by the other Reviewers**, these insights are novel, theoretically proven, empirically validated, and discussed thoroughly in the main paper.
> >
> > **Q3.** *"Also, is the analysis valid only for convex quadratics in this case?"*
> > This is an important question to which the answer is no: **none** of our SDEs are limited to quadratic functions. The only results specifically on quadratic functions are Lemma 3.7 and Lemma 3.14, where we derive stationary distributions calculated on quadratic functions, in line with the existing literature (see lines 142 to 145 of the **original submission** for references). All other results in the main paper apply to **general strongly convex** functions. Importantly, we also present a **generalization to non-convex smooth functions** in the Appendix (see Lemma C.17 and Lemma C.50).

---

> > > ### Comment · Reviewer_2DqW · 2024-11-25
> > >
> > > I thank the authors for their responses.
> > >
> > > I have increased my score to 6 after reading through the authors' responses to the issues raised. I am unwilling to increase the scores any further since I stand by my main issue of readability/accessibility of the work. It might not be a substantial issue for the authors but I had a hard time reading the paper (this is about the general paper outline and presentation. I am not alluding to the mathematical component or the novelty of the work which I already believe is good).

---

> > > > ### Author Response · Authors · 2024-12-02
> > > > **Thank You So Much**
> > > >
> > > > Dear Reviewer 2DqW,
> > > >
> > > > Thank you for your thoughtful engagement and for increasing your score after reviewing our responses. We appreciate your feedback, which we believe has already helped improve the clarity of our manuscript.
> > > >
> > > > Best regards,
> > > >
> > > > The Authors

---

### Official Review · Reviewer_ahk4 · 2024-11-07

**Soundness:** 3
**Presentation:** 3
**Contribution:** 3
**Rating:** 8
**Confidence:** 3

**Summary:**

This paper investigates the learning dynamics of adaptive optimizers in machine learning.  The authors analyze a simplified model, $f(x) = x^THx$, with noisy gradients defined as $g_\gamma = \nabla f(x) + \gamma$, where $\gamma$ is sampled from a distribution like a Gaussian. Focusing on SGD, sign-SGD,  and AdamW, they demonstrate that the learning dynamics can be effectively approximated by a stochastic differential equation (SDE) of the form $X_t = \text{function}(X_t)$. This allows them to analyze the first and second moments of $X_t$ for these optimizers as $t$ approaches infinity.  The authors support their theoretical findings with experiments on practical models like ResNets and small Transformers, trained on MNIST, CIFAR, and Shakespeare datasets.

While generally well-written, mathematically sound and provide many insights, the paper's reliance on a simplified model and unrealistic noise design limits its applicability.


--- updates ---

Thank you for the clarifications. While I believe this work warrants a score of 7, I understand that's not an available option.  Considering that other reviewers have mostly given scores of 6, I am raising my score to 8 to better reflect the paper's merit.

**Strengths:**

- generally well-written

- mathematically seems sound, though i didn't check the proof

- papper contains rich set of results that may be insightful for practicioners.

- theoretical results are supported by real networks, though on

**Weaknesses:**

* The toy model is too simple and far from practical. I would expect at least a linear regression model with a random (Gaussian) design.
* The noise design is artificial and unrealistic. It doesn't capture practical batch noise, which depends on the loss, parameters, and changes over time. The current noise function uses a time-independent, identical distribution.
*  Experiments use full batch with synthetic (Gaussian) noise instead of mini-batches, which is consistent with the paper's problem setup but unrealistic.
* The paper is overly dense, with little intuition provided for the theorems and proofs. It also relies heavily on in-line equations, hindering readability.

**Questions:**

* The notation $\nabla f_\gamma$ is unclear and could be misinterpreted as $\nabla (f_\gamma)$.
* All algorithms in the paper use full batch gradients with added noise, which is misleading and doesn't accurately reflect real-world implementations of algorithms like Adam.
* It would be beneficial to generalize the results to mini-batch linear regression with random Gaussian design, similar to the approach in https://arxiv.org/abs/2405.15074.
* The paper should provide more intuition behind the proof of Theorem 3.2 to help readers understand the key ideas without having to delve into the appendix.

---

> ### Author Response · Authors · 2024-11-19
> **Many Thanks - Addressed Weaknesses and Questions**
>
> We thank the reviewer for their very positive review.
>
> We are pleased to hear that the reviewer found our work 1) to provide a **"well-written, mathematically sound, [and] good contribution"** towards understanding adaptive optimizers; 2) to show **"a rich set of results"** that are **"insightful for practitioners"**; and 3) to **support** the theoretical results with experiments on a variety of DNNs.
>
> Here is how we addressed the weaknesses and questions: we included them in the revised PDF.
>
> ## Weaknesses
>
> 1. *"The paper's reliance on a simplified model and unrealistic noise design limits its applicability."*: We would like to clarify that **contrary to the reviewer's statement**, **none** of our SDEs derivation is restricted to quadratic functions or Gaussian noise: The theory applies to general smooth functions and general noise structures (see Assumption C.3 and lines 134-142 **of the original submission**, respectively). **Regarding the applicability of our theory**, we highlight, for instance, that although our novel scaling rule for AdamW was derived for strongly convex functions, **we validated its effectiveness on a Pythia-like 160M LLM trained on $2.5B$ tokens from the SlimPajama dataset** (see Appendix F.8 and Figure 14 **of the original submission**): This demonstrates the broad applicability of our theory beyond the theoretical setting.
>
> 2. *"The toy model is too simple and far from practical."*: We highlight that **the only results** specific for quadratic functions are Lemma 3.7 and Lemma 3.14, where we derive stationary distributions, which are calculated on quadratic functions **in line with the existing literature** (see lines 142 to 145 **of the original submission** for references, especially [5,7,8,9,10] whose joint citation count is 3000+). All other results are valid for general strongly convex functions. For the sake of simplicity and to avoid crowding the main paper, we already included their **generalization to general non-convex smooth functions** in the Appendix (see Lemma C.17 and Lemma C.50 **of the original submission**).
>
> 3. *"The noise design is artificial and unrealistic. It doesn't capture practical batch noise, which depends on the loss, parameters, and changes over time. The current noise function uses a time-independent, identical distribution."*: While we agree that this is an artificial design, this assumption is **very common and accepted in the literature**, especially that of SDEs for optimization (see lines 139-142 **of the original submission** for reference, especially [1,2,3,4,5,6,7] whose joint citation count is 1700+). We employ this design in the main paper only for **didactical** reasons, but more complex and realistic noise structures are presented in the appendix with analogous conclusions — see Section C.3 ALTERNATIVE NOISE ASSUMPTIONS **of the original submission**: It is quite rare to tackle these many noise structures, thus we are more complete than other papers. Therefore, we kindly disagree with the reviewer: Our setup is **not restrictive and is more general or, at least, in line with the literature**.
>
> 4. *"Experiments use full batch with synthetic (Gaussian) noise instead of mini-batches, which is consistent with the paper's problem setup but unrealistic."*: As we discussed above, these assumptions are widely accepted in the literature (see lines 139-142 **of the original submission** for reference, especially [1,2,3,4,5,6,7] whose total citation count is 1700+) and already bring novel insights into the dynamics of these optimizers.
>
> [1] Li, Qianxiao, et al. "Stochastic modified equations and adaptive stochastic gradient algorithms." *ICML*, 2017.
>
> [2] Mertikopoulos, Panayotis, et al. "On the convergence of gradient-like flows with noisy gradient input." *SIAM Journal on Optimization* 2018.
>
> [3] Raginsky, Maxim, et al. "Continuous-time stochastic mirror descent on a network: Variance reduction, consensus, convergence." *CDC* IEEE, 2012.
>
> [4] Zhu, Zhanxing, et al. "The anisotropic noise in stochastic gradient descent: Its behavior of escaping from sharp minima and regularization effects." *ICML* 2019.
>
> [5]  Mandt, Stephan, et al. "A variational analysis of stochastic gradient algorithms." *ICML* 2016.
>
> [6] Ahn, Sungjin, et al.. "Bayesian posterior sampling via stochastic gradient Fisher scoring." *ICML,* 2012.
>
> [7] Jastrzebski, Stanislaw, et al. "Three factors influencing minima in SGD." *ICANN,* 2018.
>
> [8] Ge, Rong, et al. "Escaping from saddle points—online stochastic gradient for tensor decomposition." *CLT*, 2015.
>
> [9] Levy, Kfir Y. "The power of normalization: Faster evasion of saddle points." (2016).
>
> [10] Jin, Chi, et al. "How to escape saddle points efficiently." *ICML*, 2017.
>
> [11] Xie, Zeke, et al. "Positive-negative momentum: Manipulating stochastic gradient noise to improve generalization." *ICML*, 2021.

---

> > ### Author Response · Authors · 2024-12-03
> > **Thank You So Much**
> >
> > Dear Reviewer ahk4,
> >
> > Thank you for increasing your score from **6** to **8** and for recognizing the merit of our work. We also sincerely appreciate your feedback and the challenging questions you raised, which have helped us strengthen our paper.
> >
> > Best regards,
> >
> > The Authors

---

> ### Author Response · Authors · 2024-11-19
> **Many Thanks - Continuation**
>
> ## Continuation of Rebuttal on Weaknesses
>
> 5. *"The paper is overly dense, with little intuition provided for the theorems and proofs. It also relies heavily on in-line equations, hindering readability."*: We highlight that we already complemented our results with interpretations in **the original submission** (see reply to **Reviewer 2DqW** for a precise guide indicating where each result is interpreted). To address your point, we added the proof to three key results: Other results follow similar strategies. Additionally, we reduced the use of in-line equations and used the full $10$ pages to reduce the density of the paper.
>
> ## Questions
>
> **Q1**. While this is the notation of the seminal paper by Li et al. (2017) and has become standard in the literature, we have redefined this in our paper as well.
>
> **Q2**. See the answer to Weakness 4 above. We add here that the reason why we inject noise with this structure is that we want to **empirically verify our SDEs** on a variety of neural architectures and datasets: To our knowledge, **we are the first to take on such a challenge** to this extent --- We welcome future work to improve our initial effort in this direction.
>
> **Q3**. Please find our insights for this setup below and the details in Appendix C.5 **of the updated version** together with **other additional** alternative (realistic) noise structures.
>
> **Q4**. We added a sketch of the proof of some key results to guide the reader's intuition.
>
> ## Linear Regression with Gaussian design
>
> In this section, we adopt the notation of Paquette et al.. As per Eq. 5 of Paquette et al., the loss function can be rewritten as
>
> \begin{equation}
>     f(\theta)= \frac{1}{2} \langle D(W \theta-b),(W \theta-b)\rangle, \quad \text { where } D=\operatorname{diag}\left(j^{-2 \alpha}\right) \in \mathbb{R}^{v \times v}, \text{ and} 1 \leq j \leq v.
> \end{equation}
>
> Without loss of generality, we define $\phi:= W \theta-b$, which implies that
>
> \begin{equation}
>     f(\theta)= \frac{\phi^{\top} D \phi}{2}, \quad \text { where } D=\operatorname{diag}\left(j^{-2 \alpha}\right) \in \mathbb{R}^{v \times v}, \text{ and } 1 \leq j \leq v.
> \end{equation}
>
> The stochastic gradient is unbiased and its covariance is
>
> $$B \Sigma(\phi) = (\phi^{\top} D \phi) D + D \phi \phi^{\top} D = 2 f(\phi) D + \nabla f(\phi) \nabla f(\phi)^{\top},$$
> where $B$ is the batch size.
>
> Under this assumption, we have that for $Y_t := \frac{\sqrt{B}(\Sigma(\phi_t))^{-\frac{1}{2}}\nabla f(\phi_t)}{\sqrt{2}}$ and $\mathcal{S}(\phi_t)=\mathbb{E}[(Sign(\nabla f_{\gamma}(\phi_t))(Sign(\nabla f_{\gamma}(\phi_t))^{\top}]$ the SDE of SignSGD becomes:
>
> \begin{align}
> d \phi_t = - Erf \left( Y_t \right) dt + \sqrt{\eta} \sqrt{\mathcal{S}(\phi_t) - Erf \left(Y_t \right) Erf \left(Y_t \right)^{\top}} d W_t.
> \end{align}
>
> As a consequence, Lemma 3.5 becomes:
>
> Let $f$ be as above, $f_t:=f(\phi_t)$, and $\mathcal{L}_{\tau} := Tr(D)$. Let $\mu$ be the minimum eigenvalue of $D$, and $L$ be its maximum one. Then, during
> 1. Phase 1, the loss will reach $0$ before $t_* = 2 \sqrt{\frac{S_0}{\mu}}$ because $f_t \leq \frac{1}{4} \left( \sqrt{\mu}t - 2 \sqrt{f_0}\right)^2$;
> 2. Phase 2 with $\beta := \frac{\eta}{2}  \mathcal{L}_{\tau}$ and $\alpha:=  \frac{ m \mu \sqrt{B}}{\sqrt{2 L}}$,
> \begin{equation}
> \mathbb{E}[S_t] \leq \frac{\beta^2 \left( \mathcal{W}\left( \frac{(\beta + \sqrt{S_0} \alpha)}{\beta} \exp\left(-\frac{\alpha^2 t - 2 \sqrt{S_0} \alpha}{2 \beta} - 1 \right) \right) + 1 \right)^2}{\alpha^2} \overset{t \rightarrow \infty}{\rightarrow} \frac{\beta^2}{\alpha^2};\end{equation}
> 3. Phase 3 with $\beta := \frac{\eta}{2}  \mathcal{L}_{\tau}$ and $\alpha:= \sqrt{\frac{2}{\pi}} \frac{\mu \sqrt{B}}{\sqrt{L}}$,
> \begin{equation}
> \mathbb{E}[S_t] \leq \frac{\beta^2 \left( \mathcal{W}\left( \frac{(\beta + \sqrt{S_0} \alpha)}{\beta} \exp\left(-\frac{\alpha^2 t - 2 \sqrt{S_0} \alpha}{2 \beta} - 1 \right) \right) + 1 \right)^2}{\alpha^2} \overset{t \rightarrow \infty}{\rightarrow} \frac{\beta^2}{\alpha^2},
> \end{equation}
> where $\mathcal{W}$ is the Lambert $\mathcal{W}$ function.

---

> > ### Author Response · Authors · 2024-12-02
> > **Friendly Reminder**
> >
> > Dear Reviewer ahk4,
> >
> > We sincerely appreciate your earlier comments and believe we have addressed your concerns in both the rebuttal and the revised manuscript. We would greatly appreciate it if you could take a moment to review our rebuttal.
> >
> > Best regards,
> >
> > The Authors

---

### Official Review · Reviewer_2nWX · 2024-11-08

**Soundness:** 3
**Presentation:** 3
**Contribution:** 3
**Rating:** 6
**Confidence:** 3

**Summary:**

This paper presents a theoretical analysis of adaptive optimization methods—specifically SignSGD, AdamW, and RMSpropW—through the lens of stochastic differential equations (SDEs), unveiling the intricate interplay between adaptivity, gradient noise, and curvature. Key contributions include:

- **SignSGD Dynamics:** Identification of three distinct phases in SignSGD’s dynamics, with noise inversely affecting both convergence rates of the loss and the iterates.
- **Enhanced SDE Models:** Derivation of new and improved SDEs for AdamW and RMSpropW, leading to a novel batch size scaling rule and an examination of the stationary distribution and stationary loss value in convex quadratic settings.

The derivation of new SDEs for SignSGD, AdamW, and RMSpropW provides a solid foundation for understanding the dynamics of these optimizers.

**Strengths:**

Strengths includes:
- **Intriging Insights:** Identifying three distinct phases in SignSGD dynamics and the inverse relationship between noise and convergence rates offer valuable perspectives on optimizer behavior.
- **Practical Implications:** Introducing a novel batch size scaling rule and examining stationary distributions have direct implications for better training practices in deep learning.
- **Clear background:** The preliminaries and background information provided in the appendix enhance the readability and understanding of the main results.
- **Validation:** Theoretical findings are supported by extensive experimental evidence across various neural network architectures, including MLPs, CNNs, ResNets, and Transformers.

**Weaknesses:**

- **Unclear Terminology and Notation:** Certain terms and symbols used in the paper are not clearly defined, which may lead to confusion. What does $\mathbb{P}$ represent in Theorem 3.2? How is the error function defined in Corollary 3.3? Does Lemma 3.7 explicitly establish the stationary distribution of SignSGD?

**Questions:**

See weaknesses above.

---

> ### Author Response · Authors · 2024-11-19
> **Many Thanks**
>
> We thank the reviewer very much for their dedication to the review process and for taking the time to carefully study our manuscript.
>
> We are pleased to hear that the reviewer found our work to provide a "**solid foundation for understanding the dynamics of these optimizers**", providing "**intriguing insights**", with "**direct implications for better training practices in deep learning**". Importantly, the reviewer noted that the **presentation is good**, with a **"clear background"** that "**enhances the readability and understanding of the main results**", which are "**supported by extensive experimental evidence across various neural network architectures**".
>
> Here is how we addressed the weaknesses in the updated PDF:
>
> 1. *"What does $\mathbb{P}$ represent in Theorem 3.2?"*: As per Appendix B, all our analyses take place on a probability space $\left(\Omega, \mathcal{F}, \mathbb{P}\right)$: Therefore, $\mathbb{P}$ represents the probability measure of the probability space --- We added this to the paragraph dedicated to notation. In that equation, we wanted to measure the probability that the **stochastic** gradient is (element-wise) smaller than $0$.
>
> 2. *"How is the error function defined in Corollary 3.3?"*: The error function (also called the Gauss error function) is a function erf: $\mathbb{C} \rightarrow \mathbb{C}$ defined as:
>
>    $$
>    \operatorname{erf} z=\frac{2}{\sqrt{\pi}} \int_0^z e^{-t^2} \mathrm{~d} t.
>    $$
>
>    Despite it being a known object in probability, we included its definition in the manuscript for clarity.
>
>
> 3. *"Does Lemma 3.7 explicitly establish the stationary distribution of SignSGD?"*: Yes. Point 1 of Lemma 3.7 shows that the expected value of the iterates converges to $0$. Point 2 of the same lemma shows that the covariance matrix of the iterates converges to $\frac{\eta}{2} \left( \sqrt{\frac{2}{\pi}} I_d + \frac{\eta}{\pi} H \Sigma^{-\frac{1}{2}}\right)^{-1} H^{-1} \Sigma^{\frac{1}{2}}$. Therefore, we have that the stationary distribution of SignSGD is:
>
> $$
> \left(\mathbb{E} [X_\infty], Cov[X_\infty]\right)=\left(0, \frac{\eta}{2} \left( \sqrt{\frac{2}{\pi}} I_d + \frac{\eta}{\pi} H \Sigma^{-\frac{1}{2}}\right)^{-1} H^{-1} \Sigma^{\frac{1}{2}}  \right).
> $$
>
>
> We clarified this in the updated version.

---

> > ### Comment · Reviewer_2nWX · 2024-11-25
> > **Reviewer Feedback Update**
> >
> > Thank you for your detailed response. After a more careful review of the manuscript, the reviewer plans to maintain a positive rating for the paper, recognizing the significance of its contributions. However, due to current time constraints, the reviewer regrets being unable to provide further detailed technical feedback at this time. Your kind understanding will be appreciated.

---

> > > ### Author Response · Authors · 2024-12-02
> > > **Thank You**
> > >
> > > Dear Reviewer 2nWX,
> > >
> > > Thank you for the careful review of our manuscript,  for confirming the positive rating, and for recognizing the significance of our contributions. We truly appreciate the thoughtful feedback you have provided and understand the time constraints: We are grateful for your engagement.
> > >
> > > Best regards,
> > >
> > > The Authors

---

### Author Response · Authors · 2024-11-19
**General Answer**

Dear Reviewers and AC,

We sincerely appreciate your thorough reviews, insightful comments, and interesting questions regarding our paper: Your feedback has greatly contributed to the finalization of our work.


We are pleased that three out of four reviewers found our paper to be "**well-presented**" and to provide a "**solid (mathematical) foundation**" for understanding the dynamics of SignSGD, Adam(W), and RMSprop(W). These reviewers appreciated our theoretical contributions which, quoting their words, "**offer valuable perspectives on optimizer behavior**" and "**have direct implications for better training practices in deep learning**", noting that our "**intriguing**" findings are "**well-supported by extensive experimental evidence across various architectures**" and datasets. Reviewer QC2v described our **presentation as excellent**, highlighting the clarity of our results and interpretations (e.g. stating "**Results are presented very clearly and interpretations are made clear**").

We of course take the criticisms very seriously and we devoted time to addressing them below. Specifically, the weaknesses and questions mainly concern relatively **minor issues** around notation, the need for some rewording, the request for some additional technical details, and the intuition behind some proofs. All of these are properly taken care of in the updated version.

Finally, all concerns regarding assumptions or simplistic setups have been properly addressed: We referred reviewers to very well-cited papers from the literature that use our same (if not more restrictive) assumptions. Importantly, we added the study of a linear regression model with a random Gaussian design (suggested by Reviewer ahk4) to showcase the flexibility and generality of our approach.

To highlight the (relevant) changes made to the original submission, we marked them using **blue** in the revised version.
Once again, we are thankful to the reviewers for their constructive feedback. We look forward to the upcoming author-reviewer discussion period.

Thank you for your attention.

Best regards,

The Authors

---

### Author Response · Authors · 2024-11-24
**Kind Reminder**

Dear Reviewers,

As the discussion period approaches its conclusion, we kindly encourage your engagement in providing feedback or addressing any points from our rebuttal.

We greatly appreciate your time and look forward to continuing this discussion.

Best regards,

The Authors

---

### Meta-Review · Area_Chair_mDCA · 2024-12-20

**Metareview:**

This paper introduces novel stochastic differential equations (SDEs) to model adaptive optimization methods like SignSGD, AdamW, and RMSpropW, providing insights into their dynamics and the role of noise, adaptivity, and curvature in convergence. The authors support their theoretical findings with experiments on various neural network architectures, demonstrating the accuracy of their SDE models compared to previous approaches. A weakness is the reliance on simplified or unrealistic models and noise assumptions in some analyses, which may limit the direct applicability to real-world deep learning scenarios; however, the strength lies in the introduction of a rigorous SDE framework for analyzing adaptive optimizers, supported by empirical validation, offering a deeper understanding of their behavior and potentially informing improved training practices. Given these considerations, I recommend accepting this paper, as it provides valuable insights into adaptive optimization methods.

**Additional Comments On Reviewer Discussion:**

Reviewers felt the model's reliance on simplified functions and unrealistic noise assumptions limited its real-world applicability, also questioning the use of full-batch experiments. There were also concerns that the paper was dense and difficult to follow due to a lack of intuitive explanations for theorems, heavy use of in-line equations, and unclear notation/definitions in certain areas. The authors clarified their SDEs apply broadly beyond quadratic functions and Gaussian noise, validated their theory on a large language model, and defended their noise assumptions as standard in the field, with more complex noise models explored in the appendix. They also added proof sketches, reduced in-line equations, and expanded the notation section, while also pointing out that interpretations of results were already present in the original submission.

After the discussion period, some concerns remained, particularly about readability, but in the end all reviewers agreed that the paper should be accepted.

---

### Decision · Program_Chairs · 2025-01-22

Accept (Poster)